# Entropy-Guided Sampling of Flat Modes in Discrete Spaces

## Abstract

Sampling from flat modes in discrete spaces is a crucial yet under-explored direction. Flat modes represent robust solutions and have broad applications in combinatorial optimization and discrete generative modeling. However, existing sampling algorithms often overlook the mode volume and struggle to capture flat modes effectively. To address this limitation, we propose *Entropic Discrete Langevin Proposal* (EDLP), which incorporates local entropy into the sampling process through a continuous auxiliary variable under a joint distribution. The local entropy term guides the discrete sampler toward flat modes with a small overhead. We provide non-asymptotic convergence guarantees for EDLP in locally log-concave discrete distributions. Empirically, our method consistently outperforms traditional approaches across tasks that require sampling from flat basins, including Bernoulli distributions, restricted Boltzmann machines, combinatorial optimization, and binary neural networks.

## 1 Introduction

Sampling in discrete spaces is an integral component of machine learning, underpinning a diverse range of applications. Tasks such as text generation in natural language processing (NLP) (Gu et al., 2018; Devlin et al., 2019; Lewis et al., 2020), sequence alignment in bioinformatics (Shendure & Ji, 2008), the development of low-precision neural networks (Hubara et al., 2017; Peters & Welling, 2018), data generation in discrete Generative Adversarial Networks (GANs) (Goodfellow et al., 2014), and policy-based action sampling in reinforcement learning (Sutton & Barto, 2018) all rely on robust discrete sampling techniques. Recent advancements in gradient-based methods have significantly enhanced the efficiency of discrete samplers in exploring complex and multimodal landscapes, leveraging gradient-information, setting new benchmarks for prob-

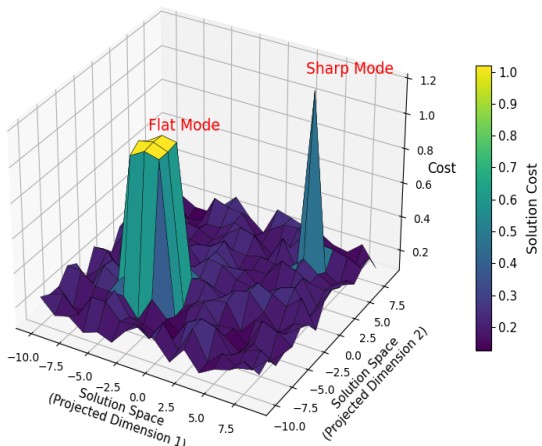

Figure 1: Cost landscape visualization on TSP: flat modes imply robust solutions under budget, whereas sharp modes are highly sensitive to small changes, leading to abrupt cost increases.

abilistic inference and combinatorial optimization (Grathwohl et al., 2021; Zhang et al., 2022; Rhodes & Gutmann, 2022; Sun et al., 2022; 2023; Li & Zhang, 2025).

While the pursuit of flat minima (regions where neighboring states have similar energies) has revolutionized continuous neural network optimization and robustness (Hochreiter & Schmidhuber, 1997; Arbel et al., 2021; Izmailov et al., 2021; Chaudhari et al., 2019; Chang & Khanna, 2025), the geometry of discrete landscapes remains largely untouched. Through our work, we address this direction and argue that flatness is essentially desirable and beneficial for reasons that are distinct to the discrete topology. For instance, in constrained combinatorial optimization tasks, a "flat mode" represents a large cluster of valuable, distinct solutions that are structurally similar and have nearly identical costs within a given budget, which are of special

interest in vehicle-routing optimization (Laporte, 2009; Toth & Vigo, 2002) . Figure 1 provides a conceptual, low-dimensional projection of such a cost landscape for the Traveling Salesman Problem (TSP), a problem involving categorical variables.

In this paper, we aim bridge two fields by proposing *Entropic Discrete Langevin Proposal* (EDLP), that incorporates the concept of flatness-aware local entropy (Baldassi et al., 2016) into Discrete Langevin Proposal (DLP) (Zhang et al., 2022). By coupling discrete and flat-mode-guided variables, we obtain a broader, entropy-informed joint target distribution that biases sampling towards discrete flat modes under a computational budget. Specifically, while updating the primary discrete variable using DLP, we simultaneously perform continuous Langevin updates on the auxiliary variable. Through the interaction between discrete and auxiliary variables, the discrete sampler will be steered toward flat regions before full equilibration. We summarize our core contributions as follows:

- We propose Entropic DLP (EDLP), a novel entropy-guided, gradient-based proposal for sampling from discrete flat basins. EDLP efficiently incorporates local entropy guidance by coupling discrete and continuous variables within a joint distribution (Section 4).
- We provide non-asymptotic convergence guarantees for EDLP in locally log-concave distributions. To our knowledge, we are the first to offer such bound for unadjusted gradient-based discrete sampling (Theorem 5.5 in Section 5).
- Through extensive experiments, we demonstrate that EDLP outperforms existing discrete samplers, with improvements up to 88.7%, in capturing flat-mode configurations across various tasks, including restricted Boltzmann machines, combinatorial optimization, and binary Bayesian neural networks (Section 6). We release the code at `https://anonymous.4open.science/r/EDLP-C0E8/README.md`.

Additionally our method remains scalable in $d$ as gradient modification is minimal (Section 4.1) and we do provide runtime analysis to highlight this (Fig 5). We also provide a Gibbs-like-Update (Appendix C) version of our algorithm, that reduces runtime by $\approx 50\%$.

## 2 Related Works

**State Augmentation in Sampling.** While state augmentation is quite common in Markov Chain Monte Carlo(MCMC), EDLP positions itself uniquely against its predecessors. Hamiltonian Monte Carlo (HMC)(Ch 5. in Brooks et al. (2011))introduces an auxiliary continuous momentum variable to leverage inertia for efficient exploration. In EDLP, the auxiliary variable represents a continuous position, not to simulate inertia, but to smooth the energy landscape locally—effectively simulating a Moreau envelope to guide the discrete sampler toward flat basins. Also, HMC targets the marginal, whereas EDLP deliberately samples from an entropy-biased joint distribution to prioritize flatness guidance. Similarly, Slice Sampling (Neal, 2003) uses the auxiliary height variable to adapt the step size to the local flatness for better mixing, whereas EDLP uses its auxiliary variable to bias the stationary distribution itself. Compared to global flattening techniques like Parallel Tempering (Geyer, 1991), which require simulating multiple chains at different temperatures, EDLP achieves local smoothing via Gaussian convolution with a single chain. Unlike Swendsen-Wang (Swendsen & Wang, 1987), which augments the state with discrete bonds for cluster updates, EDLP utilizes gradient-based guidance from discrete-continuous coupling for efficient exploration of flat basins. EDLP also shares a fundamental mathematical structure with Proximal Sampling (Pereyra, 2016) through a quadratic auxiliary coupling that mirrors the Moreau-Yosida envelope. In this view, EDLP functions as a proximal sampler on a restricted discrete domain. We further draw parallels between proximal samplers and EDLP in Section 5.

**Gradient-Based Discrete Sampling.** Gradient-based methods have significantly improved sampling efficiency in discrete spaces. Locally informed proposals method by Zanella (2020) leverages probability ratios to explore discrete spaces more effectively. Building on this, Grathwohl et al. (2021) introduced a gradient-based approach to approximate the probability ratio, further improving sampling efficiency. Discrete Langevin Proposal (DLP), introduced by Zhang et al. (2022), adapts the principles of the Langevin algorithm (Grenander & Miller, 1994; Roberts & Tweedie, 1996; Roberts & Stramer, 2002), originally designed for continuous spaces, to discrete settings. This algorithm enables parallel updates of multiple coordinates using a single gradient computation, boosting both computational efficiency and scalability. EDLP builds on the DLP proposal mechanism, enabling EDLP to inherit DLP's multi-coordinate update for efficient mixing.

However, it does so using the coupled energy instead. The proposal therefore sees a smoothened version of the original energy landscape drawing locally-balanced moves towards flat modes by construction, rather than by an external reweighting or post-hoc selection.

**Gaussian Auxiliary Variable in Discrete Sampling.** Using a continuous Gaussian auxiliary variables to simplify sampling in discrete spaces is a well-known trick in statistical computing. This technique has a rich history, from its use in Bayesian data augmentation (Albert & Chib, 1993) to linearizing quadratic terms in energy-based models for efficient Gibbs sampling (Martens & Sutskever, 2010). More recently, this framework has been central to developing modern, general-purpose gradient-based samplers for discrete variables (Pakman & Paninski, 2013; Rhodes & Gutmann, 2022).

**Flatness in Discrete Spaces.** Local Entropy is the canonical metric for discrete flatness used in Focused Belief Propagation(fBP)( Eq. 2 in (Baldassi et al., 2016)). So while Focused Belief Propagation explores related flatness ideas via a different algorithmic regime (message-passing), no comparable flatness-seeking MCMC algorithm exists specifically for discrete spaces. Our work fills that specific gap. A detailed comparison against fBP is presented in Appendix D. We also formally introduce Local Entropy in Section 3.

Combining flatness and discrete sampling under an MCMC framework is non-trivial: a naive combination of the two would rely on expensive, ad-hoc external checks (e.g., computing Hessians or Local Entropy at every step to check for flatness). This combination will be computationally inefficient( and asymptotic at best) and will break formal MCMC guarantees like detailed balance or convergence to any well-defined stationary distribution in high dimensions. EDLP avoids this by defining a rigorous joint probability distribution on discrete and continuous spaces to gain a richer understanding of the discrete landscape.

**Flatness-aware Optimization.** In early neural network optimization, flatness in energy landscapes emerged as crucial for improving generalization. Hochreiter & Schmidhuber (1994) linked flat minima to better generalization due to their robustness to parameter perturbations. Hochreiter & Schmidhuber (1997) further emphasized the stability advantages of flat regions. Further, LeCun et al. (1990) linked learning algorithm stability to flatness, suggesting optimization methods to exploit this. Later, Gardner (1988) analyzed training algorithms using a statistical mechanics framework, highlighting energy landscape topology's role.

In Bayesian deep learning, Li & Zhang (2024) introduced Entropy MCMC (EMCMC) to bias posterior sampling towards flat regions in continuous spaces, achieving better generalization of Bayesian neural networks. While both EMCMC and EDLP build on the idea of purposing Local Entropy as a mechanism for flat mode sampling in finite time, EMCMC works in a purely continuous space whereas EDLP operates in a hybrid(discrete-continuous) state-space. Because of this difference in design, EMCMC collects both the main and auxiliary continuous variables, whereas EDLP only retains the main discrete variable. Interestingly, since EMCMC's joint SGLD has no acceptance step, EMCMC is thus closest in spirit to EDLP's unadjusted variant i.e. EDULA (formally introduced later in Section 4).

## 3 Preliminaries

**Target Distribution.** We define a target distribution over a discrete space using an energy function. The target distribution is given by $\pi(\boldsymbol{\theta}) = \frac{1}{Z} \exp(U(\boldsymbol{\theta}))$, where $\boldsymbol{\theta}$ is a $d$-dimensional discrete variable within domain $\boldsymbol{\Theta}$, $U(\boldsymbol{\theta})$ represents the energy function, and $Z$ is the normalizing constant ensuring $\pi(\boldsymbol{\theta})$ is a proper probability distribution. We make the following assumptions consistent with the literature on gradient-based discrete sampling (Grathwohl et al., 2021; Sun et al., 2022; Zhang et al., 2022): 1. The domain $\boldsymbol{\Theta}$ is factorized coordinatewisely i.e. $\boldsymbol{\Theta} = \Pi_{i=1}^{d} \boldsymbol{\Theta}_i$, and mainly consider it to be $\{0,1\}^d$ or $\{0, 1, \cdots S\}^d$. 2. Under the Functional Extension framework, discussed in Section 3 of Grathwohl et al. (2021), discrete distributions defined by energy functions sometimes possess a 'natural differentiable extension': $U : \mathbb{R}^d \to \mathbb{R}$ to compute $\nabla_{\boldsymbol{\theta}} U(\boldsymbol{\theta})$ at discrete points. EDLP strictly adheres to this framework: we utilize the gradient of the coupled energy purely to bias the transition probabilities of the categorical proposal, ensuring that $\boldsymbol{\theta}$ remains strictly discrete throughout the process.

**Langevin Algorithm.** In continuous spaces, the Langevin algorithm is a powerful sampling method that follows a Langevin diffusion to update current sample $\boldsymbol{\theta}_a \in \mathbb{R}^d$: $\boldsymbol{\theta}'_{\boldsymbol{a}} = \boldsymbol{\theta}_{\boldsymbol{a}} + \frac{\alpha}{2} \nabla_{\boldsymbol{\theta}_a} U(\boldsymbol{\theta}_{\boldsymbol{a}}) + \sqrt{\alpha}\boldsymbol{\epsilon}$, where $\boldsymbol{\epsilon} \sim \mathcal{N}(\mathbf{0}, \mathbf{I}_{d \times d})$. The gradient assists the sampler in efficiently exploring high-probability regions.

**Discrete Langevin Proposal.** The Discrete Langevin Proposal (DLP) is an extension of the Langevin algorithm tailored for discrete spaces, introduced by Zhang et al. (2022). At a given position $\boldsymbol{\theta}$, the proposal distribution $q(\cdot|\boldsymbol{\theta})$ determines the next position. The proposal distribution in DLP is formulated as:

$$q(\boldsymbol{\theta}'|\boldsymbol{\theta}) = \frac{\exp\left(-\frac{1}{2\alpha}\|\boldsymbol{\theta}' - \boldsymbol{\theta} - \frac{\alpha}{2}\nabla U(\boldsymbol{\theta})\|^2\right)}{Z_{\boldsymbol{\Theta}}(\boldsymbol{\theta})}, \tag{1}$$

where $Z_{\boldsymbol{\Theta}}(\boldsymbol{\theta})$ is the normalizing constant. DLP can be employed without or with a Metropolis-Hastings (MH) step, resulting in the discrete unadjusted Langevin algorithm (DULA) and the discrete Metropolis-adjusted Langevin algorithm (DMALA), respectively.

**Local Entropy.** Local entropy is a critical concept in flatness-aware optimization techniques, which is used to understand the geometric characteristics of energy landscapes. Though originally introduced by Baldassi et al. (2016) as a measure of solution density for discrete systems, local entropy was repurposed by Chaudhari et al. (2019) to characterize geometrically flat minima in continuous optimization landscapes. Here, we utilize this influential conceptual framework and explicitly choose local entropy as our metric for *flatness in discrete spaces*. Specifically, for $\boldsymbol{\Theta}$ as our discrete space equipped with an implicit Hamming metric, the local entropy at any $\boldsymbol{\theta}_a \in \mathbb{R}^d$, with a scalar parameter $\eta$ controlling the sensitivity to flatness in the landscape is defined in equation 2 as,

$$\mathcal{F}(\boldsymbol{\theta}_a; \eta) = \log\left(\sum_{\boldsymbol{\theta} \in \boldsymbol{\Theta}} \exp\left\{U(\boldsymbol{\theta}) - \frac{1}{2\eta}\|\boldsymbol{\theta} - \boldsymbol{\theta}_a\|^2\right\}\right), \tag{2}$$

This definition is directly applicable to problems involving categorical variables, not just ordinal variables via standard embeddings (e.g., one-hot encoding; (Grathwohl et al., 2021)). In this embedding space, a well-defined distance metric between $\boldsymbol{\theta}, \boldsymbol{\theta}_a$ is established. Consequently, our local entropy formulation then operates directly on these vector representations, penalizing discrete states far from the auxiliary variable $\boldsymbol{\theta}_a$.

# 4 Entropic Discrete Langevin Proposal

## 4.1 Target Joint Distribution: Coupling Mechanism

We propose leveraging local entropy (Eq.2) to construct an auxiliary distribution that emphasizes flat regions of the target distribution. This auxiliary distribution smoothens the energy landscape, acting as an external force, driving the exploration of flat basins. Figure 3 in the Appendix B illustrates the motivation behind our approach and the impact of the parameter $\eta$ on the smoothened target distribution.

We start with the original target distribution $p(\boldsymbol{\theta}) \propto \exp(U(\boldsymbol{\theta}))$. By incorporating local entropy, we derive a smoothed version of the target distribution in terms of a new variable $\boldsymbol{\theta}_a$:

$$p(\boldsymbol{\theta}_a) \propto \exp\mathcal{F}(\boldsymbol{\theta}_a; \eta) = \sum_{\boldsymbol{\theta} \in \boldsymbol{\Theta}} \exp\left\{U(\boldsymbol{\theta}) - \frac{1}{2\eta}\|\boldsymbol{\theta} - \boldsymbol{\theta}_a\|^2\right\} \tag{3}$$

Inspired by the coupling method introduced by Li & Zhang (2024) in their Section 4.1, we couple $\boldsymbol{\theta}$ and $\boldsymbol{\theta}_a$ as follows:

**Lemma 4.1.** *Given $\widetilde{\boldsymbol{\theta}} = [\boldsymbol{\theta}^T, \boldsymbol{\theta}_a^T]^T \in \boldsymbol{\Theta} \times \mathbb{R}^d$, the joint distribution $p(\widetilde{\boldsymbol{\theta}})$ is:*

$$p(\widetilde{\boldsymbol{\theta}}) = p(\boldsymbol{\theta}, \boldsymbol{\theta}_a) \propto \exp\left\{U(\boldsymbol{\theta}) - \frac{1}{2\eta}\|\boldsymbol{\theta} - \boldsymbol{\theta}_a\|^2\right\} \tag{4}$$

*By construction, the marginal distributions of $\boldsymbol{\theta}$ and $\boldsymbol{\theta}_a$ are the original distribution $p(\boldsymbol{\theta})$ and the smoothed distribution $p(\boldsymbol{\theta}_a)$ (Eq. 3).*

This result directly follows from Lemma 1 under Section 4.1 in Li & Zhang (2024) [1]. The joint hybrid-variable, $\widetilde{\boldsymbol{\theta}}$ lies in a product space where first d coordinates are discrete-valued and the remaining d coordinates lie in

---

[1]In EMCMC, $\boldsymbol{\theta}$(main) and $\boldsymbol{\theta}_a$(auxiliary) are continuous vectors operating in Euclidean space. In our framework, the main variable $\boldsymbol{\theta}$ is discrete, while the auxiliary variable $\boldsymbol{\theta}_a$ remains continuous. Thus, our notation reflects not just the functional role of the variables (main vs. auxiliary), but also the distinct topological spaces they inhabit.

$\mathbb{R}^d$. Consequently, the energy function of $\widetilde{\boldsymbol{\theta}}$ becomes,

$$U_\eta(\widetilde{\boldsymbol{\theta}}) = U(\boldsymbol{\theta}) - \frac{1}{2\eta}\|\boldsymbol{\theta} - \boldsymbol{\theta}_a\|^2$$

, and its gradient is given by:

$$\nabla_{\widetilde{\boldsymbol{\theta}}} U_\eta(\widetilde{\boldsymbol{\theta}}) = \left[ \begin{array}{c} \nabla_{\boldsymbol{\theta}} U_\eta(\widetilde{\boldsymbol{\theta}}) \\ \nabla_{\boldsymbol{\theta}_a} U_\eta(\widetilde{\boldsymbol{\theta}}) \end{array} \right] = \left[ \begin{array}{c} \nabla_{\boldsymbol{\theta}} U(\boldsymbol{\theta}) - \frac{1}{\eta}(\boldsymbol{\theta} - \boldsymbol{\theta}_a) \\ \frac{1}{\eta}(\boldsymbol{\theta} - \boldsymbol{\theta}_a) \end{array} \right]. \tag{5}$$

It is worth mentioning, given our primary objective of designing a MCMC method that can cater to high dimensional variables, we avoid the bias introduced by relaxation methods (like Gumbel-Softmax (Jang et al., 2017), Concrete distributions (Maddison et al., 2017)). Instead, our differentiable extension of the energy function maintains strict discreteness of the state space while allowing valid gradient guidance.

### 4.2 Sampling Algorithm: Local Entropy Guidance in Discrete Langevin Proposals

We propose EDLP, an extension of DLP designed to enhance sampling efficiency from flat modes. In our framework (Algorithm 1), the Langevin update for $\boldsymbol{\theta}_a$ follows the distribution $q_{\alpha_a}(\boldsymbol{\theta}'_a|\widetilde{\boldsymbol{\theta}})$:

$$q_{\alpha_a}(\boldsymbol{\theta}'_a|\widetilde{\boldsymbol{\theta}}) = \frac{1}{\sqrt{2\pi\alpha_a}^d} \exp\left( -\frac{1}{2\alpha_a}\|\boldsymbol{\theta}'_a - \boldsymbol{\theta}_a - \frac{\alpha_a}{2}\nabla_{\boldsymbol{\theta}_a} U_\eta(\widetilde{\boldsymbol{\theta}})\|^2 \right). \tag{6}$$

Unlike the standard DLP, where transitions are purely between discrete states, EDLP leverages the current joint variables $\widetilde{\boldsymbol{\theta}} = [\boldsymbol{\theta}^T, \boldsymbol{\theta}_a^T]^T$ to propose the next discrete state. By incorporating the coupling between the variables, we refine the DLP proposal by replacing $\nabla U(\boldsymbol{\theta})$ with $\nabla_{\boldsymbol{\theta}} U_\eta(\widetilde{\boldsymbol{\theta}})$. This adjustment results in the modified proposal:

$$q_\alpha(\boldsymbol{\theta}'|\widetilde{\boldsymbol{\theta}}) \propto \exp\left( -\frac{1}{2\alpha}\|\boldsymbol{\theta}' - \boldsymbol{\theta} - \frac{\alpha}{2}\nabla_{\boldsymbol{\theta}} U_\eta(\widetilde{\boldsymbol{\theta}})\|^2 \right). \tag{7}$$

To further simplify, we use coordinate-wise factorization from DLP to obtain $q_\alpha(\boldsymbol{\theta}'|\widetilde{\boldsymbol{\theta}}) = \prod_{i=1}^d q_{\alpha_i}(\theta'_i|\widetilde{\boldsymbol{\theta}})$, where $q_{\alpha_i}(\theta'_i|\widetilde{\boldsymbol{\theta}})$ is a categorical distribution:

$$\text{Cat}\left( \text{Softmax}\left( \frac{1}{2}\nabla_{\theta} U_\eta(\widetilde{\boldsymbol{\theta}})_i(\theta'_i - \theta_i) - \frac{(\theta'_i - \theta_i)^2}{2\alpha} \right) \right). \tag{8}$$

By synthesizing Equations equation 6 and equation 8, we derive the full proposal distribution:

$$q_\gamma(\widetilde{\boldsymbol{\theta}'}|\widetilde{\boldsymbol{\theta}}) \propto q_\alpha(\boldsymbol{\theta}'|\widetilde{\boldsymbol{\theta}})q_{\alpha_a}(\boldsymbol{\theta}'_a|\widetilde{\boldsymbol{\theta}}) \tag{9}$$

where $\gamma = (\alpha, \alpha_a)$.

This factorized proposal in Eq. equation 9 is purely a design choice to simplify sampling. The proposal distribution is called the *Entropic Discrete Langevin Proposal* (EDLP). At the current joint position $\widetilde{\boldsymbol{\theta}}$, EDLP generates the next joint position $\widetilde{\boldsymbol{\theta}'}$. EDLP can be paired with or without a Metropolis-Hastings step (Metropolis et al., 1953; Hastings, 1970) to ensure the Markov chain's reversibility. These algorithms are referred to as EDULA (Entropic Discrete Unadjusted Langevin Algorithm) and EDMALA (Entropic Discrete Metropolis-Adjusted Langevin Algorithm), respectively. We only collect $\boldsymbol{\theta}$ samples to target flat discrete basins in finite time. We emphasize sampling from the joint-distribution introduces bias that is a transient feature of the coupling that prioritizes high-volume modes during finite-time exploration; in the limit, the chain remains ergodic with respect to $\pi(\boldsymbol{\theta})$ (Lemma 4.1).

Unlike EMCMC, where coupling between continuous vectors in Euclidean space is trivial, EDLP must reconcile the topological mismatch between the discrete target $\boldsymbol{\theta}$ and the continuous auxiliary $\boldsymbol{\theta}_a$ for meaningful structural guidance. This extension is challenging: it requires handling diverse discrete data types (e.g., categorical vs. ordinal). For categorical variables, "distance" to a continuous vector is undefined without a continuous embedding space. EDLP's coupling effortlessly extends in the embedding dimension, ensuring

---
**Algorithm 1** Entropic Discrete Langevin Proposal: EDULA and EDMALA
---

**Inputs:** Main variable $\boldsymbol{\theta} \in \boldsymbol{\Theta}$ , Auxiliary variable $\boldsymbol{\theta}_a \in \mathbb{R}^d$, Main stepsize $\alpha$, Auxiliary stepsize $\alpha_a$, Flatness parameter $\eta$

**Initialize:** $\boldsymbol{\theta}_a \leftarrow \boldsymbol{\theta}, \mathcal{S} \leftarrow \emptyset$

**loop**
    **Construct** $\nabla_{\widetilde{\boldsymbol{\theta}}} U_\eta(\widetilde{\boldsymbol{\theta}})$ as in Equation equation 5
    **for** $i = 1$ **to** $d$ **do**
        **Construct** $q_{i_\alpha}(\cdot | \widetilde{\boldsymbol{\theta}})$ as in Equation equation 8
        **Sample** $\boldsymbol{\theta}_i{}' \sim q_{i_\alpha}(\cdot | \widetilde{\boldsymbol{\theta}})$
    **end for**
    **Compute** $\boldsymbol{\theta}'_{\boldsymbol{a}} \leftarrow \boldsymbol{\theta}_a + \frac{\alpha_a}{2} \nabla_{\boldsymbol{\theta}_a} U_\eta(\widetilde{\boldsymbol{\theta}}) + \sqrt{\alpha_a} \boldsymbol{\epsilon}$     where $\boldsymbol{\epsilon} \sim \mathcal{N}(\boldsymbol{0}, \boldsymbol{I})$

    ▷ Optionally, do the MH step for EDMALA
    **Compute** $q_\alpha(\widetilde{\boldsymbol{\theta}'} | \widetilde{\boldsymbol{\theta}}) = \prod_i q_{i_\alpha}(\widetilde{\boldsymbol{\theta}'_i} | \widetilde{\boldsymbol{\theta}})$
            and $q_\alpha(\widetilde{\boldsymbol{\theta}} | \widetilde{\boldsymbol{\theta}'}) = \prod_i q_{i_\alpha}(\widetilde{\boldsymbol{\theta}_i} | \widetilde{\boldsymbol{\theta}'})$
    **Set** $\boldsymbol{\theta} \leftarrow \boldsymbol{\theta}'$ and $\boldsymbol{\theta}_a \leftarrow \boldsymbol{\theta}'_{\boldsymbol{a}}$ with probability

$$\min \left( 1, \frac{q_\alpha(\boldsymbol{\theta} | \widetilde{\boldsymbol{\theta}'})}{q_\alpha(\boldsymbol{\theta}' | \widetilde{\boldsymbol{\theta}})} \frac{q_{\alpha_a}(\boldsymbol{\theta}_{\boldsymbol{a}} | \widetilde{\boldsymbol{\theta}'})}{q_{\alpha_a}(\boldsymbol{\theta}'_{\boldsymbol{a}} | \widetilde{\boldsymbol{\theta}})} \frac{\pi(\widetilde{\boldsymbol{\theta}'})}{\pi(\widetilde{\boldsymbol{\theta}})} \right).$$

    **if** *after burn-in* **then**
        **Update** $\mathcal{S} \leftarrow \mathcal{S} \cup \{\boldsymbol{\theta}\}$
    **end if**
**end loop**
**Output:** $\mathcal{S}$

---

the gradient signal $\nabla \| \text{Embed}(\boldsymbol{\theta}) - \boldsymbol{\theta}_a \|^2$ is well defined. These implementation choices require careful thought, challenges absent in pure continuous sampling frameworks.

**Why does EDLP work?** One notices, in flat regions the sum in Equation 2 is high due to many contributing terms, giving $\boldsymbol{\theta}_a$ *large freedom to move*. Conversely, in sharp regions, $\boldsymbol{\theta}_a$ is strongly incentivized to *stay put* as the Gaussian term rapidly decays to zero, leaving no other low-energy states nearby. Within a flat basin, $\boldsymbol{\theta}_a$ has more freedom to move due to many favorable neighboring states, while in a sharp, isolated peak, it is constrained. This builds the intuition behind our approach: EDLP induces local entropy behavior through finite-time dynamics, guiding the sampler to regions offering the greatest configurational volume with flatter basins.

Alongside the vanilla EDLP, we introduce a computationally efficient *Gibbs-like-update* (GLU) version, in the Appendix C, which involves alternating updates instead of simultaneous updates of our variables. Sensitivity analysis of the hyperparameters and runtime analysis are presented in Appendix, in sections B and F.4 respectively.

## 5 Theoretical Analysis

In this section, we provide a theoretical analysis of the convergence rates for EDULA and EDMALA. We adopt the standard assumptions[2] used in analyzing flatness-aware MCMC algorithms (Li & Zhang, 2024), which are also consistent with the broader sampling literature (Pynadath et al., 2024; Bottou et al., 2018; Dalalyan, 2017; Durmus & Moulines, 2017b).

**Assumption 5.1** (Smoothness). *The function $U(\cdot) \in C^2(\mathbb{R}^d)$ has $M$-Lipschitz gradient.*

**Assumption 5.2** (Strong Concavity). *For each $\boldsymbol{\theta} \in \mathbb{R}^d$, there exists an open ball containing $\boldsymbol{\theta}$ of some radius $r_{\boldsymbol{\theta}}$, denoted by $B(\boldsymbol{\theta}, r_{\boldsymbol{\theta}})$, such that the function $U(\cdot)$ is $m_{\boldsymbol{\theta}}$-strongly concave in $B(\boldsymbol{\theta}, r_{\boldsymbol{\theta}})$ for some $m_{\boldsymbol{\theta}} > 0$.*

---
[2]Our assumptions are purely for the ease of theoretical analysis. Our experimental setups in Section 6 extend beyond this.

**Assumption 5.3** (Bounded Auxiliary Space). *$\boldsymbol{\theta}_a$ is restricted to a compact subset of $\mathbb{R}^d$ labeled $\boldsymbol{\Theta}_a$.*

We define $\delta(\boldsymbol{\Theta}) = \inf_{\boldsymbol{\theta},\boldsymbol{\theta}'\in\boldsymbol{\Theta}} \|\boldsymbol{\theta}-\boldsymbol{\theta}'\|$, $\mathrm{diam}(\boldsymbol{\Theta}) = \sup_{\boldsymbol{\theta},\boldsymbol{\theta}'\in\boldsymbol{\Theta}} \|\boldsymbol{\theta}-\boldsymbol{\theta}'\|$, and $\mathrm{diam}(\boldsymbol{\Theta_a}) = \sup_{\boldsymbol{\theta}_a,\boldsymbol{\theta}'_a\in\boldsymbol{\Theta}_a} \|\boldsymbol{\theta}_a-\boldsymbol{\theta}'_a\|$. Let $\vartheta(\boldsymbol{\Theta},\boldsymbol{\Theta_a}) = \inf_{\boldsymbol{\theta},\boldsymbol{\theta}'\in\boldsymbol{\Theta};\boldsymbol{\theta}_a,\boldsymbol{\theta}'_a\in\boldsymbol{\Theta}_a} (\boldsymbol{\theta}-\boldsymbol{\theta}_a)^\top(\boldsymbol{\theta}'-\boldsymbol{\theta}'_a)$ and $\Delta(\boldsymbol{\Theta},\boldsymbol{\Theta}_a) = \sup_{\boldsymbol{\theta}\in\boldsymbol{\Theta},\,\boldsymbol{\theta}_a\in\boldsymbol{\Theta}_a} \|\boldsymbol{\theta_a}-\boldsymbol{\theta}\|$. Let the joint valid bounded space be $\widetilde{\boldsymbol{\Theta}}$ and finally define $a \in \arg\min_{\boldsymbol{\theta}\in\boldsymbol{\Theta}} \|\nabla U(\boldsymbol{\theta})\|$ as the set of values which minimizes the energy function in $\boldsymbol{\Theta}$. $\delta(\boldsymbol{\Theta}) = 1$ is trivially true for most settings, even the ones discussed in Section 6.

Assumption 5.1 ensures the energy landscape does not fluctuate arbitrarily fast between any two discrete states. Under Assumption 5.2, $U(\cdot)$ is $m$-strongly concave on $\mathrm{conv}(\boldsymbol{\Theta})$. In our context, this still allows for flat modes—characterized by wide basins with small $m$ relative to sharp modes. Assumption 5.3 prevents the continuous auxiliary variable from wandering to infinity, ensuring the entropic coupling between $\boldsymbol{\theta}$ and $\boldsymbol{\theta}_a$ remains active and meaningful. The total variation distance between two probability measures $\mu$ and $\nu$, defined on some space $\boldsymbol{\theta} \subset \mathbb{R}^d$ is $\|\mu-\nu\|_{TV} = \sup_{A\subseteq B(\boldsymbol{\theta})} |\mu(A)-\nu(A)|$ where $B(\boldsymbol{\theta})$ is the set of all measurable sets in $\boldsymbol{\theta}$.

## 5.1 Convergence Analysis for EDULA

Since EDULA does not have the target as the stationary distribution, we establish mixing bounds for it in two steps. We first prove that when both the stepsizes ($\alpha$, $\alpha_a$) tend to zero, the asymptotic bias of EDULA is zero for target distribution $\tilde{\pi} \propto e^{(U(\boldsymbol{\theta})-\frac{1}{2\eta}\|\boldsymbol{\theta}-\boldsymbol{\theta}_a\|^2)}$[3]

**Proposition 5.4.** *Under Assumptions 5.1 and 5.3, the Markov chain as defined in equation 9 is reversible with respect to some distribution $\pi_\gamma$ and $\pi_\gamma$ converges weakly to $\pi$ as $\alpha \to 0$ and $\alpha_a \to 0$. Further, for any $\frac{2}{M} > \alpha > 0, \alpha_a > 0$,*

$$\|\pi_\gamma - \tilde{\pi}\|_1 \le |\boldsymbol{\Theta}| \cdot \exp\left( \frac{M\,diam(\boldsymbol{\Theta})}{2} + (\frac{M}{4} - \frac{1}{2\alpha})\delta(\boldsymbol{\Theta})^2 + \frac{1}{2\eta}\|\boldsymbol{\theta}-\boldsymbol{\theta}_a\|^2 - \frac{\vartheta(\boldsymbol{\Theta},\boldsymbol{\Theta}_a)}{2\eta} \right),$$

*Sketch of Proof of Proposition 5.4.* Since our chain is unadjusted (no MH step), it does not leave $\tilde{\pi}$ invariant. Instead, we identify a "ghost" distribution $\pi_\gamma \propto \tilde{\pi} \cdot e^{\frac{\alpha_a}{8\eta^2}\|\boldsymbol{\theta}-\boldsymbol{\theta}_a\|^2}$ with respect to which $q_\gamma$ satisfies detailed balance. We then show that the correction factor$(Z'_\gamma(\tilde{\boldsymbol{\theta}}))$ becomes unity as step sizes $\alpha, \alpha_a \to 0$. This implies pointwise convergence: $\pi_\gamma \to \tilde{\pi}$. Using Scheffé's Lemma, pointwise convergence implies convergence in $L_1$ norm. Hence, for finite step sizes we bound the distance $\|\pi_\gamma - \tilde{\pi}\|_1$ by expanding the transition kernel and applying the Lipschitz smoothness of $\nabla U$ (Assumption 5.1). $\qquad\square$

The parameter $\alpha_a$ is consumed during the computation of the stationary distribution $\pi_\gamma$, explicitly not appearing in the bound. However, $\alpha_a$ indirectly influences the geometric terms $\Delta(\boldsymbol{\Theta},\boldsymbol{\Theta}_a)$ and $\vartheta(\boldsymbol{\Theta},\boldsymbol{\Theta}_a)$. Larger $\alpha_a$ increases $\Delta^2(\boldsymbol{\Theta},\boldsymbol{\Theta}_a)$ due to a greater diameter and reduces $\vartheta(\boldsymbol{\Theta},\boldsymbol{\Theta}_a)$ due to weaker alignment, thereby loosening the bound. In contrast, smaller $\alpha_a$ tightens convergence guarantees. This parallels the observable role of $\alpha$ in the bound, i.e. bias vanishes to 0 as $\alpha \to 0$. Next we establish our main result for EDULA which establishes a bound on how far our Markov chain is from the target given the step-sizes. Consider $\bar{\eta}^*$ to be a constant that can be explicitly computed (see equation 20 in the Appendix).

**Theorem 5.5.** *Under Assumptions 5.1, and 5.3 in Algorithm 1, Markov chain P exhibits,*

$$\|P^k(x,\cdot) - \tilde{\pi}\|_{TV} \le (1-\bar{\eta}^*)^k + |\boldsymbol{\Theta}| \cdot \exp\left( \frac{M\,diam(\boldsymbol{\Theta})}{2} + (\frac{M}{4} - \frac{1}{2\alpha})\delta(\boldsymbol{\Theta})^2 + \frac{1}{2\eta}\|\boldsymbol{\theta}-\boldsymbol{\theta}_a\|^2 - \frac{\vartheta(\boldsymbol{\Theta},\boldsymbol{\Theta}_a)}{2\eta} \right)$$

*Sketch of Proof of Theorem 5.5.* The key idea of the proof leverages Proposition 5.4 and the ergodicity of the EDULA chain, as a consequence of Lemma E.6. Lemma E.6 states that the transition kernel for the Markov chain as defined by Algorithm 1 without the accept/reject step satisfies the Doeblin's condition Jones (2004) uniformly over the entire space $\boldsymbol{\Theta} \times \boldsymbol{\Theta}_a$. This allows us to determine that the chain is uniformly ergodic with the rate of convergence completely determined by the minorization coefficient.

---

[3]Previously we used $p(\widetilde{\boldsymbol{\theta}})$ as generic notation to introduce the concept of auxiliary coupling. Here, we adopt $\tilde{\pi}$ to specifically denote the joint target equilibrium distributions defined by the energy function. This follows the notational convention established in EMCMC.

The proof for Lemma E.6, follows by lower bounding the transition kernel to obtain the key inequality

$$P(\boldsymbol{\theta}, \cdot) \geq \bar{\eta}^* \text{Unif}(\boldsymbol{\Theta}_a \times \boldsymbol{\Theta})(\cdot)$$

which is a result of examining the behavior of the potential function $U(\cdot)$ around the minimum point in the domain of interest. This allows us to establish the aforementioned rate as $1 - \eta^*$.

However, the convergence as established via lemma E.6 is not to the target but the stationary distribution of the Markov chain which differs from the target. This difference is captured and bounded in Proposition 5.4. Combining, we get the result. □

As mentioned in the Appendix, $\bar{\eta}^* = f(\alpha, \alpha_a, \text{diam}(\boldsymbol{\Theta}), \text{diam}(\boldsymbol{\Theta}_a), \Delta(\boldsymbol{\Theta}_a, \boldsymbol{\Theta}))$, where $f$ is increasing exponentially in the first two arguments and decreasing exponentially in the last three arguments. Theorem 5.5 shows that sufficiently small learning rates bring the samples generated by Algorithm 1 closer to the target distribution. However, excessively small rates hinder convergence by limiting exploration, while large rates cause the sampler to overshoot the target. Thus, choosing an appropriate learning rate is critical for balancing exploration and convergence.

## 5.2 Convergence Analysis for EDMALA

We establish a non-asymptotic convergence guarantee for EDMALA using a uniform minorization argument.

**Theorem 5.6.** *Under Assumptions 5.1 ,5.2, and 5.3 , and $\alpha < \frac{2}{M}$ in Algorithm 1, Markov chain $P$ is uniformly ergodic with,*

$$\|P^k(x, \cdot) - \tilde{\pi}\|_{TV} \leq (1 - \epsilon_\gamma)^k$$

*where,* $\epsilon_\gamma = \exp\left\{ -\left(\frac{M}{2} + \frac{1}{\alpha} - \frac{m}{4}\right) diam(\boldsymbol{\Theta})^2 - \frac{1}{2}\|\nabla U(a)\| \, diam(\boldsymbol{\Theta}) - \left(\frac{3\alpha_a}{8\eta^2} + \frac{2}{\eta}\right)\Delta(\boldsymbol{\Theta}, \boldsymbol{\Theta}_a)^2 + \frac{\vartheta(\boldsymbol{\Theta}, \boldsymbol{\Theta}_a)}{\eta} \right\}$

*Sketch of Proof of Theorem 5.6.* We establish the convergence rate of the EDMALA chain by proving a uniform minorization condition for its transition kernel $P$. First, using the Lipschitz smoothness of $\nabla U$ (Assumption 5.1) and strong convexity (Assumption 5.2), we first derive a uniform lower bound on the proposal probability $q_\gamma(\widetilde{\boldsymbol{\theta}}'|\widetilde{\boldsymbol{\theta}})$ that holds for any pair of states. Next, we analyze the Metropolis-Hastings acceptance ratio $\rho(\widetilde{\boldsymbol{\theta}}'|\widetilde{\boldsymbol{\theta}})$ and show that the acceptance probability is uniformly bounded away from zero. Finally, we combine the bounds on the proposal and acceptance ratio, and show that the transition kernel satisfies $P(\widetilde{\boldsymbol{\theta}}'|\widetilde{\boldsymbol{\theta}}) \geq \epsilon\nu(\widetilde{\boldsymbol{\theta}}')$ for a probability measure $\nu$ and constant $\epsilon > 0$. This uniform Doeblin condition implies that the chain is uniformly ergodic, converging to its stationary distribution $\tilde{\pi}$ at a geometric rate $(1 - \epsilon)^k$. □

One notices, $\epsilon_\gamma$ is exponentially decreasing in the size of the set, $\boldsymbol{\Theta}$, its distance from $\boldsymbol{\Theta}_a$. Further, as $\alpha \to 0$, $\epsilon_\gamma \to 0$, causing the convergence factor $1 - \epsilon_\gamma$ to approach 1. This slows the convergence rate, as the chain takes longer to approach the stationary distribution. Further, for $\eta \to \infty$ (weaker coupling), the bounds in Proposition 5.4 and Theorem 5.6 align with those of DULA (Zhang et al., 2022) and DMALA (Pynadath et al., 2024), respectively. Note that the convergence of the chains for both EDULA and EDMALA imply convergence of the marginals as the projection maps are continuous. In fact, deriving a rate of convergence for them is also possible, but we omit it here as that is not the goal of this paper. We focus our analysis on the joint distribution to better capture the flatness-seeking behavior induced by the auxiliary variable( as in Theorem 1 in Li & Zhang (2024)) in finite time.

**Impact of $\eta$ on finite-time flatness preference and asymptotic target fidelity.** In EDLP, the auxiliary variable $\boldsymbol{\theta}_a$ and its associated coupling parameter $\eta$ serve as a bias mechanism toward flatter regions under limited computational budget. For judicious chosen values, $\eta$ provides a meaningful bias toward flat modes by guiding discrete samples to regions where multiple states with similar energies exist nearby. In these flat regions, gradient signals are weaker, and standard samplers (like DLP) typically have difficulty exploring efficiently. Thus, moderate $\eta$ encourages sampling from flat regions at the cost of slightly slower finite-time convergence. As $\eta$ grows large, the coupling weakens and the sampler's behavior naturally approaches that of standard DLP (Figure 3 in Appendix). This recovers unbiased sampling that targets high density regions as

now the sampler solely utilizes $U(.)$'s gradient. Hence, large $\eta$ values place minimal bias toward flatness but provide faster finite-time convergence to high-density regions.

Quantitatively, the balance can be seen explicitly in our non-asymptotic convergence guarantees (Theorems 5.5 and 5.6). Comparing our EDMALA bounds with standard DMALA bounds (Pynadath et al., 2024), one can observe that introducing any $\eta > 0$ does introduce an additional *drag* in the bound, slightly impeding finite-time convergence. However, it is crucial to emphasize that this effect is explicitly finite-time. Given sufficient iterations (i.e. $k \to \infty$), our method, like all valid MCMC algorithms, is guaranteed to converge exactly to the true joint distribution $(\tilde{\pi})($ and hence, $\pi(\boldsymbol{\theta})$ ensuring long-run correctness). In practice however, targeting beneficial sites in discrete landscape often occurs within a limited computational budget. Thus for EDLP its primary utility lies in finite-time exploration. This also implies for a fixed smaller(larger) $\eta$, more(less) compute is required to reach equilibrium.

Additionally, a stronger coupling (small $\eta$) constrains the movement of the variables, decreasing the joint MH acceptance probability in EDMALA. Conversely, relaxing the coupling (large $\eta$) allows for faster mixing and higher acceptance rates, but at the loss of flatness awareness (Figure 4 in Appendix).

For practical tuning, we recommend a coarse grid search as shown in our Sensitivity Analysis in Appendix B. In practice, $\eta$ can be effectively tuned by monitoring the $L_2$ distance $\|\boldsymbol{\theta} - \boldsymbol{\theta}_a\|$; an intermediate distance that remains stable during sampling indicates structural guidance. Future directions may include automating this selection by repurposing preconditioning matrices from gradient-based optimizers to estimate landscape curvature directly (Li et al., 2015; Girolami & Calderhead, 2011).

**Theorem 5.7.** *Under assumptions 5.1, 5.2, 5.3, and $\alpha < \frac{2}{M}$ in Algorithm 1 . Then, for any function $f : \mathbb{R}^p \to \mathbb{R}$ with $\|f\|_{\mathbb{L}^2_\pi} < \infty$, one has*

$$\sqrt{n} \left( \bar{f} - \mathbb{E}_{\pi_\gamma} f \right) \xrightarrow{d} N(0, \sigma_f^2)$$

*as $n \to \infty$, where $\sigma_f^2 \in [0, \infty)$.*

*Proof.* Theorem 5.7 is true due to direct consequence of using Theorems 5.5, 5.6 and Jones (2004)[Corollary 5]. □

**Comparison with Proximal Samplers.** While current proximal sampling (Chen et al., 2022) focuses on mixing rates in continuous spaces, when framed as a Proximal Sampler on a restricted discrete domain, EDLP is shown to have geometric ergodicity for the coupled chain (Theorems 5.5, 5.6). This provides necessary condition for robust inferential guarantees, such as the existence of CLT for the Markov chain(Theorem 5.7). This distinguishes our contribution from standard proximal sampling analyses, offering a novel treatment of proximal dynamics in discrete landscapes. Finally, for a joint hybrid-chain (where one variable jumps and the other diffuses) providing non-asymptotic guarantees adds an addition layer of complexity beyond pure continuous diffusion as seen in Raginsky et al. (2017); Durmus et al. (2019); Durmus & Moulines (2017a). Unlike standard Langevin analyses which operate on a single homogeneous domain, our hybrid framework introduces a novel source of bias arising from the geometric interaction between the discrete target and the continuous auxiliary space: our bounds reveal how the geometric alignment terms govern the trade-off between exploring flat regions and maintaining fidelity to the target distribution.

## 6 Experiments

We conducted an empirical evaluation of the Entropic Discrete Langevin Proposal (EDLP) to demonstrate its effectiveness in sampling from flat regions compared to existing discrete samplers. Our experimental setups mainly follow Zhang et al. (2022). EDLP is benchmarked against a range of popular baselines, including Gibbs sampling, Gibbs with Gradient (GWG) (Grathwohl et al., 2021), Hamming Ball (HB) (Titsias & Yau, 2017), Discrete Unadjusted Langevin Algorithm (DULA), and Discrete Metropolis-Adjusted Langevin Algorithm (DMALA) (Zhang et al., 2022). We retain Zhang et al. (2022)'s notation for consistency: Gibbs-*X* for Gibbs sampling, GWG-*X* for Gibbs with Gradient, and HB-*X-Y* for Hamming Ball. To the best of our knowledge, fBP (Baldassi et al., 2016) is the only algorithm that targets flat regions in discrete spaces.

However, it is not directly comparable to EDLP and the other samplers in our study due to methodological and practical reasons (see Appendix D for details).

In previous works on flatness-awareness sampling and optimization, the goal has exclusively centered around better generalization or bayesian inference in deep learning. In this section, we adopt a nuanced and bespoke multifaceted approach showcasing broader utility of flatness in discrete domains under : 1. Flatness ensures structural robustness against perturbations, crucial for real-world constraints. 2. Flat modes represent data-centric learning prototypes, capturing the core structure of the data manifold in generative modeling. 3. We confirm the classic generalization benefit.

## 6.1 Motivational Synthetic Example

We consider sampling from a joint quadrivariate Bernoulli distribution. Let $\boldsymbol{\theta} = (\theta_1, \theta_2, \theta_3, \theta_4)$ be a 4-dimensional binary random vector, where each $\theta_i \in \{0, 1\}$. The joint probability distribution is specified by $p_{\boldsymbol{\theta}}$, which represents the probability of the vector $(\theta_1, \theta_2, \theta_3, \theta_4)$.

For a given state $\boldsymbol{\theta}$, energy function is given by :

$$U(\boldsymbol{\theta}) = \sum_{a \in \{0,1\}^4} \left( \prod_{n=1}^{4} \theta_n^{a_n} (1 - \theta_n)^{1-a_n} \right) \ln p_a,$$

The target distribution over the 4D Joint Bernoulli space contains both sharp and flat modes, each analyzed over their 1-Hamming distance neighborhoods. Sharp modes, such as 0010 and 0111, have high probability mass but are surrounded by neighbors with significantly lower probabilities, indicating steep local gradients. In contrast, flat modes like 0100 and

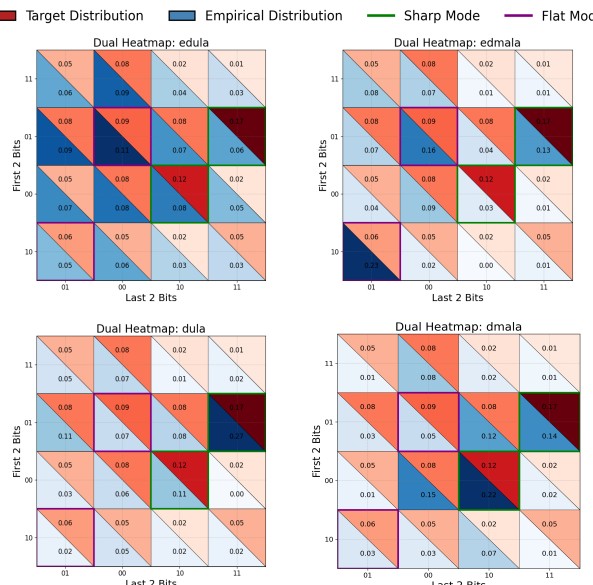

Figure 2: Overlap heatmaps demonstrating EDLP's flatness preference over DLP.

1001 are characterized by relatively uniform probabilities among their immediate neighbors, reflecting smoother local geometry. For the true target distribution's visualization refer to Figure 10 in Appendix F.1.

To gauge EDLP's impact in finite time, we ran chains of DULA, EDULA, DMALA, and EDMALA in parallel for 1000 iterations. From Figure 2, EDMALA and EDULA demonstrate a strong preference to visit flat modes, without becoming stuck in the high-probability sharp modes. In contrast, DULA and DMALA show a bias toward the sharp modes, showing to be less adept at exploring the flat areas where the probability mass is more evenly distributed. The Total Variational Distance(TVD) and KL-Divergence(KL) for for DULA (0.1546, 0.0933), DMALA (0.2683,0.2395), EDULA (0.1893, 0.1244), EDMALA (0.2879, 0.3094) indicate poorer fidelity to $\pi(\boldsymbol{\theta})$ for the latter two in finite time.

We report frequency counts, provide extended results and more experiments on EDLP's finite-time mode-seeking behavior in Appendix section F.1.

## 6.2 Sampling for Traveling Salesman Problems

In TSP, the objective is to find the shortest route visiting $n$ cities exactly once and returning to the origin, choosing from $n!$ paths. However, in practical logistics and routing, structural stability and cost similarity are sometimes of greater interest than raw optimality. A sharp global minimum is brittle; if a single edge becomes infeasible (e.g., due to road closures or driving preference), re-optimization might yield a completely disjoint route , disrupting downstream logistics (Laporte, 2009; Golden et al., 2008). Therefore, in practice, the cost function is often an incomplete proxy for complex, unmodeled constraints—such as driver familiarity, road safety, or dynamic traffic patterns. Consequently, our goal here shifts from pure minimization to reliable candidate generation.

We posit that flat modes in the solution landscape naturally model robust solution basins. A flat mode corresponds to a cluster of similar-cost tours that share a high degree of structural similarity (i.e., they share many common edges). Practitioners may perform a post-hoc check to verify if the identified cluster falls within their budgetary constraints (Pugh et al., 2016; Marler & Arora, 2004). This selection eventually creates a safety buffer, allowing practitioners to choose from options based on unmodeled constraints without fear that a minor structural deviation will cause a catastrophic violation of the budget (Groër et al., 2009). Thus, sampling from flat basins allows us to implicitly impose the similarity constraint, proposing a "menu of high-quality options".

In standard TSP, $U(\boldsymbol{\theta})$ is designed to capture the total cost of a particular route configuration $\boldsymbol{\theta} = (\theta_1, \theta_2, \ldots, \theta_n)$. The expression of $U(\boldsymbol{\theta})$ is:

$$U(\boldsymbol{\theta}) = -\left( \sum_{i=1}^{n-1} \sqrt{(x_i - x_{i+1})^2 + (y_i - y_{i+1})^2} + \sqrt{(x_n - x_1)^2 + (y_n - y_1)^2} \right),$$

where each city location is given by $\theta_i = (x_i, y_i) \in \mathbb{R}^2$. The final term, $\sqrt{(x_n - x_1)^2 + (y_n - y_1)^2}$, ensures that the route forms a closed loop, thereby completing the tour. We use the *eil30* dataset, a 30-city problem extracted from the 51-city Christofides/Eilon instance, available in the TSPLIB95 benchmark repository, first introduced in Reinelt (1991). The dataset consists of 2D Euclidean coordinates of cities and is publicly available.[4] Each solution is represented as a binary square matrix of size $n$, where each row corresponds to a position in the tour and each column corresponds to a city, with $|\boldsymbol{\Theta}| = 2.6525 \times 10^{32}$. If a proposed solution violates the uniqueness of city visits, we reject the sample and retain the current solution.

Table 1: EDLP finding cost-wise and structure-wise similar solutions than DLP.

| Sampler | PMC ($\downarrow$) | Jaccard Similarity ($\uparrow$) | Range of Cost ($\downarrow$) | Average Time per Sample(s)($\downarrow$) |
|---|---|---|---|---|
| DULA | $413.07 \pm 94.92$ | $0.0917 \pm 0.1546$ | 336.57 | $\mathbf{0.0562 \pm 0.0032}$ |
| DMALA | $249.65 \pm 104.61$ | $0.3341 \pm 0.2060$ | 198.12 | $0.0586 \pm 0.0081$ |
| EDULA | $349.92 \pm 114.71$ | $0.1704 \pm 0.1993$ | 194.75 | $0.0628 \pm 0.0202$ |
| EDMALA | $\mathbf{178.93 \pm 81.03}$ | $\mathbf{0.4818 \pm 0.1790}$ | $\mathbf{62.51}$ | $0.0637 \pm 0.0265$ |

We report the PMC (Pairwise Mismatch Count) inspired by Pairwise Order Discrepancy (Zaefferer et al., 2014; Liao et al., 2012) and Kendall's Tau (Fagin et al., 2003), and Jaccard similarity (Li et al., 2022), Range of Cost( difference between maximum and minimum of negative of $U(\boldsymbol{\theta})$), Time taken per sample(s) after collecting solutions for each sampler. From Table 1, by guiding the sampling trajectory towards flat modes, EDLP yields a set of routes that are not only cost-wise similar but also robust to local perturbations, a property that pure cost-minimization fails to capture.

We offer additional insights in Appendix F.2.

### 6.3 Sampling From Restricted Boltzmann Machines

Restricted Boltzmann Machines (RBMs) are a class of generative stochastic neural networks that learn a probability distribution over their input data. The energy function for an RBM, which defines the joint configuration of visible and hidden units, is given by:

$$U(\boldsymbol{\theta}) = \sum_i \text{Softplus}(\mathbf{W}\boldsymbol{\theta} + a)_i + b^\top \boldsymbol{\theta},$$

where $\{\mathbf{W}, a, b\}$ are the weight matrix and bias parameters, respectively, and $\boldsymbol{\theta} \in \{0, 1\}^d$ represents the binary state of the visible units.

When the RBM assigns high probability to specific digit representations, a sharp mode for digit 3 (for instance) might appear as an idealized version without extraneous strokes. This configuration represents the model's interpretation of a quintessential '3' with a prominent probability peak. Any minor alteration, like flipping a single pixel, lowers the altered image's probability. The sampler has thus learned to prioritize exact, pristine versions of each digit, marking any deviation from this high-probability state as unlikely.

---

[4]GitHub Repository: `https://github.com/jam7/tsp/tree/master`; TSPLIB Archive: `http://comopt.ifi.uni-heidelberg.de/software/TSPLIB95/`.

For MNIST (LeCun et al., 2010), this narrow focus limits flexibility. The model assigns high probability to only a few "perfect" digit versions, treating minor variations as less probable. This rigidity makes the generated images sensitive to small changes and limits the RBM's ability to recognize natural, varied handwriting. In the context of RBMs, sampling from flat modes explores a wider range of latent handwritten styles, enhancing the model's ability to capture the underlying data distribution. This reflects a broader representation of possible input variations, crucial for tasks like image generation and data reconstruction. In practice, this means that images generated from flat modes in RBMs are less likely to overfit to sharp, specific patterns in the training data and are instead more reflective of the variability inherent in the dataset.

A simple convolutional autoencoder (CAE) was used for image generation and reconstruction, allowing us to evaluate the performance and generalization capability of sampler-generated data. We train our CAE under sampler-generated images and tested them under various conditions. Initially, clean test data was used to establish baseline performance. Subsequently, we introduced Gaussian noise to evaluate the models' resilience against perturbations, a common method for assessing adversarial robustness (Madry et al., 2018). Additionally, we examined the models with occluded images, where random sections of the images were obscured by zero-valued pixel blocks. This test simulates scenarios with missing or obstructed information, a widely used technique in robustness studies to measure model performance under partial information loss (Zhang et al., 2019). For quantitative evaluation, we rely on negative reconstruction error as a surrogate for negative log-likelihood(NLL) under a Gaussian-decoder assumption to directly measure how well our method captures the underlying data distribution and, therefore, its ability to generalize (Kingma & Welling, 2022).

Table 2: EDMALA attaining the highest average surrogate NLL across 3 distinct settings on MNIST.

| Setting | BG-1 | HB-10-1 | DULA | DMALA | EDULA | EDMALA |
|---------|------|---------|------|-------|-------|--------|
| Clean | $-0.0157 \pm 0.0014$ | $-0.0134 \pm 0.0009$ | $-0.0209 \pm 0.0015$ | $-0.0156 \pm 0.0011$ | $-0.0112 \pm 0.0013$ | $\mathbf{-0.0066} \pm 0.0014$ |
| Noisy | $0.0144 \pm 0.0013$ | $0.0165 \pm 0.0011$ | $0.0097 \pm 0.0013$ | $0.0148 \pm 0.0009$ | $0.0183 \pm 0.0010$ | $\mathbf{0.0233} \pm 0.0010$ |
| Occluded | $-0.0179 \pm 0.0013$ | $-0.0154 \pm 0.0008$ | $-0.0233 \pm 0.0014$ | $-0.0182 \pm 0.0010$ | $-0.0138 \pm 0.0013$ | $\mathbf{-0.0098} \pm 0.0015$ |

The results in Table 2 indicate that EDLP methods consistently outperform their non-entropic counterparts across all test settings; EDULA improves over DULA by 46.4%, 88.7%, and 40.8% , and EDMALA improves over DMALA by 57.7%, 57.4%, and 46.2% respectively across clean, noisy, and occluded settings. The average time per sample (in seconds) for BG-1, HB-10-1, DULA, DMALA, EDULA, and EDMALA are $0.0015 \pm 0.0015$, $0.0056 \pm 0.0034$, $0.0017 \pm 0.0010$, $0.0048 \pm 0.0041$, $0.0020 \pm 0.0012$, and $0.0052 \pm 0.0036$, respectively. Overall, EDMALA achieves the best log-likelihood across all settings. This suggests that entropy guidance helps with superior generalization capabilities, making it especially effective for reconstructing unseen data accurately.

### 6.4 Binary Bayesian Neural Networks

Binary Bayesian Neural Networks (BBNNs) provide a compelling framework for deploying deep learning models in resource-constrained environments. By constraining weights to binary states ($\{-1, +1\}$), these networks can theoretically achieve a $32\times$ reduction in memory footprint compared to standard 32-bit floating-point networks, and significant acceleration via bitwise operations (Courbariaux et al., 2016; Hubara et al., 2017). Inspired by recent advances like BNN-SAM (Pu et al., 2024), which show that converging to flat minima is crucial for generalization in binary and discrete settings, we leverage our discrete sampling technique to target these robust regions. In particular, we explore the training of BBNNs on regression tasks across four UCI datasets (Dua & Graff, 2017), under identical settings (data-splits, dataset choices, training etc.) from Section 7.4 of Zhang et al. (2022). The energy function for each dataset is defined as:

$$U(\boldsymbol{\theta}) = -\sum_{i=1}^{N} ||f_{\boldsymbol{\theta}}(x_i) - y_i||^2,$$

where $D = \{x_i, y_i\}_{i=1}^{N}$ is the training dataset, and $f_{\boldsymbol{\theta}}$ denotes a two-layer neural network with `Tanh` activation and 500 hidden neurons. After burning initial samples, we calculate and report the average test RMSE (root

mean squared error) and its standard deviation. As shown in Table 3, EDMALA demonstrates improved performance on all baselines across **all** datasets. Overall, these results confirm that our method enhances generalization performance on unseen test data.

We present a runtime analysis on the Adult dataset in Figure 5 in the Appendix alongside the EDLP-GLU versions. We provide additional results in the Appendix F.4.

Table 3: EDMALA achieving least average test RMSE for various datasets.

| Dataset | **Gibbs** | **GWG** | **DULA** | **DMALA** | **EDULA** | **EDMALA** |
|---------|-----------|---------|----------|-----------|-----------|------------|
| COMPAS | $0.4850_{\pm 0.0077}$ | $0.4779_{\pm 0.0063}$ | $0.4853_{\pm 0.0044}$ | $0.4855_{\pm 0.0047}$ | $0.4789_{\pm 0.0066}$ | $\mathbf{0.4693}_{\pm 0.0018}$ |
| News | $0.1051_{\pm 0.0007}$ | $0.1067_{\pm 0.0010}$ | $0.0977_{\pm 0.0040}$ | $0.0969_{\pm 0.0040}$ | $0.0952_{\pm 0.0038}$ | $\mathbf{0.0948}_{\pm 0.0033}$ |
| Adult | $0.4940_{\pm 0.0041}$ | $0.4861_{\pm 0.0064}$ | $0.4041_{\pm 0.0035}$ | $0.3949_{\pm 0.0108}$ | $0.3952_{\pm 0.0031}$ | $\mathbf{0.3898}_{\pm 0.0024}$ |
| Blog | $0.4295_{\pm 0.0092}$ | $0.4051_{\pm 0.0205}$ | $0.3220_{\pm 0.0091}$ | $0.3198_{\pm 0.0097}$ | $0.3197_{\pm 0.0032}$ | $\mathbf{0.3172}_{\pm 0.0028}$ |

## 7 Discussion

**Limitations.** Since EDLP collects only discrete samples, it produces half as many samples per iteration as EMCMC. The coupling mechanism in Section 4.1 increases the computational load relative to DLP. However, as Li & Zhang (2024) state in their Section 4.2, the cost of gradient computation remains the same for $d$-dimensional models when $\widetilde{\boldsymbol{\theta}}$ resides in a $2d$ dimensional space. EDLP doubles memory usage compared to DLP, but the space complexity remains linear in $d$, ensuring scalability.

A compelling direction for investigation lies in scientific domains where flat-mode sampling in discrete spaces may have principled biological motivation. In molecular biology, the concept of *neutral networks* — sets of sequences with equivalent fitness or folding function, connected through single mutations — is well established for RNA (Schuster et al.; Fontana & Schuster, 1998) and proteins (Maynard Smith, 1970; Kimura, 1983). Since these sequences share similar energies under learned statistical models such as Potts models fit by direct coupling analysis (Cocco et al., 2018), flat-mode samplers are ideal for producing structurally similar candidates with equivalent fitness. Similar structure arises in genetic fine-mapping, where multiple allele configurations explain observed phenotypes equally well (Wang et al., 2020). Beyond this, in molecular optimization and drug discovery (Tan et al., 2025), the activity landscape over chemical space may exhibit broad, flat basins of bioisosteric molecules: structurally similar compounds that share nearly identical binding affinity, biochemical activity, and physicochemical properties (Patani & LaVoie, 1996). We leave this application as a direction for future work.

**Conclusion.** We propose a simple and computationally efficient gradient-based sampler designed for sampling from flat basins in discrete spaces using local entropy as a finite-time heuristic. We provide non-asymptotic convergence guarantees for both the unadjusted and Metropolis-adjusted versions. Empirical results on real-world datasets demonstrate the practical effectiveness of our method across a variety of applications. We hope our framework highlights the importance of flat-mode sampling in discrete systems, with broad utility across scientific and machine learning domains.

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

# A  Table of Notations

Table 4: Summary of probability distributions and notation used throughout this work.

| Notation | Distribution Type | Description |
|---|---|---|
| $p(\boldsymbol{\theta})$ | Marginal, Discrete | Generic Target distribution . |
| $p(\boldsymbol{\theta}_a)$ | Marginal, Continuous | Generic Smoothed version of $p(\boldsymbol{\theta})$; $p(\boldsymbol{\theta}_a) \propto \sum_{\boldsymbol{\theta} \in \Theta} p(\widetilde{\boldsymbol{\theta}})$. |
| $p(\widetilde{\boldsymbol{\theta}})$ | Joint, Hybrid | Generic Coupled distribution $p(\widetilde{\boldsymbol{\theta}}) \propto \exp\left\{U(\boldsymbol{\theta}) - \frac{1}{2\eta}\|\boldsymbol{\theta} - \boldsymbol{\theta}_a\|^2\right\}$, if $p(\boldsymbol{\theta}) \propto \exp(U(\boldsymbol{\theta}))$. |
| $\pi(\boldsymbol{\theta})$ | Marginal, Discrete | Target energy-based distribution $\pi(\boldsymbol{\theta}) \propto \exp(U(\boldsymbol{\theta}))$. |
| $\pi(\boldsymbol{\theta}_a)$ | Marginal, Continuous | Auxiliary distribution $p(\boldsymbol{\theta}_a) \propto \sum_{\boldsymbol{\theta} \in \Theta} \exp(U(\boldsymbol{\theta}) - \frac{1}{2\eta}\|\boldsymbol{\theta} - \boldsymbol{\theta}_a\|^2)$. |
| $\tilde{\pi}$ | Joint, Hybrid | EDULA and EDMALA chains target $\tilde{\pi}$ in finite time in Theorems 5.5, 5.6 $\tilde{\pi} \propto \exp\left\{U(\boldsymbol{\theta}) - \frac{1}{2\eta}\|\boldsymbol{\theta} - \boldsymbol{\theta}_a\|^2\right\}$. |
| $\pi_\gamma$ | Joint, Hybrid | Surrogate Asymptotic distribution for EDULA (no-MH) given step-sizes $\gamma = (\alpha, \alpha_a)$ in Proposition 5.4; $\pi_\gamma \propto \tilde{\pi} \cdot e^{\frac{\alpha_a}{8\eta^2}\|\boldsymbol{\theta} - \boldsymbol{\theta}_a\|^2}$ |

# B  Analysis of the Effect of Flatness Parameter $\eta$

## B.1  Intuition

Figure 3 illustrates the effect of varying the flatness parameter $\eta$ on the probability distribution $p(\boldsymbol{\theta}_a)$ for $\boldsymbol{\theta}$ drawn from a Bernoulli(0.5) distribution. The *layered* curves represent different values of $\eta$, showing how the distribution $p(\boldsymbol{\theta}_a)$ changes as $\eta$ increases.

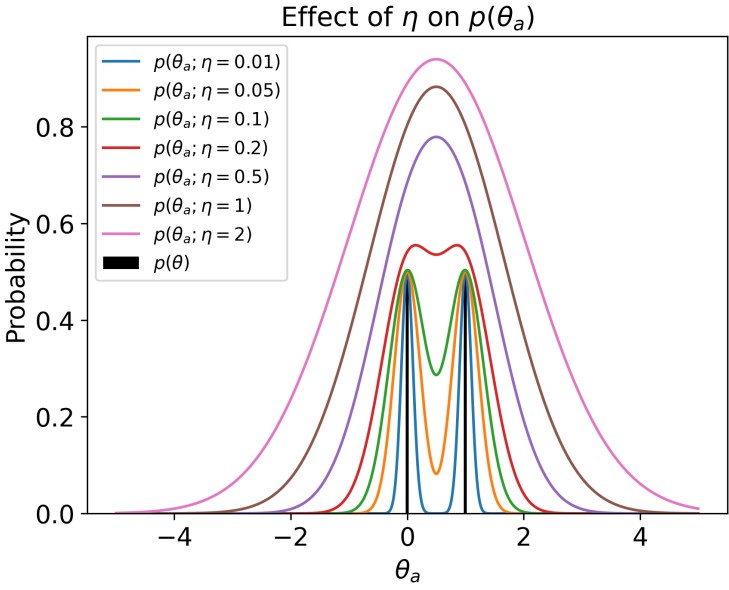

Figure 3: $p(\boldsymbol{\theta_a})$ for $\boldsymbol{\theta} \sim Bernoulli(0.5)$

### Effect of Small $\eta$ (Strong Coupling)

For very small values of $\eta$ (e.g., $\eta = 0.01$, $\eta = 0.05$, $\eta = 0.1$), the curves (blue, orange, and green) are sharply peaked and closely resemble the original $p(\boldsymbol{\theta})$. Small $\eta$ values imply strong coupling between $\boldsymbol{\theta}$ and $\boldsymbol{\theta}_a$. The auxiliary distribution $p(\boldsymbol{\theta}_a)$ remains very close to $p(\boldsymbol{\theta})$, indicating that $\boldsymbol{\theta}_a$ is tightly bound to $\boldsymbol{\theta}$, and the variance is minimal.

**Moderate $\eta$ Values (Moderate Coupling)**

As $\eta$ increases (e.g., $\eta = 0.2$), the curves (red) become wider and smoother. These moderate $\eta$ values adequately capture the flatness of the landscape. The distribution $p(\boldsymbol{\theta}_a)$ starts to diverge from $p(\boldsymbol{\theta})$, allowing $\boldsymbol{\theta}_a$ to explore a broader region around the peaks.

**Large $\eta$ (Weak Coupling)**

For larger values of $\eta$ (e.g., $\eta = 0.5$, $\eta = 1$, $\eta = 2$), the curves (purple, brown, and magenta) are much wider. Large $\eta$ values imply weak coupling between $\boldsymbol{\theta}$ and $\boldsymbol{\theta}_a$. The auxiliary distribution $p(\boldsymbol{\theta}_a)$ is excessively smoothed out compared to $p(\boldsymbol{\theta})$, indicating that $\boldsymbol{\theta}_a$ can explore a much broader range of values with less influence from $\boldsymbol{\theta}$.

**Considerations for $\eta$ Approaching Infinity**

As $\eta$ approaches infinity, the auxiliary distribution $p(\boldsymbol{\theta}_a)$ flattens, and the gradient $\nabla_{\boldsymbol{\theta}_a} U_\eta(\widetilde{\boldsymbol{\theta}})$ tends toward zero. This results in an extremely weak coupling, effectively causing the EDLP framework to behave similarly to a standard DLP. The parameter $\eta$ thus plays a critical role in determining the behavior of the sampler, necessitating careful tuning based on the specific requirements of the sampling task.

## B.2 Sensitivity Analysis

The flatness parameter $\eta$ is arguably the most crucial hyperparameter to optimize in the EDLP algorithm (Algorithm 1). Similar to the hyperparameter tuning ablation strategies employed in Li & Zhang (2024) (Appendix, Section E), we conduct hyperparameter tuning on the COMPAS dataset's validation data. Specifically, we monitor the L2 norm between sampled pairs of $\boldsymbol{\theta}$ and $\boldsymbol{\theta}_a$ for various values of $\eta$. Additionally, we plot the validation RMSE for both EDULA and EDMALA across different values of $\eta$. Finally, we plot the average MH acceptance ratio for EDMALA to assess the impact of $\eta$ on the joint MH acceptance step. We maintain $\alpha = 0.1$ for both samplers and $\alpha_a = 0.01$ for EDULA and $\alpha_a = 0.001$ for EDMALA( see Figure 4).

We observe that as $\eta$ increases, the coupling between the variables weakens, allowing both variables to move more freely, thus increasing the norm. This behavior is consistent across both EDULA and EDMALA. However, EDMALA exhibits a more conservative behavior at the same coupling strength compared to EDULA due to the presence of the joint Metropolis-Hastings (MH) acceptance step, which imposes stricter alignment between the variables, hence maintaining a tighter coupling.

Both samplers demonstrate robustness across a wide range of $\eta$, with relatively stable validation RMSE performance. However, EDULA shows slightly less robustness, particularly at extremely small coupling values, resulting in increased variability and higher RMSE. EDMALA maintains a stable, consistent performance, indicating better robustness to changes in the coupling parameter.

The final plot shows how the MH acceptance probability varies with coupling strength $\eta$ for EDMALA. Initially, with very tight coupling , the acceptance probability is near zero, indicating overly restricted movements due to the strong alignment requirement between the discrete and continuous variables. As $\eta$ increases (coupling relaxes), the acceptance probability rises significantly, reflecting greater freedom in proposing moves that the joint MH criterion accepts. After a certain coupling threshold (around 0.8 here), the acceptance rate plateaus, suggesting diminishing returns from further relaxation in coupling strength. Thus, an intermediate coupling provides a balance, allowing effective exploration without overly compromising the sampler's consistency.

# C Gibbs-like Update Procedure

Gibbs-like updating procedures have been widely employed across various contexts in the sampling literature, particularly within Bayesian hierarchical models, latent variable models, and non-parametric Bayesian approaches. For instance, Gibbs sampling is a fundamental technique in hierarchical Bayesian models, where parameters are partitioned into blocks and updated conditionally on others to facilitate efficient sampling

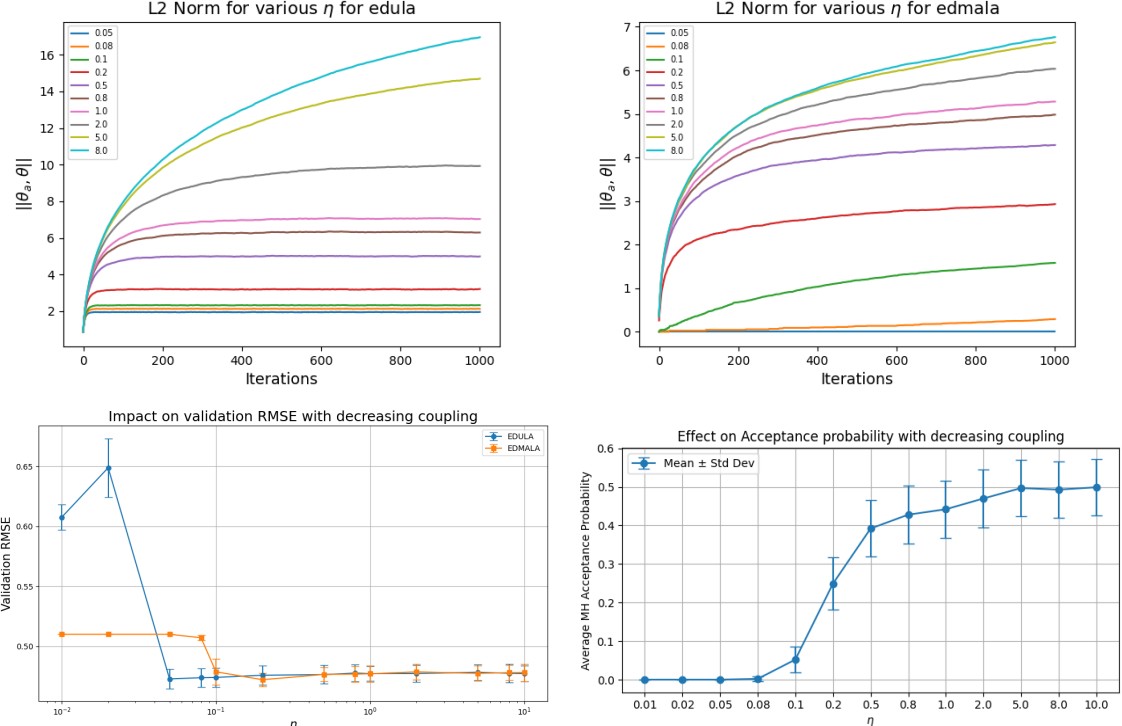

Figure 4: Diagnostics for EDLP

(Casella & George, 1992). In latent variable models, such as Hidden Markov Models (HMMs) and mixture models, Gibbs-like updates allow for alternating between sampling latent variables and model parameters, thereby simplifying the overall process (Diebolt & Robert, 1994). Additionally, these updates are crucial in non-parametric Bayesian approaches, such as Dirichlet Process Mixture Models (DPMMs), where they enable the efficient sampling of cluster assignments and hyperparameters (Neal, 2000). Gibbs-like updates are also prominently used in spatial statistics, particularly in Conditional Autoregressive (CAR) models, where the value at each spatial location is updated based on its neighbors (Besag, 1974).

Since our goal is to sample from a joint distribution, rather than simultaneously updating $\boldsymbol{\theta}$ and $\boldsymbol{\theta}_a$, we alternatively update these variables iteratively. The conditional distribution for the primary variable $\boldsymbol{\theta}$ is given by:

$$p(\boldsymbol{\theta}|\boldsymbol{\theta}_a) \propto \frac{1}{Z_{\boldsymbol{\theta}_a}} \exp\left\{ U(\boldsymbol{\theta}) - \frac{1}{2\eta}\|\boldsymbol{\theta} - \boldsymbol{\theta}_a\|^2 \right\},$$

where $Z_{\boldsymbol{\theta}_a} = \exp \mathcal{F}(\boldsymbol{\theta}_a; \eta)$ serves as the normalization constant. Correspondingly, the conditional distribution for the auxiliary variable $\boldsymbol{\theta}_a$ is:

$$p(\boldsymbol{\theta}_a|\boldsymbol{\theta}) \propto \frac{1}{Z_{\boldsymbol{\theta}}} \exp\left\{ -\frac{1}{2\eta}\|\boldsymbol{\theta} - \boldsymbol{\theta}_a\|^2 \right\},$$

where $Z_{\boldsymbol{\theta}} = \exp\left(U(\boldsymbol{\theta})\right)$ is the associated normalization constant. This formulation reveals that $\boldsymbol{\theta}_a$ is sampled from $\mathcal{N}(\boldsymbol{\theta}, \eta\boldsymbol{I})$, with the variance $\eta$ controlling the expected distance between $\boldsymbol{\theta}$ and $\boldsymbol{\theta}_a$. During the Metropolis-Hastings (MH) step, the acceptance probability is now calculated as:

$$\min\left(1, \frac{q_\alpha(\boldsymbol{\theta}|\widetilde{\boldsymbol{\theta}'})}{q_\alpha(\boldsymbol{\theta}'|\widetilde{\boldsymbol{\theta}})} \frac{\pi(\widetilde{\boldsymbol{\theta}'})}{\pi(\widetilde{\boldsymbol{\theta}})}\right).$$

This Gibbs-like alternating update scheme offers distinct advantages: (1) exact sampling of $\boldsymbol{\theta}_a$, (2) elimination of the need for the $\alpha_a$ parameter, (3) a less intensive computation of the MH acceptance probability, and (4) reduced overall computational overhead, especially when the proposal step involves an MH correction. This gibbs-like updating also shares similarities with the proximal sampling methods (Pereyra, 2016; Liang & Chen, 2023). This innovation can potentially allow DLP to generalize effectively to more complex, high-dimensional, and non-differentiable discrete target distributions such as the discrete Laplace distribution, which is commonly used in privacy-preserving mechanisms(Dwork et al., 2006; Gupte & Sundararajan, 2010). We leave out the theoretical analysis of the GLU versions for future work.

In Figure 5, we present the measured elapsed time per sample for the adult dataset to demonstrate these computational efficiencies, under the same settings as in Section 6, extending to include the GLU versions of the EDLP framework(Section C), alongside the results for the standard DLP and EDLP methods.

As illustrated, the EDLP versions exhibit an increase in runtime compared to DLP, due to the modifications discussed in Section 4.1. While the runtime difference between the DULA and EDULA algorithms (without MH correction) is negligible, the time difference between DMALA and EDMALA is more pronounced. This can be attributed to the more complex joint acceptance probability calculation required by EDMALA. Despite these variations, the overall runtime overhead for EDLP samplers is not substantial and remains practical.

For the EDLP-GLU variants, we maintained the same $\eta$ and $\alpha$ values as their corresponding vanilla DLP samplers. The EDLP-GLU variants naturally achieve an approximate 50% reduction in runtime compared to EDLP. This efficiency stems from the alternating updates between sampling from a modified isotropic Gaussian and conditional DLP, designed to match the conditional distributions more effectively. However, this approach also introduces a higher standard deviation in runtime. The variability is primarily attributed to the contrasting computational costs between the two update types: sampling from the modified Gaussian is relatively lightweight, whereas the conditional DLP update is computationally intensive. As a result, the EDLP-GLU variants exhibit greater fluctuations in runtime compared to other samplers. Furthermore, the negative lower bounds are not physically meaningful and stem from the high variability in runtime measurements.

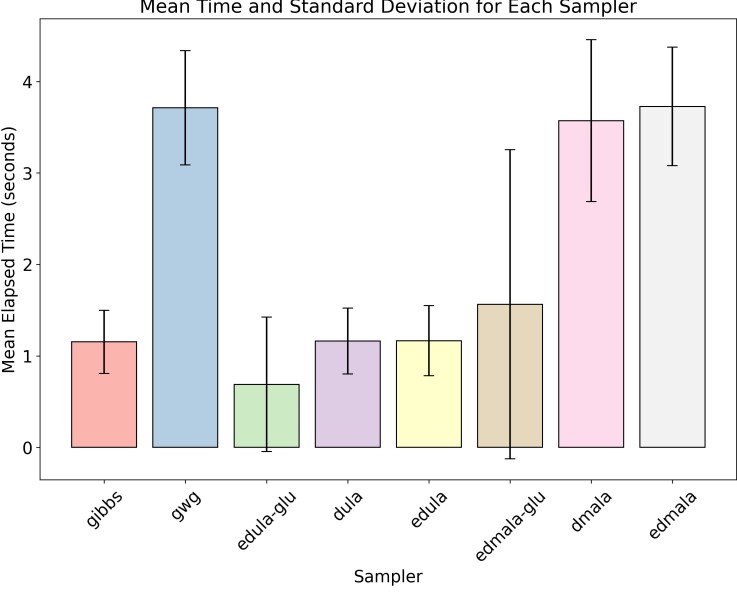

Figure 5: Runtime Analysis on Adult Dataset

# D   Considerations for Excluding Focussed Belief Propagation from Benchmarking

**1. Fundamental Differences in Sampling Mechanism:**   Most of the sampling algorithms we use generate samples sequentially, with each sample $x_{t+1}$ derived from the previous sample $x_t$. This sequential dependency is essential for building a Markov Chain that explores the distribution space and gradually converges to the target distribution. fBP produces samples sequentially, but instead employs a *message-passing algorithm* aimed at converging to a fixed solution or configuration. It operates to converge deterministically to a solution, rather than generating a sequence of probabilistic samples. Moreover, fBP lacks a formal proof of convergence, relying instead on heuristic principles rooted in replica theory. This absence of theoretical guarantees or established convergence rates means that even if fBP appears to perform well, we cannot interpret or quantify its reliability, efficiency, or consistency across varying datasets and tasks. In contrast, MCMC-based methods like Langevin dynamics and Gibbs sampling come with well-understood convergence properties, enabling meaningful performance evaluations and robust benchmarking. This interpretability gap makes fBP less suitable for our study, where theoretical soundness and predictable behavior are critical.

**2. Technical and Practical Constraints with using fBP:** While fBP is originally implemented in Julia[5], a Python wrapper[6] is also available. However, this wrapper still depends on the underlying Julia or C++ implementations, introducing potential cross-language communication overhead. This dependency complicates integration in Python workflows and creates an inherent performance disparity when compared to purely Pythonic implementations, making direct runtime comparisons less meaningful. Despite fBP's speed advantage, its execution becomes slow as sample dimensions increase and network ensembles grow larger. The volume of message-passing in high-dimensional contexts limits its scalability. As task complexity increases, fBP faces challenges in achieving stable convergence, further limiting its suitability for our high-dimensional setup. Past studies have excluded computationally expensive methods from experimental evaluations Zhang et al. (2022).

**3. Computational Overhead and Efficiency Concerns Resource Demands for Multiple Runs:** If we were to use fBP to generate multiple samples, we would need to reinitialize and re-run the algorithm for each sample with a new seed, effectively solving the problem from scratch each time. This is highly inefficient compared to MCMC methods, where each subsequent sample builds on the previous one without needing to restart the entire algorithm. For larger models and datasets, this repeated initialization and execution would result in a significant computational burden.

**4. Nature of Tasks:** In certain structured sampling tasks, such as the TSP, we enforce constraints to ensure that each proposed state is a valid TSP solution. This entails accepting only those configurations that satisfy specific requirements of the TSP. However, fBP does not adhere to such constraints, as it lacks mechanisms for directly enforcing the validity of the sampled states. Consequently, fBP is unsuitable for tasks where such structural constraints are critical, placing it outside the scope for comparison in these applications.

We conducted preliminary experiments using fBP for Restricted Boltzmann Machine (RBM) sampling on the MNIST dataset to assess its effectiveness in image generation. Figure 6 shows random image samples generated by fBP on MNIST, which resemble random unstructured noise rather than recognizable digits, compared to MNIST samples by DMALA and EDMALA in Figures 7, 8 respectively. These outputs suggest that fBP doesn't capture the underlying structure of the MNIST data.

---

[5]Carlo   Baldassi,   *BinaryCommitteeMachinefBP.jl*,   GitHub   repository,   `https://github.com/carlobaldassi/BinaryCommitteeMachinefBP.jl`, accessed November 8, 2024.

[6]Curti, Nico and Dall'Olio, Daniele and Giampieri, Enrico, *ReplicatedFocusingBeliefPropagation*, GitHub repository, `https://github.com/Nico-Curti/rFBP`, accessed November 8, 2024.

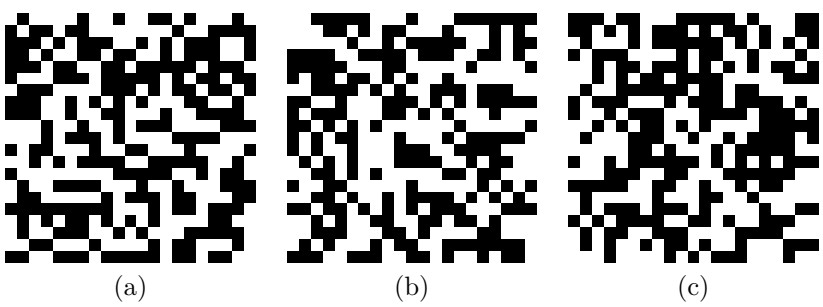

(a)                    (b)                    (c)

Figure 6: Random Image Samples for MNIST using fBP

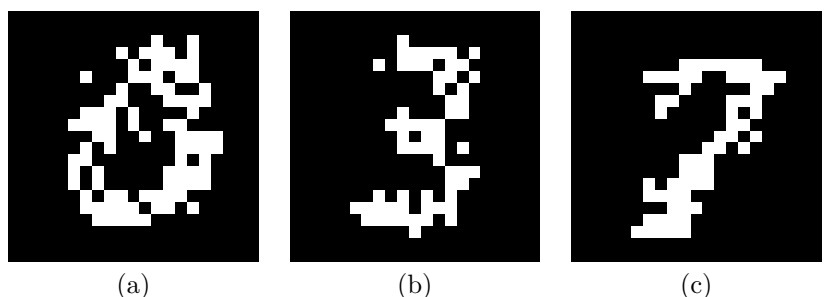

(a)                    (b)                    (c)

Figure 7: Random Image Samples for MNIST using DMALA

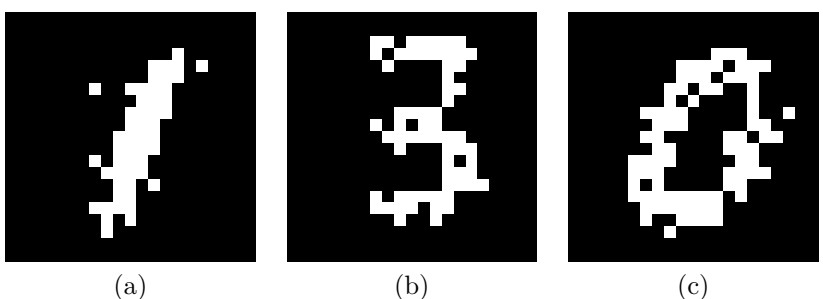

(a)                    (b)                    (c)

Figure 8: Random Image Samples for MNIST using EDMALA

fBP lacks direct use of the energy function $U(.)$ during optimization, preventing accurate data modeling. Figure 9 illustrates this through a distribution analysis of generated MNIST classes, showing significant mode collapse. Most generated samples cluster around a few classes, with an imbalance favoring certain digits and ignoring others.

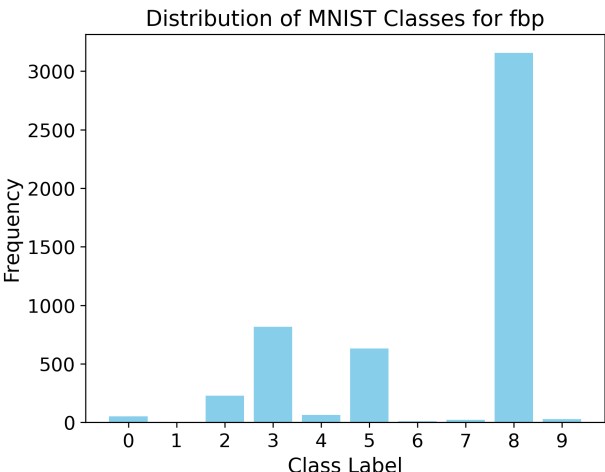

Figure 9: Mode Collapse using fBP

These findings highlight a fundamental issue with fBP in image generation tasks. Mode collapse suggests fBP struggles to explore diverse data regions, making it unsuitable for generating realistic, structured outputs that adhere to specific distribution characteristics, like image data in the MNIST dataset.

In summary, fBP diverges significantly from the MCMC-based sampling methods used in our study due to its deterministic message-passing mechanism, which converges to fixed configurations rather than generating sequential probabilistic samples. While a Python wrapper exists, its reliance on the underlying Julia or C++ implementations introduces potential cross-language communication overhead, creating performance inconsistencies when compared to native Python implementations. Moreover, fBP's lack of constraint adherence and dependence on spin-like variable encoding make it unsuitable for complex, structured sampling tasks like TSP or data-driven applications requiring diverse sampling, such as image generation on MNIST. Our preliminary experiments confirm that fBP struggles with mode collapse and fails to capture essential data distribution characteristics.

## E  Proofs

### E.1  Proof of Lemma 4.1

Assume $\widetilde{\boldsymbol{\theta}} = [\boldsymbol{\theta}^T, \boldsymbol{\theta}_a^T]^T$ is sampled from the joint posterior distribution:

$$p(\widetilde{\boldsymbol{\theta}}) = p(\boldsymbol{\theta}, \boldsymbol{\theta}_a) \propto \exp\left\{ U(\boldsymbol{\theta}) - \frac{1}{2\eta} \|\boldsymbol{\theta} - \boldsymbol{\theta}_a\|^2 \right\}. \tag{10}$$

Then the marginal distribution for $\boldsymbol{\theta}$ is:

$$
\begin{aligned}
p(\boldsymbol{\theta}) &= \int p(\boldsymbol{\theta}, \boldsymbol{\theta}_a) d\boldsymbol{\theta}_a \\
&= (2\pi\eta)^{-\frac{d}{2}} Z^{-1} \int \exp\left\{ U(\boldsymbol{\theta}) - \frac{1}{2\eta} \|\boldsymbol{\theta} - \boldsymbol{\theta}_a\|^2 \right\} d\boldsymbol{\theta}_a \\
&= Z^{-1} \exp(U(\boldsymbol{\theta})) (2\pi\eta)^{-\frac{d}{2}} \int \exp\left\{ -\frac{1}{2\eta} \|\boldsymbol{\theta} - \boldsymbol{\theta}_a\|^2 \right\} d\boldsymbol{\theta}_a \\
&= Z^{-1} \exp(U(\boldsymbol{\theta})),
\end{aligned}
\tag{11}
$$

where $Z = \sum_{\boldsymbol{\Theta}} \exp(U(\boldsymbol{\theta}))$ is the normalizing constant, and it is obtained by:

$$\sum_{\boldsymbol{\Theta}} \int \exp\left\{ U(\boldsymbol{\theta}) - \frac{1}{2\eta} \|\boldsymbol{\theta} - \boldsymbol{\theta}_a\|^2 \right\} d\boldsymbol{\theta}_a = (2\pi\eta)^{\frac{d}{2}} \sum_{\boldsymbol{\Theta}} \exp(U(\boldsymbol{\theta})) := (2\pi\eta)^{\frac{d}{2}} Z. \tag{12}$$

This verifies that the joint posterior distribution $p(\boldsymbol{\theta}, \boldsymbol{\theta}_a)$ is mathematically well-defined[7]. Similarly, the marginal distribution for $\boldsymbol{\theta}_a$ is:

$$
\begin{aligned}
p(\boldsymbol{\theta}_a) &= \sum_{\boldsymbol{\Theta}} p(\boldsymbol{\theta}, \boldsymbol{\theta}_a) \\
&\propto \sum_{\boldsymbol{\Theta}} \exp\left\{ U(\boldsymbol{\theta}) - \frac{1}{2\eta} \|\boldsymbol{\theta} - \boldsymbol{\theta}_a\|^2 \right\} \\
&= \exp \mathcal{F}(\boldsymbol{\theta}_a; \eta).
\end{aligned}
\tag{13}
$$

### E.2 Proof of Proposition 5.4

We follow a similar-style analysis as seen in Theorem 5.1 of Zhang et al. (2022).

Using Equation equation 9,

$$
\begin{aligned}
q_\gamma(\widetilde{\boldsymbol{\theta}}' | \widetilde{\boldsymbol{\theta}}) &\propto \exp\left( \frac{1}{2} \nabla_{\boldsymbol{\theta}} U_\eta(\widetilde{\boldsymbol{\theta}})^\top (\boldsymbol{\theta}' - \boldsymbol{\theta}) - \frac{1}{2\alpha} \|\boldsymbol{\theta}' - \boldsymbol{\theta}\|^2 \right) \cdot \frac{1}{\sqrt{2\pi\alpha_a}^d} \exp\left( -\frac{1}{2\alpha_a} \|\boldsymbol{\theta}'_a - \boldsymbol{\theta}_a - \frac{\alpha_a}{2} \nabla_{\boldsymbol{\theta}_a} U_\eta(\widetilde{\boldsymbol{\theta}})\|^2 \right) \\
&= \frac{1}{\sqrt{2\pi\alpha_a}^d} \exp\left( \frac{1}{2} \nabla_{\boldsymbol{\theta}} U(\boldsymbol{\theta})^\top (\boldsymbol{\theta}' - \boldsymbol{\theta}) - \frac{1}{2\alpha} \|\boldsymbol{\theta}' - \boldsymbol{\theta}\|^2 - \frac{1}{2\eta} (\boldsymbol{\theta} - \boldsymbol{\theta}_a)^\top (\boldsymbol{\theta}' - \boldsymbol{\theta}) \right) \cdot \\
&\quad \left( -\frac{1}{2\alpha_a} \|\boldsymbol{\theta}'_a - \boldsymbol{\theta}_a\|^2 + \frac{1}{2\eta} (\boldsymbol{\theta} - \boldsymbol{\theta}_a)^\top (\boldsymbol{\theta}'_a - \boldsymbol{\theta}_a) - \frac{\alpha_a}{8\eta^2} \|\boldsymbol{\theta} - \boldsymbol{\theta}_a\|^2 \right) \\
&= \frac{1}{\sqrt{(2\pi\alpha_a)^d}} \exp\left( \frac{1}{2}(-U(\boldsymbol{\theta}) + U(\boldsymbol{\theta}')) - (\boldsymbol{\theta} - \boldsymbol{\theta}')^\top \left( \frac{1}{2\alpha} I + \frac{1}{4} \int_0^1 \nabla^2 U((1-s)\boldsymbol{\theta} + s\boldsymbol{\theta}')ds \right) (\boldsymbol{\theta} - \boldsymbol{\theta}') \right. \\
&\quad \left. -\frac{1}{2\eta} (\boldsymbol{\theta} - \boldsymbol{\theta}_a)^\top (\boldsymbol{\theta}' - \boldsymbol{\theta} + \boldsymbol{\theta}_a - \boldsymbol{\theta}'_a) - \frac{1}{2\alpha_a} \|\boldsymbol{\theta}'_a - \boldsymbol{\theta}_a\|^2 - \frac{\alpha_a}{8\eta^2} \|\boldsymbol{\theta} - \boldsymbol{\theta}_a\|^2 \right) \\
&= \frac{1}{\sqrt{(2\pi\alpha_a)^d}} \exp\left( \frac{1}{2}(-U(\boldsymbol{\theta}) + U(\boldsymbol{\theta}')) - (\boldsymbol{\theta} - \boldsymbol{\theta}')^\top \left( \frac{1}{2\alpha} I + \frac{1}{4} \int_0^1 \nabla^2 U((1-s)\boldsymbol{\theta} + s\boldsymbol{\theta}')ds \right) (\boldsymbol{\theta} - \boldsymbol{\theta}') \right. \\
&\quad \left. -\frac{1}{2\eta} (\boldsymbol{\theta} - \boldsymbol{\theta}_a)^\top (\boldsymbol{\theta}' - \boldsymbol{\theta}'_a) - \frac{1}{2\alpha_a} \|\boldsymbol{\theta}'_a - \boldsymbol{\theta}_a\|^2 + \frac{4\eta - \alpha_a}{8\eta^2} \|\boldsymbol{\theta} - \boldsymbol{\theta}_a\|^2 \right)
\end{aligned}
$$

The third line is true because we replace the linear gradient term $\nabla_{\boldsymbol{\theta}} U(\boldsymbol{\theta})^\top (\boldsymbol{\theta}' - \boldsymbol{\theta})$ with Taylor expansion with integral remainder.

The normalizing constant for Equation equation 9 $Z_{\widetilde{\boldsymbol{\Theta}}}(\widetilde{\boldsymbol{\theta}})$ is computed by integrating over $\mathbb{R}^d$ and summing over $\boldsymbol{\Theta}$:

$$
Z_{\widetilde{\boldsymbol{\Theta}}}(\widetilde{\boldsymbol{\theta}}) = \frac{1}{\sqrt{2\pi\alpha_a}^d} \int_{\boldsymbol{\theta}'_a} \sum_{\boldsymbol{\theta}' \in \boldsymbol{\Theta}} \exp\left( \frac{1}{2} \nabla_{\boldsymbol{\theta}} U_\eta(\widetilde{\boldsymbol{\theta}})^\top (\boldsymbol{\theta}' - \boldsymbol{\theta}) - \frac{1}{2\alpha} \|\boldsymbol{\theta}' - \boldsymbol{\theta}\|^2 - \frac{1}{2\alpha_a} \|\boldsymbol{\theta}'_a - \boldsymbol{\theta}_a - \frac{\alpha_a}{2} \nabla_{\boldsymbol{\theta}_a} U_\eta(\widetilde{\boldsymbol{\theta}})\|^2 \right) d\boldsymbol{\theta}'_a
\tag{14}
$$

We note that since $\nabla^2 U(\cdot)$ is continuous( from Assumption 5.2), we know that

$$
\min_{x,y \in \boldsymbol{\Theta}} (x-y)^T \left( \int_0^1 \nabla^2 U((1-s)x + sy)ds \right) (x-y)
$$

is well-defined.

Consequently, the modified normalizing constant(Equation equation 14), $Z_\gamma(\widetilde{\boldsymbol{\theta}})$, becomes

$$
\begin{aligned}
Z_\gamma(\widetilde{\boldsymbol{\theta}}) &= \frac{1}{\sqrt{(2\pi\alpha_a)^d}} \int_{\boldsymbol{\theta}'_a} \sum_{\boldsymbol{\theta}' \in \boldsymbol{\Theta}} \exp\left( \frac{1}{2}\left( -U(\boldsymbol{\theta}) + U(\boldsymbol{\theta}') \right) - (\boldsymbol{\theta} - \boldsymbol{\theta}')^\top \left( \frac{1}{2\alpha} I + \frac{1}{4} \int_0^1 \nabla^2 U\left( (1-s)\boldsymbol{\theta} + s\boldsymbol{\theta}' \right) ds \right) (\boldsymbol{\theta} - \boldsymbol{\theta}') \right. \\
&\quad \left. -\frac{1}{2\eta} (\boldsymbol{\theta} - \boldsymbol{\theta}_a)^\top (\boldsymbol{\theta}' - \boldsymbol{\theta}'_a) - \frac{1}{2\alpha_a} \|\boldsymbol{\theta}'_a - \boldsymbol{\theta}_a\|^2 + \frac{4\eta - \alpha_a}{8\eta^2} \|\boldsymbol{\theta} - \boldsymbol{\theta}_a\|^2 \right)
\end{aligned}
\tag{15}
$$

---

[7]The exact form of the joint posterior is $p(\boldsymbol{\theta}, \boldsymbol{\theta}_a) = (2\pi\eta)^{-\frac{d}{2}} Z^{-1} \exp(U(\boldsymbol{\theta}) - \frac{1}{2\eta} \|\boldsymbol{\theta} - \boldsymbol{\theta}_a\|^2)$.

Now, we establish that $q_\gamma(\widetilde{\boldsymbol{\theta}}|\widetilde{\boldsymbol{\theta}}')$ is reversible with respect to $\pi_\gamma$, where

$$\pi_\gamma = \frac{Z_\gamma(\widetilde{\boldsymbol{\theta}}) \exp\{\frac{\alpha_a}{8\eta^2}\|\boldsymbol{\theta} - \boldsymbol{\theta}_a\|^2\}\pi(\widetilde{\boldsymbol{\theta}})}{\int_y \sum_{x \in \Theta} Z_\gamma([x^\top, y^\top]^\top) \exp \frac{\alpha_a}{8\eta^2}\|x - y\|^2 \pi([x^\top, y^\top]^\top) dy}$$

.

Note that,

$$\pi_\gamma(\widetilde{\boldsymbol{\theta}})q_\gamma(\widetilde{\boldsymbol{\theta}}'|\widetilde{\boldsymbol{\theta}}) = \frac{Z_\gamma(\widetilde{\boldsymbol{\theta}}) \exp\left(\frac{\alpha_a}{8\eta^2}\|\boldsymbol{\theta} - \boldsymbol{\theta}_a\|^2\right) \pi(\widetilde{\boldsymbol{\theta}})}{\int_y \sum_{x \in \Theta} Z_\gamma([x^\top, y^\top]^\top) \exp \left(\frac{\alpha_a}{8\eta^2}\|x - y\|^2\right) \pi([x^\top, y^\top]^\top) dy} \frac{1}{Z_\gamma(\widetilde{\boldsymbol{\theta}})} \frac{1}{(\sqrt{2\pi\alpha_a})^d}$$

$$\exp\left(\frac{1}{2}\left(- U(\boldsymbol{\theta}) + U(\boldsymbol{\theta}')\right) - (\boldsymbol{\theta} - \boldsymbol{\theta}')^\top \left(\frac{1}{2\alpha}I + \frac{1}{4}\int_0^1 \nabla^2 U((1-s)\boldsymbol{\theta} + s\boldsymbol{\theta}') ds\right)(\boldsymbol{\theta} - \boldsymbol{\theta}')\right.$$

$$\left. - \frac{1}{2\eta}(\boldsymbol{\theta} - \boldsymbol{\theta}_a)^\top(\boldsymbol{\theta}' - \boldsymbol{\theta}_a') - \frac{1}{2\alpha_a}\|\boldsymbol{\theta}_a' - \boldsymbol{\theta}_a\|^2 + \frac{4\eta - \alpha_a}{8\eta^2}\|\boldsymbol{\theta} - \boldsymbol{\theta}_a\|^2\right)$$

$$= \frac{1}{\int_y \sum_{x \in \Theta} Z_\gamma([x^\top, y^\top]^\top) \exp \left(\frac{\alpha_a}{8\eta^2}\|x - y\|^2\right) \pi([x^\top, y^\top]^\top) dy} \frac{1}{(\sqrt{2\pi\alpha_a})^d}$$

$$\exp\left(\frac{1}{2}\left(U(\boldsymbol{\theta}) + U(\boldsymbol{\theta}')\right) - \frac{1}{2}(\boldsymbol{\theta} - \boldsymbol{\theta}')^\top \left(\frac{1}{\alpha}I + \frac{1}{2}\int_0^1 \nabla^2 U((1-s)\boldsymbol{\theta} + s\boldsymbol{\theta}') ds\right)(\boldsymbol{\theta} - \boldsymbol{\theta}')\right.$$

$$\left. - \frac{1}{2\eta}(\boldsymbol{\theta} - \boldsymbol{\theta}_a)^\top(\boldsymbol{\theta}' - \boldsymbol{\theta}_a') - \frac{1}{2\alpha_a}\|\boldsymbol{\theta}_a' - \boldsymbol{\theta}_a\|^2\right)$$

$$= \pi_\gamma(\boldsymbol{\theta}')q_\gamma(\boldsymbol{\theta}|\boldsymbol{\theta}').$$

Chain looks symmetric and reversible with respect to $\pi_\gamma$.

Now, given this, note that $Z_\gamma'(\widetilde{\boldsymbol{\theta}})$ converges to 1 as $\alpha \to 0$ and $\alpha_a \to 0$.

$$Z_\gamma'(\widetilde{\boldsymbol{\theta}}) = Z_\gamma(\widetilde{\boldsymbol{\theta}}) \exp\left(\frac{\alpha_a}{8\eta^2}\|\boldsymbol{\theta} - \boldsymbol{\theta}_a\|^2\right)$$

$$= \frac{1}{\sqrt{(2\pi\alpha_a)^d}} \int_y \sum_x \exp\left(- \frac{1}{2}(U(\boldsymbol{\theta}) - U(x)) - (\boldsymbol{\theta} - x)^\top \left(\frac{1}{2\alpha}I + \frac{1}{4}\int_0^1 \nabla^2 U((1-s)\boldsymbol{\theta} + s\boldsymbol{\theta}') ds\right)(\boldsymbol{\theta} - x)\right.$$

$$\left. - \frac{1}{2\alpha_a}\|y - \boldsymbol{\theta}_a\|^2 + \frac{4\eta}{8\eta^2}\|\boldsymbol{\theta} - \boldsymbol{\theta}_a\|^2\right) dy$$

$$\stackrel{=}{\scriptstyle \alpha \to 0} \frac{1}{\sqrt{(2\pi\alpha_a)^d}} \int_y \sum_x \exp\left(\frac{1}{2}(U(x) - U(\boldsymbol{\theta})) - \frac{1}{2\alpha_a}\|y - \boldsymbol{\theta}_a\|^2 + \frac{1}{2\eta}\|\boldsymbol{\theta} - \boldsymbol{\theta}_a\|^2 - \frac{1}{2\eta}(\boldsymbol{\theta} - \boldsymbol{\theta}_a)^\top(x - y)\right)\delta_{\boldsymbol{\theta}}(x) dy$$

$$= \int_y \exp\left(\frac{1}{2\eta}\|\boldsymbol{\theta} - \boldsymbol{\theta}_a\|^2 - \frac{1}{2\eta}(\boldsymbol{\theta} - \boldsymbol{\theta}_a)^\top(\boldsymbol{\theta} - y)\right) dy$$

$$\stackrel{=}{\scriptstyle \alpha_a \to 0} \int_y \exp\left(\frac{1}{2\eta}\|\boldsymbol{\theta} - \boldsymbol{\theta}_a\|^2 - \frac{1}{2\eta}(\boldsymbol{\theta} - \boldsymbol{\theta}_a)^\top(\boldsymbol{\theta} - \boldsymbol{\theta}_a)\right) dy$$

$$= 1.$$

where $\delta_\theta(.)$ is a Dirac delta. It follows that $\pi_\gamma$ converges pointwisely to $\pi(\widetilde{\boldsymbol{\theta}})$ as $\alpha \to 0$ and $\alpha_a \to 0$.

By Scheffé's Lemma, this implies global convergence in total variation distance. Let us consider the convergence rate in terms of the $L_1$-norm

$$\|\pi_\gamma - \tilde\pi\|_1 = \int_{\theta_a} \sum_{\theta \in \boldsymbol\Theta} \left| \frac{Z'_\gamma(\widetilde{\boldsymbol\theta})\pi(\widetilde{\boldsymbol\theta})}{\int_y \sum_{x \in \boldsymbol\Theta} Z'_\gamma([x^\top, y^\top]^\top)\pi([x^\top, y^\top]^\top)dy} - \pi(\widetilde{\boldsymbol\theta}) \right| d\boldsymbol\theta_a$$

We write out each absolute value term

$$\left| \frac{Z'_\gamma(\widetilde{\boldsymbol\theta})\pi(\widetilde{\boldsymbol\theta})}{\int_y \sum_{x \in \boldsymbol\Theta} Z'_\gamma([x^\top, y^\top]^\top)\pi([x^\top, y^\top]^\top)dy} - \pi(\widetilde{\boldsymbol\theta}) \right| = \pi(\widetilde{\boldsymbol\theta}) \left| \frac{Z'_\gamma(\widetilde{\boldsymbol\theta})}{\int_y \sum_{x \in \boldsymbol\Theta} Z'_\gamma([x^\top, y^\top]^\top)\pi([x^\top, y^\top]^\top)dy} - 1 \right|$$

First, recall under Assumption 5.1, we note that since $U$ is M-gradient Lipschitz and $\frac{\alpha}{2} < \frac{1}{M}$, this makes matrix

$$\frac{1}{2\alpha}I - \frac{1}{4}\int_0^1 \nabla^2 U((1-s)\boldsymbol\theta + s\boldsymbol\theta')ds > \frac{1}{4}\left(\frac{2}{\alpha} - M\right)I$$

positive definite.

We notice for the sub-expression in Eq. 15,

$$-\frac{1}{2}(\boldsymbol\theta - x)^\top \left(\frac{1}{\alpha}I + \frac{1}{2}\int_0^1 \nabla^2 U((1-s)\boldsymbol\theta + sx)\,ds\right)(\boldsymbol\theta - x) = -\frac{1}{2\alpha}\|\boldsymbol\theta - x\|^2 - \frac{1}{4}(\boldsymbol\theta - x)^\top\left(\int_0^1 \nabla^2 U((1-s)\boldsymbol\theta + sx)\,ds\right)(\boldsymbol\theta - x)$$
$$\leq -\frac{1}{2\alpha}\|\boldsymbol\theta - x\|^2 + \frac{M}{4}\|\boldsymbol\theta - x\|^2$$
$$= (-\frac{1}{2\alpha} + \frac{M}{4})\|\boldsymbol\theta - x\|^2$$

The second line is true because, $(\boldsymbol\theta - x)^\top \nabla^2 U(\boldsymbol\theta - x) \geq -M\|\boldsymbol\theta - x\|^2$ from Assumption 5.1.

We know from above, $\frac{\alpha}{2} < \frac{1}{M}$ for stability. Since the coefficient is negative, the expression is maximized when the distance $\|\boldsymbol\theta - x\|^2$ is minimized. Therefore,

$$-\frac{1}{2}(\boldsymbol\theta - x)^\top \left(\frac{1}{\alpha}I + \frac{1}{2}\int_0^1 \nabla^2 U((1-s)\boldsymbol\theta + sx)\,ds\right)(\boldsymbol\theta - x) \leq (\frac{M}{4} - \frac{1}{2\alpha})\delta(\boldsymbol\Theta)^2 \tag{16}$$

Second, for $x' \in \boldsymbol\Theta$ and $y' \in \boldsymbol\Theta_a$ (under Assumptions 5.1 and 5.3), we know that the following minimum exists and is well-defined: $\min_{\substack{x \in \boldsymbol\Theta\backslash\{x'\} \\ y \in \boldsymbol\Theta_a\backslash\{y'\}}} (x - y)^\top(x' - y')$

Thus when, $\dfrac{Z'_\gamma(\widetilde{\boldsymbol\theta})}{\int_y \sum_{x \in \boldsymbol\Theta} Z'_\gamma\left(\begin{bmatrix} x^\top \\ y^\top \end{bmatrix}\right)\pi\left(\begin{bmatrix} x^\top \\ y^\top \end{bmatrix}\right)dy} - 1 \geq 0$, we get,

$$\left| \frac{Z'_\gamma(\widetilde{\boldsymbol\theta})\pi(\widetilde{\boldsymbol\theta})}{\int_y \sum_{x \in \boldsymbol\Theta} Z'_\gamma\left(\begin{bmatrix} x^\top \\ y^\top \end{bmatrix}\right)\pi\left(\begin{bmatrix} x^\top \\ y^\top \end{bmatrix}\right)dy} - \pi(\widetilde{\boldsymbol\theta}) \right| = \pi(\widetilde{\boldsymbol\theta}) \left| \frac{Z'_\gamma(\widetilde{\boldsymbol\theta})}{\int_y \sum_{x \in \boldsymbol\Theta} Z'_\gamma\left(\begin{bmatrix} x^\top \\ y^\top \end{bmatrix}\right)\pi\left(\begin{bmatrix} x^\top \\ y^\top \end{bmatrix}\right)dy} - 1 \right|$$
$$\leq \pi(\widetilde{\boldsymbol\theta})\left(1 + \frac{1}{\sqrt{(2\pi\alpha_a)^d}}\int_{y \neq \theta_a}\sum_{x \neq \theta}\exp\left(\frac{1}{2}(U(x) - U(\boldsymbol\theta)) - \frac{1}{2}(\boldsymbol\theta - x)^\top\left(\frac{1}{\alpha}I + \frac{1}{2}\int_0^1 \nabla^2 U((1-s)\boldsymbol\theta + sx)\,ds\right)(\boldsymbol\theta - x)\right.\right.$$
$$\left.\left. -\frac{1}{2\alpha_a}\|y - \boldsymbol\theta_a\|^2 + \frac{4\eta}{8\eta^2}\|\boldsymbol\theta - \boldsymbol\theta_a\|^2 - \frac{1}{2\eta}(\boldsymbol\theta - \boldsymbol\theta_a)^\top(x - y)\right)dy - 1\right)$$
$$\leq \frac{\pi(\widetilde{\boldsymbol\theta})}{\sqrt{(2\pi\alpha_a)^d}}\exp\left((\frac{M}{4} - \frac{1}{2\alpha})\delta(\boldsymbol\Theta)^2 + \frac{1}{2\eta}\|\boldsymbol\theta - \boldsymbol\theta_a\|^2 - \frac{\vartheta(\boldsymbol\Theta, \boldsymbol\Theta_a)}{2\eta}\right) \cdot \left(\int_{y \neq \theta_a}\sum_{x \neq \theta}\exp\left(\frac{1}{2}U(x) - \frac{1}{2}U(\boldsymbol\theta) - \frac{1}{2\alpha_a}\|y - \boldsymbol\theta_a\|^2\right)dy\right)$$
$$\leq \pi(\widetilde{\boldsymbol\theta}) \cdot \exp\left((\frac{M}{4} - \frac{1}{2\alpha})\delta(\boldsymbol\Theta)^2 + \frac{1}{2\eta}\|\boldsymbol\theta - \boldsymbol\theta_a\|^2 - \frac{\vartheta(\boldsymbol\Theta, \boldsymbol\Theta_a)}{2\eta}\right) \cdot |\boldsymbol\Theta| \cdot \left(\exp\left(\frac{M\text{diam}(\boldsymbol\Theta)}{2}\right)\right)$$
$$= \pi(\widetilde{\boldsymbol\theta}) \cdot |\boldsymbol\Theta| \cdot \exp\left(\frac{M\text{diam}(\boldsymbol\Theta)}{2} + (\frac{M}{4} - \frac{1}{2\alpha})\delta(\boldsymbol\Theta)^2 + \frac{1}{2\eta}\|\boldsymbol\theta - \boldsymbol\theta_a\|^2 - \frac{\vartheta(\boldsymbol\Theta, \boldsymbol\Theta_a)}{2\eta}\right)$$

Third line is true because of Eq. 16. Fourth line is true because of Assumption 5.1.

Similarly, when $\dfrac{Z'_\gamma(\widetilde{\boldsymbol{\theta}})}{\int_y \sum_{x\in\Theta} Z'_\gamma\left(\begin{bmatrix} x^\top \\ y^\top \end{bmatrix}\right)\pi\left(\begin{bmatrix} x^\top \\ y^\top \end{bmatrix}\right)dy} - 1 < 0$, we get

$$\left| \frac{Z'_\gamma(\widetilde{\boldsymbol{\theta}})\pi(\widetilde{\boldsymbol{\theta}})}{\int_y \sum_{x\in\Theta} Z'_\gamma\left(\begin{bmatrix} x^\top \\ y^\top \end{bmatrix}\right)\pi\left(\begin{bmatrix} x^\top \\ y^\top \end{bmatrix}\right)dy} - \pi(\widetilde{\boldsymbol{\theta}}) \right|$$

$$= \pi(\widetilde{\boldsymbol{\theta}})\left(1 - \frac{1 + \frac{1}{\sqrt{(2\pi\alpha_a)^d}}\int_{y\neq\theta_a}\sum_{x\neq\theta}\exp\left(\frac{1}{2}(U(x)-U(\boldsymbol{\theta})) - \frac{1}{2}(\boldsymbol{\theta}-x)^\top\left(\frac{1}{\alpha}I + \frac{1}{2}\int_0^1\nabla^2 U((1-s)\theta+sx)ds\right)(\boldsymbol{\theta}-x) - \frac{1}{2\alpha_a}\|y-\boldsymbol{\theta}_a\|^2 + \frac{4\eta}{8\eta^2}\|\boldsymbol{\theta}-\boldsymbol{\theta}_a\|^2 - \frac{1}{2\eta}(\boldsymbol{\theta}-\boldsymbol{\theta}_a)^\top(x-y)\right)dy}{1 + \frac{1}{\sqrt{2\pi\alpha_a}}\int_p\frac{1}{\sqrt{\pi^d}}\exp(-p^2)\int_{q\neq p}\sum_r\frac{1}{Z}\exp(U(r))\sum_{s\neq r}\exp\left(\frac{1}{2}\left(U(s)-\frac{1}{2}U(r)\right) - \frac{1}{2}(r-s)^\top\left(\frac{1}{\alpha}I + \frac{1}{2}\int_0^1\nabla^2 U((1-l)r+ls)dl\right)(r-s) - \frac{1}{2\alpha_a}\|q-p\|^2 + \frac{4\eta}{8\eta^2}\|r-p\|^2 - \frac{1}{2\eta}(r-p)^\top(s-q)\right)dq\,dp}\right)$$

$$\leq \pi(\widetilde{\boldsymbol{\theta}})\left(1 - \frac{1}{1 + \frac{1}{\sqrt{2\pi\alpha_a}}\int_p\frac{1}{\sqrt{\pi^d}}\exp(-p^2)\int_{q\neq p}\exp\left(-\frac{1}{2\alpha_a}\|q-p\|^2\right)\sum_r\exp\left(\frac{4\eta}{8\eta^2}\|r-p\|^2\right)\frac{1}{2}\exp(U(r))\sum_{s\neq r}\exp\left(\frac{1}{2}(U(s)-U(r)) - \frac{1}{2}(r-s)^\top\left(\frac{1}{\alpha}I + \frac{1}{2}\int_0^1\nabla^2 U((1-l)r+ls)dl\right)(r-s) - \frac{1}{2\eta}(r-p)^\top(s-q)\right)dq\,dp}\right)$$

$$= \pi(\widetilde{\boldsymbol{\theta}})\left(\frac{\frac{1}{\sqrt{2\pi\alpha_a}}\int_p\frac{1}{\sqrt{\pi^d}}\exp(-p^2)\int_{q\neq p}\exp\left(-\frac{1}{2\alpha_a}\|q-p\|^2\right)\sum_r\exp\left(\frac{4\eta}{8\eta^2}\|r-p\|^2\right)\frac{1}{2}\exp(U(r))\sum_{s\neq r}\exp\left(\frac{1}{2}(U(s)-U(r)) - \frac{1}{2}(r-s)^\top\left(\frac{1}{\alpha}I + \frac{1}{2}\int_0^1\nabla^2 U((1-l)r+ls)dl\right)(r-s) - \frac{1}{2\eta}(r-p)^\top(s-q)\right)dq\,dp}{1 + \frac{1}{\sqrt{2\pi\alpha_a}}\int_p\frac{1}{\sqrt{\pi^d}}\exp(-p^2)\int_{q\neq p}\exp\left(-\frac{1}{2\alpha_a}\|q-p\|^2\right)\sum_r\exp\left(\frac{4\eta}{8\eta^2}\|r-p\|^2\right)\frac{1}{2}\exp(U(r))\sum_{s\neq r}\exp\left(\frac{1}{2}(U(s)-U(r)) - \frac{1}{2}(r-s)^\top\left(\frac{1}{\alpha}I + \frac{1}{2}\int_0^1\nabla^2 U((1-l)r+ls)dl\right)(r-s) - \frac{1}{2\eta}(r-p)^\top(s-q)\right)dq\,dp}\right)$$

$$\leq \frac{\pi(\widetilde{\boldsymbol{\theta}})}{\sqrt{2\pi\alpha_a}^d}\left(\int_p\frac{1}{\sqrt{\pi^d}}\exp(-p^2)\int_{q\neq p}\exp\left(-\frac{1}{2\alpha_a}\|q-p\|^2\right)\sum_r\exp\left(\frac{4\eta}{8\eta^2}\|r-p\|^2\right)\frac{1}{Z}\exp(U(r))\sum_{s\neq r}\exp\left(\frac{1}{2}(U(s)-U(r)) - \frac{1}{2}(r-s)^\top\left(\frac{1}{\alpha}I + \frac{1}{2}\int_0^1\nabla^2 U((1-l)r+ls)dl\right)(r-s) - \frac{1}{2\eta}(r-p)^\top(s-q)\right)\right)dq\,dp$$

$$\leq \frac{\pi(\widetilde{\boldsymbol{\theta}})}{\sqrt{(2\pi\alpha_a)^d}}\exp\left(\left(\frac{M}{4}-\frac{1}{2\alpha}\right)\delta(\boldsymbol{\Theta})^2\right)\left(\int_p\frac{1}{\sqrt{\pi^d}}\exp(-p^2)\right)\int_{q\neq p}\exp\left(-\frac{1}{2\alpha_a}\|q-p\|^2\right)\sum_r\exp\left(\frac{1}{2\eta}\|r-p\|^2\right)\frac{1}{Z}\exp(U(r))\sum_{s\neq r}\exp\left(\frac{1}{2}(U(s)-U(r)) - \frac{1}{2\eta}(r-p)^\top(s-q)\right)dq\,dp$$

$$\leq \frac{\pi(\widetilde{\boldsymbol{\theta}})}{\sqrt{2\pi\alpha_a}^d}\exp\left(\left(\frac{M}{4}-\frac{1}{2\alpha}\right)\delta(\boldsymbol{\Theta})^2 + \frac{\Delta(\boldsymbol{\Theta},\boldsymbol{\Theta}_a)^2 - \vartheta(\boldsymbol{\Theta},\boldsymbol{\Theta}_a)}{2\eta}\right)\left(\int_p\frac{1}{\sqrt{\pi^d}}\exp(-p^2)\int_{q\neq p}\exp\left(-\frac{1}{2\alpha_a}\|q-p\|^2\right)\sum_r\frac{1}{Z}(U(r))\sum_{s\neq r}\exp\left(\frac{1}{2}(U(s)-U(r))\right)dq\,dp\right)$$

$$\leq \frac{\pi(\widetilde{\boldsymbol{\theta}})}{\sqrt{2\pi\alpha_a}^d}\cdot|\boldsymbol{\Theta}|\cdot\left(\exp\left(\frac{M\mathrm{diam}(\boldsymbol{\Theta})}{2}\right)\right)\cdot\exp\left(\left(\frac{M}{4}-\frac{1}{2\alpha}\right)\delta(\boldsymbol{\Theta})^2 + \frac{\Delta(\boldsymbol{\Theta},\boldsymbol{\Theta}_a)^2 - \vartheta(\boldsymbol{\Theta},\boldsymbol{\Theta}_a)}{2\eta}\right)\left(\int_p\frac{1}{\sqrt{\pi^d}}\exp(-p^2)\int_{q\neq p}\exp\left(-\frac{1}{2\alpha_a}\|q-p\|^2\right)dq\,dp\right)$$

$$= \pi(\widetilde{\boldsymbol{\theta}})\cdot|\boldsymbol{\Theta}|\cdot\left(\exp\left(\frac{M\mathrm{diam}(\boldsymbol{\Theta})}{2}\right)\right)\cdot\exp\left(\left(\frac{M}{4}-\frac{1}{2\alpha}\right)\delta(\boldsymbol{\Theta})^2 + \frac{\Delta(\boldsymbol{\Theta},\boldsymbol{\Theta}_a)^2 - \vartheta(\boldsymbol{\Theta},\boldsymbol{\Theta}_a)}{2\eta}\right)\int_p\left(\frac{1}{\sqrt{\pi^d}}\exp(-p^2)\right)dp$$

$$= \pi(\widetilde{\boldsymbol{\theta}})\cdot|\boldsymbol{\Theta}|\cdot\exp\left(\frac{M\mathrm{diam}(\boldsymbol{\Theta})}{2} + \left(\frac{M}{4}-\frac{1}{2\alpha}\right)\delta(\boldsymbol{\Theta})^2 + \frac{1}{2\eta}\|\boldsymbol{\theta}-\boldsymbol{\theta}_a\|^2 - \frac{\vartheta(\boldsymbol{\Theta},\boldsymbol{\Theta}_a)}{2\eta}\right)$$

Therefore, the difference between $\pi_\gamma$ and $\tilde{\pi}$ can be bounded as follows

$$\|\pi_\gamma - \tilde{\pi}\|_1 \leq \int_{\theta_a}\sum_{\theta\in\boldsymbol{\Theta}}\pi(\widetilde{\boldsymbol{\theta}})\cdot|\boldsymbol{\Theta}|\cdot\left(\exp\left(\frac{M\mathrm{diam}(\boldsymbol{\Theta})}{2}\right)\right)\cdot\exp\left(\left(\frac{M}{4}-\frac{1}{2\alpha}\right)\delta(\boldsymbol{\Theta})^2 + \frac{\Delta(\boldsymbol{\Theta},\boldsymbol{\Theta}_a)^2 - \vartheta(\boldsymbol{\Theta},\boldsymbol{\Theta}_a)}{2\eta}\right)d\theta_a$$

$$\leq |\boldsymbol{\Theta}|\cdot\exp\left(\frac{M\mathrm{diam}(\boldsymbol{\Theta})}{2} + \left(\frac{M}{4}-\frac{1}{2\alpha}\right)\delta(\boldsymbol{\Theta})^2 + \frac{1}{2\eta}\|\boldsymbol{\theta}-\boldsymbol{\theta}_a\|^2 - \frac{\vartheta(\boldsymbol{\Theta},\boldsymbol{\Theta}_a)}{2\eta}\right)$$

### E.3 Proofs for EDULA

We start by establishing results for a more general case in which Assumption 5.3 is dropped. We establish that in this setting geometric rates of convergence exist. However, in this case proving that the stationary distribution is close to the target remains an open problem.

**Remark:** Assumption 5.3 from Section 5, restricts the auxiliary variable $\boldsymbol{\theta}_a$ to a compact subset $\boldsymbol{\Theta}_a \subset \mathbb{R}^d$. In the context of our theoretical analysis, this implies that the transition kernel for the auxiliary update, denoted as $q_{\alpha_a}(\boldsymbol{\theta}'_a|\tilde{\boldsymbol{\theta}})$ in Eq. 6, is formally treated as a Truncated Normal distribution supported on $\boldsymbol{\Theta}_a$:

$$q_{\alpha_a}^{\mathrm{trunc}}(\boldsymbol{\theta}'_a|\tilde{\boldsymbol{\theta}}) \propto \mathcal{N}\left(\boldsymbol{\theta}_a + \frac{\alpha_a}{2}\nabla_{\boldsymbol{\theta}_a}U_\eta(\tilde{\boldsymbol{\theta}}), \alpha_a I\right)\cdot\mathbb{I}(\boldsymbol{\theta}'_a\in\boldsymbol{\Theta}_a)$$

The normalization constant of this truncated distribution is strictly lower-bounded away from zero because $\boldsymbol{\Theta}_a$ is compact and the mean of the un-truncated normal is a continuous function of $\tilde{\boldsymbol{\theta}}$. Crucially, this ensures that the transition probabilities are bounded away from zero within the compact set, a necessary condition to establish the minorization condition required for geometric ergodicity. Note that while the mean of the proposal distribution may theoretically lie outside $\boldsymbol{\Theta}_a$ (e.g., if $\tilde{\boldsymbol{\theta}}$ is near the boundary), the truncation ensures

the sample remains within the valid domain, preserving the well-definedness of the chain without affecting the validity of the subsequent drift analysis. Thus this assumption is used solely to establish uniform ergodicity and non-asymptotic convergence bounds. In practice, our implementation (Algorithm 1) employs standard Gaussian Langevin dynamics without truncation. We note that the auxiliary step size $\alpha_a$ is typically small (e.g., $10^{-3}$ to $10^{-2}$), which, combined with the quadratic coupling $\frac{1}{2\eta}\|\boldsymbol{\theta} - \boldsymbol{\theta}_a\|^2$, naturally tethers $\boldsymbol{\theta}_a$ to the discrete state $\boldsymbol{\theta}$, precluding divergence and ensuring the chain remains stable in the absence of explicit truncation.

**Theorem E.1.** *Let Assumption 5.1 hold. Then for the Markov chain with transition operator $P$ as in Algorithm 1, the drift condition is satisfied as follows:*

$$PV(\tilde{\boldsymbol{\theta}}) \leq \alpha_a\, d + 2\left(1 - \frac{\alpha_a}{\eta}\right)^2 V(\tilde{\boldsymbol{\theta}}) + 2\frac{\alpha_a^2}{\eta^2} \sup_{\boldsymbol{\theta} \in \boldsymbol{\Theta}} \|\boldsymbol{\theta}\|^2.$$

*Proof.* We establish an explicit drift and minorization condition for the joint chain, which confirms the convergence rate. Note that

$$p((\boldsymbol{\theta}_a', \boldsymbol{\theta}') \mid (\boldsymbol{\theta}_a', \boldsymbol{\theta}')) = p(\boldsymbol{\theta}_a' \mid \boldsymbol{\theta}, \boldsymbol{\theta}_a) \cdot p(\boldsymbol{\theta}' \mid \boldsymbol{\theta}_a, \boldsymbol{\theta}).$$

Now,

$$p(\boldsymbol{\theta}_a' \mid \boldsymbol{\theta}, \boldsymbol{\theta}_a) = \frac{1}{(2\pi\alpha_a)^{d/2}} \exp\left\{-\frac{1}{2\alpha_a}\left\|\boldsymbol{\theta}_a' - \boldsymbol{\theta}_a\left(1 - \frac{\alpha_a}{\eta}\right) - \frac{\alpha_a}{\eta}\boldsymbol{\theta}\right\|^2\right\}$$

and

$$p(\boldsymbol{\theta}' \mid \boldsymbol{\theta}_a, \boldsymbol{\theta}) = \frac{\exp\left\{-\frac{1}{2\alpha}\left\|\boldsymbol{\theta}' - \boldsymbol{\theta} + \alpha\nabla U(\boldsymbol{\theta}) - \frac{\alpha}{\eta}(\boldsymbol{\theta} - \boldsymbol{\theta}_a)\right\|^2\right\}}{\sum_{\boldsymbol{x} \in \boldsymbol{\Theta}} \exp\left\{-\frac{1}{2\alpha}\left\|\boldsymbol{x} - \boldsymbol{\theta} + \alpha\nabla U(\boldsymbol{\theta}) - \frac{\alpha}{\eta}(\boldsymbol{\theta} - \boldsymbol{\theta}_a)\right\|^2\right\}}.$$

Therefore, our Markov transition operator $P$ is given as

$$P((\boldsymbol{\theta}_a, \boldsymbol{\theta}), A) = \int_A p((\boldsymbol{\theta}_a', \boldsymbol{\theta}') \mid (\boldsymbol{\theta}, \boldsymbol{\theta}_a))\, d\mu,$$

where $A \in \boldsymbol{\Theta} \times \mathbb{R}^d$ and $\mu$ is the product of the counting measure and Lebesgue measure.

We shall first establish a drift condition:

$$PV \leq \lambda V + b,$$

where we choose the Lyapunov function $V(\boldsymbol{x}_1, \boldsymbol{x}_2) = \|\boldsymbol{x}_1\|^2$ and some constant $b > 0$.

We note that

$$PV(\boldsymbol{\theta}_a, \boldsymbol{\theta}) = \frac{1}{(2\pi\alpha_a)^{d/2}} \sum_{\boldsymbol{\theta}' \in \boldsymbol{\Theta}} \int \|\boldsymbol{\theta}_a'\|^2 \exp\left\{-\frac{1}{2\alpha_a}\left\|\boldsymbol{\theta}_a' - \boldsymbol{\theta}_a\left(1 - \frac{\alpha_a}{\eta}\right) - \frac{\alpha_a}{\eta}\boldsymbol{\theta}\right\|^2\right\}$$

$$\cdot \frac{\exp\left\{-\frac{1}{2\alpha}\left\|\boldsymbol{\theta}' - \boldsymbol{\theta} + \alpha\nabla U(\boldsymbol{\theta}) - \frac{\alpha}{\eta}(\boldsymbol{\theta} - \boldsymbol{\theta}_a)\right\|^2\right\}}{\sum_{\boldsymbol{x} \in \boldsymbol{\Theta}} \exp\left\{-\frac{1}{2\alpha}\left\|\boldsymbol{x} - \boldsymbol{\theta} + \alpha\nabla U(\boldsymbol{\theta}) - \frac{\alpha}{\eta}(\boldsymbol{\theta} - \boldsymbol{\theta}_a)\right\|^2\right\}}\, d\boldsymbol{\theta}_a.$$

Using a change of variables, we have

$$PV(\boldsymbol{\theta_a}, \boldsymbol{\theta}) = \frac{1}{(2\pi\alpha_a)^{d/2}} \sum_{\boldsymbol{\theta}' \in \Theta} \int \left\| u + \boldsymbol{\theta_a}\left(1 - \frac{\alpha_a}{\eta}\right) + \frac{\alpha_a}{\eta}\boldsymbol{\theta} \right\|^2 \exp\left\{-\frac{1}{2\alpha_a}\|u\|^2\right\}$$

$$\cdot \frac{\exp\left\{-\frac{1}{2\alpha}\left\|\boldsymbol{\theta}' - \boldsymbol{\theta} + \alpha\nabla U(\boldsymbol{\theta}) - \frac{\alpha}{\eta}(\boldsymbol{\theta} - \boldsymbol{\theta_a})\right\|^2\right\}}{\sum_{\boldsymbol{x} \in \Theta} \exp\left\{-\frac{1}{2\alpha}\left\|\boldsymbol{x} - \boldsymbol{\theta} + \alpha\nabla U(\boldsymbol{\theta}) - \frac{\alpha}{\eta}(\boldsymbol{\theta} - \boldsymbol{\theta_a})\right\|^2\right\}} du$$

$$\leq \alpha_a d + 2\left(1 - \frac{\alpha_a}{\eta}\right)^2 \|\boldsymbol{\theta_a}\|^2 + 2\frac{\alpha_a^2}{\eta^2}\sup_{\boldsymbol{\theta} \in \Theta}\|\boldsymbol{\theta}\|^2.$$

Note that when $\lambda = 2\left(1 - \frac{\alpha_a}{\eta}\right)^2 < 1$, then this is a proper drift condition with $b = \alpha_a d + 2\frac{\alpha_a^2}{\eta^2}\sup_{\boldsymbol{\theta} \in \Theta}\|\boldsymbol{\theta}\|^2$.

**Theorem E.2.** *Under Assumption 5.1, the Markov chain with transition operator $P$ as in Algorithm 1 satisfies,*

$$P(\tilde{\boldsymbol{\theta}}, A) \geq \bar{\eta}\mu(A)$$

*where $\bar{\eta} > 0$ is defined in equation 18 and $\mu(\cdot)$ is the product of Lebesgue measure and counting measure and $\tilde{\boldsymbol{\theta}} \in C_\alpha$ as in equation 17 .*

*Proof.* We establish a minorization on the set,

$$C_{\alpha_a} = \left\{ x : V(x) \leq \frac{2\left(\alpha_a d + 2\frac{\alpha_a^2}{\eta^2}\sup_{\boldsymbol{\theta} \in \Theta}\|\boldsymbol{\theta}\|^2\right)}{\left(1 - \frac{\alpha_a}{\eta}\right)^2} \right\} \tag{17}$$

We define

$$\bar{\eta} = \frac{1}{(2\pi\alpha_a)^{d/2}} \exp\left\{-\frac{4}{\alpha_a}\frac{\left(\alpha_a d + 2\frac{\alpha_a^2}{\eta^2}\sup_{\boldsymbol{\theta} \in \Theta}\|\boldsymbol{\theta}\|^2\right)}{\left(1 - \frac{\alpha_a}{\eta}\right)^2}\right\} \cdot \frac{1}{|\Theta|}$$

$$\cdot \exp\left\{-\frac{1}{2\alpha}\left[\left((\alpha M + 1)^2 + \alpha M^2\right)\text{diam}(\Theta)^2 + (2(M + \alpha) + 2\alpha M)\|\nabla U(a)\|\text{diam}(\Theta) + \left(\alpha^2 + \alpha\right)\|\nabla U(a)\|^2\right.\right.$$

$$\left.\left. +2\frac{\alpha}{\eta}\left[(\alpha M + 1)^2\text{diam}(\Theta)^2 + 2(M + \alpha)\|\nabla U(a)\|\text{diam}(\Theta) + \alpha^2\|\nabla U(a)\|^2\right]^{1/2}\text{diam}(\Theta)\right]\right\} \tag{18}$$

We start with considering any $(\boldsymbol{\theta_1}, \boldsymbol{\theta_2}) \in C_\alpha$. Further, we also have $(\boldsymbol{\theta_a}, \boldsymbol{\theta}) \in C_{\alpha_a}$. Therefore

$$p((\boldsymbol{\theta_1}, \boldsymbol{\theta_2}) \mid (\boldsymbol{\theta_a}, \boldsymbol{\theta})) = \frac{1}{(2\pi\alpha_a)^{d/2}}\exp\left\{-\frac{1}{2\alpha_a}\left\|\boldsymbol{\theta_1} - \boldsymbol{\theta_a}\left(1 - \frac{\alpha_a}{\eta}\right) - \frac{\alpha_a}{\eta}\boldsymbol{\theta}\right\|^2\right\}$$

$$\cdot \frac{\exp\left\{-\frac{1}{2\alpha}\left\|\boldsymbol{\theta_2} - \boldsymbol{\theta} + \alpha\nabla U(\boldsymbol{\theta}) - \frac{\alpha}{\eta}(\boldsymbol{\theta} - \boldsymbol{\theta_a})\right\|^2\right\}}{\sum_{x \in \Theta}\exp\left\{-\frac{1}{2\alpha}\left\|x - \boldsymbol{\theta} + \alpha\nabla U(\boldsymbol{\theta}) - \frac{\alpha}{\eta}(\boldsymbol{\theta} - \boldsymbol{\theta_a})\right\|^2\right\}}.$$

For the first term, we note that

$$\left\|\boldsymbol{\theta_1} - \boldsymbol{\theta_a}\left(1 - \frac{\alpha_a}{\eta}\right) - \frac{\alpha_a}{\eta}\boldsymbol{\theta}\right\|^2 \leq 2\|\boldsymbol{\theta_1}\|^2 + 2\left\|\left(1 - \frac{\alpha_a}{\eta}\right)\boldsymbol{\theta_a} + \frac{\alpha_a}{\eta}\boldsymbol{\theta}\right\|^2$$

$$\leq 2\|\boldsymbol{\theta_1}\|^2 + 2\left(1 - \frac{\alpha_a}{\eta}\right)\|\boldsymbol{\theta_a}\|^2 + 2\frac{\alpha_a}{\eta}\|\boldsymbol{\theta}\|^2$$

$$\leq 8\frac{\left(\alpha_a d + 2\frac{\alpha_a^2}{\eta^2}\sup_{\boldsymbol{\theta} \in \Theta}\|\boldsymbol{\theta}\|^2\right)}{\left(1 - \frac{\alpha_a}{\eta}\right)^2}.$$

Therefore, the first term is greater than

$$\frac{1}{(2\pi\alpha_a)^{d/2}} \exp\left\{-\frac{1}{2\alpha_a}\left\|\boldsymbol{\theta_1} - \boldsymbol{\theta_a}\left(1 - \frac{\alpha_a}{\eta}\right) - \frac{\alpha_a}{\eta}\boldsymbol{\theta_2}\right\|^2\right\}$$

$$\geq \frac{1}{(2\pi\alpha_a)^{d/2}} \exp\left\{-\frac{4}{\alpha_a}\frac{\left(\alpha_a\, d + 2\frac{\alpha_a^2}{\eta^2}\sup_{\boldsymbol{\theta}\in\boldsymbol{\Theta}}\|\boldsymbol{\theta}\|^2\right)}{\left(1 - \frac{\alpha_a}{\eta}\right)^2}\right\}.$$

For the second term, note that

$$\frac{\exp\left\{-\frac{1}{2\alpha}\left\|\boldsymbol{\theta_2} - \boldsymbol{\theta} + \alpha\nabla U(\boldsymbol{\theta}) - \frac{\alpha}{\eta}(\boldsymbol{\theta} - \boldsymbol{\theta_a})\right\|^2\right\}}{\sum_{x\in\boldsymbol{\Theta}}\exp\left\{-\frac{1}{2\alpha}\left\|x - \boldsymbol{\theta} + \alpha\nabla U(\boldsymbol{\theta}) - \frac{\alpha}{\eta}(\boldsymbol{\theta} - \boldsymbol{\theta_a})\right\|^2\right\}} \geq \frac{1}{|\boldsymbol{\Theta}|}\exp\left\{-\frac{1}{2\alpha}\left\|\boldsymbol{\theta_2} - \boldsymbol{\theta} + \alpha\nabla U(\boldsymbol{\theta}) - \frac{\alpha}{\eta}(\boldsymbol{\theta} - \boldsymbol{\theta_a})\right\|^2\right\}.$$

For the numerator, one sees,

$$\left\|\boldsymbol{\theta_2} - \boldsymbol{\theta} + \alpha\nabla U(\boldsymbol{\theta}) - \frac{\alpha}{\eta}(\boldsymbol{\theta} - \boldsymbol{\theta_a})\right\|^2 \leq \|\boldsymbol{\theta_2} - \boldsymbol{\theta} + \alpha\nabla U(\boldsymbol{\theta})\|^2 + \frac{\alpha^2}{\eta^2}\|\boldsymbol{\theta} - \boldsymbol{\theta_a}\|^2$$

$$+ 2\frac{\alpha}{\eta}\|\boldsymbol{\theta_2} - \boldsymbol{\theta} + \alpha\nabla U(\boldsymbol{\theta})\|\,\|\boldsymbol{\theta} - \boldsymbol{\theta_a}\|.$$

For the first term, we have

$$\|\boldsymbol{\theta_2} - \boldsymbol{\theta} + \alpha\nabla U(\boldsymbol{\theta})\|^2 \leq \|\boldsymbol{\theta_2} - \boldsymbol{\theta}\|^2 + \alpha^2\|\nabla U(\boldsymbol{\theta})\|^2 + 2\,\alpha\,\|\boldsymbol{\theta_2} - \boldsymbol{\theta}\|\,\|\nabla U(\boldsymbol{\theta})\|.$$

Define $a = \operatorname{argmin}_{\boldsymbol{\theta}\in\boldsymbol{\Theta}}\|\nabla U(\boldsymbol{\theta})\|$. Therefore, the above expression is less than

$$\|\boldsymbol{\theta_2} - \boldsymbol{\theta} + \alpha\nabla U(\boldsymbol{\theta})\|^2 \leq \operatorname{diam}(\boldsymbol{\Theta})^2 + \alpha^2\left(M^2\operatorname{diam}(\boldsymbol{\Theta})^2 + \|\nabla U(a)\|^2 + 2\,M\operatorname{diam}(\boldsymbol{\Theta})\|\nabla U(a)\|\right)$$

$$+ 2\alpha\operatorname{diam}(\boldsymbol{\Theta})\left(M\operatorname{diam}(\boldsymbol{\Theta}) + \|\nabla U(a)\|\right)$$

$$\leq (\alpha\,M + 1)^2\operatorname{diam}(\boldsymbol{\Theta})^2 + 2\,(M + \alpha)\|\nabla U(a)\|\operatorname{diam}(\boldsymbol{\Theta}) + \alpha^2\|\nabla U(a)\|^2.$$

For the second term, we have

$$\alpha\|\nabla U(\boldsymbol{\theta})\|^2 \leq \alpha M^2\operatorname{diam}(\boldsymbol{\Theta})^2 + \alpha\|\nabla U(a)\|^2 + 2\alpha\,M\operatorname{diam}(\boldsymbol{\Theta})\|\nabla U(a)\|$$

and for the final term we have

$$2\frac{\alpha}{\eta}\|\boldsymbol{\theta_2} - \boldsymbol{\theta} + \alpha\nabla U(\boldsymbol{\theta})\|\,\|\boldsymbol{\theta} - \boldsymbol{\theta_a}\| \leq 2\frac{\alpha}{\eta}\left[(\alpha M + 1)^2\operatorname{diam}(\boldsymbol{\Theta})^2 + 2(M + \alpha)\|\nabla U(a)\|\operatorname{diam}(\boldsymbol{\Theta})\right.$$

$$\left. + \alpha^2\|\nabla U(a)\|^2\right]^{1/2}\operatorname{diam}(\boldsymbol{\Theta}). \tag{19}$$

Therefore we have

$$\frac{\exp\left\{-\frac{1}{2\alpha}\left\|\boldsymbol{\theta_2} - \boldsymbol{\theta} + \alpha\nabla U(\boldsymbol{\theta}) - \frac{\alpha}{\eta}(\boldsymbol{\theta} - \boldsymbol{\theta_a})\right\|^2\right\}}{\sum_{x\in\boldsymbol{\Theta}}\exp\left\{-\frac{1}{2\alpha}\left\|x - \boldsymbol{\theta} + \alpha\nabla U(\boldsymbol{\theta}) - \frac{\alpha}{\eta}(\boldsymbol{\theta} - \boldsymbol{\theta_a})\right\|^2\right\}}$$

$$\geq \frac{1}{|\boldsymbol{\Theta}|}\exp\left\{-\frac{1}{2\alpha}\left[\left((\alpha\,M + 1)^2 + \alpha\,M^2\right)\operatorname{diam}(\boldsymbol{\Theta})^2 + (2\,(M + \alpha) + 2\alpha M)\|\nabla U(a)\|\operatorname{diam}(\boldsymbol{\Theta}) + \left(\alpha^2 + \alpha\right)\|\nabla U(a)\|^2\right.\right.$$

$$\left.\left. + 2\frac{\alpha}{\eta}\left[(\alpha\,M + 1)^2\operatorname{diam}(\boldsymbol{\Theta})^2 + 2\,(M + \alpha)\|\nabla U(a)\|\operatorname{diam}(\boldsymbol{\Theta}) + \alpha^2\|\nabla U(a)\|^2\right]^{1/2}\operatorname{diam}(\boldsymbol{\Theta})\right]\right\}.$$

This finally gives $\tilde{\eta}$ as

$$\bar{\eta} = \frac{1}{(2\pi\alpha_a)^{d/2}}\exp\left\{-\frac{4}{\alpha_a}\frac{\left(\alpha_a\, d + 2\frac{\alpha_a^2}{\eta^2}\sup_{\boldsymbol{\theta}\in\boldsymbol{\Theta}}\|\boldsymbol{\theta}\|^2\right)}{\left(1 - \frac{\alpha_a}{\eta}\right)^2}\right\}$$

$$\cdot \frac{1}{|\boldsymbol{\Theta}|}\exp\left\{-\frac{1}{2\alpha}\left[\left((\alpha\,M + 1)^2 + \alpha\,M^2\right)\operatorname{diam}(\boldsymbol{\Theta})^2 + (2\,(M + \alpha) + 2\alpha M)\|\nabla U(a)\|\operatorname{diam}(\boldsymbol{\Theta}) + \left(\alpha^2 + \alpha\right)\|\nabla U(a)\|^2\right.\right.$$

$$\left.\left. + 2\frac{\alpha}{\eta}\left[(\alpha\,M + 1)^2\operatorname{diam}(\boldsymbol{\Theta})^2 + 2\,(M + \alpha)\|\nabla U(a)\|\operatorname{diam}(\boldsymbol{\Theta}) + \alpha^2\|\nabla U(a)\|^2\right]^{1/2}\operatorname{diam}(\boldsymbol{\Theta})\right]\right\}$$

with the reference measure $\mu(\cdot)$ is the product measure of the Lebesgue measure and the counting measure.

**Lemma E.3.** *The Markov chain defined by Algorithm 1 is irreducible, aperiodic and Harris recurrent.*

*Proof.* For any Borel measurable $A$ with $\lambda(A) > 0$ and any $\boldsymbol{\theta} \in \boldsymbol{\Theta}$, we have

$$\mathbb{P}\left(\boldsymbol{\theta}'_a \in A, \, \boldsymbol{\theta}' = \boldsymbol{\theta}^* \mid \boldsymbol{\theta}_a, \, \boldsymbol{\theta}\right) = \mathbb{P}\left(\boldsymbol{\theta}'_a \in A \mid \boldsymbol{\theta}_a, \, \theta\right) \mathbb{P}\left(\boldsymbol{\theta}' = \boldsymbol{\theta}^* \mid \boldsymbol{\theta}_a, \, \boldsymbol{\theta}\right).$$

Note that both the above terms are positive since the first distribution is Gaussian and the second term is positive by definition. We can similarly establish aperiodicity by noting that there is no partition of $\boldsymbol{\Theta} \times \mathbb{R}^d$ such that the previous probability is 1. Finally, due to the fact that the algorithm satisfies a drift condition, the Markov chain is Harris. □

We may leverage the above results to obtain a rate of convergence of the sampler using Ekvall & Jones (2021).

**Theorem E.4.** *The Markov chain has a stationary distribution dependent on $\gamma = (\alpha, \alpha_a)$, $\pi_\gamma$, and is $(M, \rho)$ geometrically ergodic with*

$$\|P^k(x, \cdot) - \pi_\gamma(\cdot)\|_{TV} \le M(x)\rho^k$$

*where*

$$M(x) = 2 + \frac{\tilde{b}}{1 - \tilde{\lambda}} + \tilde{V}(x)$$

*and*

$$\rho \le \max\left\{(1 - \bar{\eta})^r, \left(\frac{1 + 2\tilde{b} + \tilde{\lambda} + \tilde{\lambda}d}{1 + d}\right)^{1-r}\left(1 + 2\tilde{b} + 2\tilde{\lambda}d\right)^r\right\}$$

*for some free parameter $0 < r < 1$ and where $\bar{\eta}, b, \lambda$ are previously defined.*

*Proof.* The proof follows directly from Theorem E.1, Theorem E.2 and Lemma E.3 Ekvall & Jones (2021). □

**Theorem E.5.** *For any function $f : \mathbb{R}^p \to \mathbb{R}$ with $f^2(x) \le V(x)$ for all $x \in \mathbb{R}^p$ one has*

$$\sqrt{n}\left(\bar{f} - \mathbb{E}_{\pi_\gamma}f\right) \xrightarrow{d} N(0, \sigma_f^2)$$

*as $n \to \infty$, where $\sigma_f^2 \in [0, \infty)$. , where*

$$\bar{f} = \frac{1}{n}\sum_{i=1}^{n} f(X_i).$$

*Proof.* The proof follows from Theorem E.1 by noting that $PV \le \lambda V + b$ implies

$$P(V + 1) \le \lambda(V + 1) + (b + 1 - \lambda).$$

This implies a drift condition holds with $V : \mathbb{R}^d \to [1, \infty)$. Hence the result follows via Jones (2004). Note that $\sigma_f^2 = 0$ implies convergence to a Gaussian degenerate at 0. □

Define

$$\begin{aligned}
\bar{\eta}^* = \exp&\left\{-\frac{1}{\alpha_a}\operatorname{diam}(\boldsymbol{\Theta}_a)^2 - \frac{\alpha_a}{\eta^2}\Delta(\boldsymbol{\Theta}, \boldsymbol{\Theta}_a)^2\right\} \\
&\times \exp\left\{-\frac{1}{2\alpha}\left[\left((\alpha M + 1)^2 + \alpha M^2\right)\operatorname{diam}(\boldsymbol{\Theta})^2\right.\right. \\
&\qquad\qquad + (2(M + \alpha) + 2\alpha M)\|\nabla U(a)\|\operatorname{diam}(\boldsymbol{\Theta}) \\
&\qquad\qquad + \left(\alpha^2 + \alpha\right)\|\nabla U(a)\|^2 \\
&\qquad\qquad \left.\left. + 2\frac{\alpha}{\eta}\left[(\alpha M + 1)^2\operatorname{diam}(\boldsymbol{\Theta})^2 + 2(M + \alpha)\|\nabla U(a)\|\operatorname{diam}(\boldsymbol{\Theta}) + \alpha^2\|\nabla U(a)\|^2\right]^{1/2}\operatorname{diam}(\boldsymbol{\Theta})\right]\right\}
\end{aligned} \tag{20}$$

where $\operatorname{Leb}(\cdot)$ is the Lebesgue measure of a particular set.

**Lemma E.6.** *Under Assumptions 5.1 and 5.3, the Markov chain with transition operator $P$ as in Algorithm 1 satisfies,*

$$P((\boldsymbol{\theta}_a, \boldsymbol{\theta}), A) \geq \bar{\eta}^* \mu(A)$$

*where $\bar{\eta}^* > 0$ is as defined in equation 20 and $\mu(\cdot)$ is the uniform probability measure on $\Theta_a \times \Theta$.*

*Proof.* We consider the case where $\boldsymbol{\theta}_a$ is restricted to some compact subset of $\mathbb{R}^d$, which we refer to as $\boldsymbol{\Theta}_a$. In this case, note that the transition kernel changes to

$$p((\boldsymbol{\theta}_1, \boldsymbol{\theta}_2) \mid (\boldsymbol{\theta}_a, \boldsymbol{\theta})) = \frac{1}{\text{Leb}(\boldsymbol{\Theta}_a)} \exp\left\{ -\frac{1}{2\alpha_a} \left\| \boldsymbol{\theta}_1 - \boldsymbol{\theta}_a \left(1 - \frac{\alpha_a}{\eta}\right) - \frac{\alpha_a}{\eta} \boldsymbol{\theta} \right\|^2 \right\}$$

$$\times \frac{\exp\left\{ -\frac{1}{2\alpha} \left\| \boldsymbol{\theta}_2 - \boldsymbol{\theta} + \alpha \nabla U(\boldsymbol{\theta}) - \frac{\alpha}{\eta}(\boldsymbol{\theta} - \boldsymbol{\theta}_a) \right\|^2 \right\}}{\sum_{\boldsymbol{x} \in \boldsymbol{\Theta}} \exp\left\{ -\frac{1}{2\alpha} \left\| \boldsymbol{x} - \boldsymbol{\theta} + \alpha \nabla U(\boldsymbol{\theta}) - \frac{\alpha}{\eta}(\boldsymbol{\theta} - \boldsymbol{\theta}_a) \right\|^2 \right\}}.$$

The proof is similar to Theorem E.2. The key difference is that we can minorize on the entire set. Noting that

$$\left\| \boldsymbol{\theta}_1 - \boldsymbol{\theta}_a \left(1 - \frac{\alpha_a}{\eta}\right) - \frac{\alpha_a}{\eta} \boldsymbol{\theta} \right\|^2 \leq 2 \left\| \boldsymbol{\theta}_1 - \boldsymbol{\theta}_a \right\|^2 + 2 \frac{\alpha_a^2}{\eta^2} \left\| \boldsymbol{\theta}_a - \boldsymbol{\theta} \right\|^2$$

$$\leq 2 \operatorname{diam}(\boldsymbol{\Theta}_a)^2 + 2 \frac{\alpha_a^2}{\eta^2} \Delta(\boldsymbol{\Theta}, \boldsymbol{\Theta}_a)^2.$$

Using the same argument as Theorem E.2, we get a uniform minorization with

$$\bar{\eta}^* = \frac{1}{\text{Leb}(\boldsymbol{\Theta}_a)} \exp\left\{ -\frac{1}{\alpha_a} \operatorname{diam}(\boldsymbol{\Theta}_a)^2 - \frac{\alpha_a}{\eta^2} \Delta(\boldsymbol{\Theta}, \boldsymbol{\Theta}_a)^2 \right\}$$

$$\times \frac{1}{|\boldsymbol{\Theta}|} \exp\left\{ -\frac{1}{2\alpha} \left[ \left((\alpha M + 1)^2 + \alpha M^2\right) \operatorname{diam}(\boldsymbol{\Theta})^2 \right.\right.$$

$$+ (2(M + \alpha) + 2\alpha M) \|\nabla U(a)\| \operatorname{diam}(\boldsymbol{\Theta})$$

$$+ \left(\alpha^2 + \alpha\right) \|\nabla U(a)\|^2$$

$$\left.\left. + 2\frac{\alpha}{\eta} \left[ (\alpha M + 1)^2 \operatorname{diam}(\boldsymbol{\Theta})^2 + 2(M + \alpha)\|\nabla U(a)\| \operatorname{diam}(\boldsymbol{\Theta}) + \alpha^2 \|\nabla U(a)\|^2 \right]^{1/2} \operatorname{diam}(\boldsymbol{\Theta}) \right] \right\}.$$

with the reference measure

$$\mu(\cdot) = \frac{1}{\text{Leb}(\boldsymbol{\Theta}_a)|\boldsymbol{\Theta}|}$$

is the uniform measure.

*Proof of Theorem 5.5.* Using Lemma E.6 and Proposition 5.4, we further have

$$\|P^k(x, \cdot) - \tilde{\pi}\|_{TV} \leq (1 - \bar{\eta}^*)^k + |\boldsymbol{\Theta}| \cdot \exp\left( \frac{M \operatorname{diam}(\boldsymbol{\Theta})}{2} + (\frac{M}{4} - \frac{1}{2\alpha})\delta(\boldsymbol{\Theta})^2 + \frac{1}{2\eta} \|\boldsymbol{\theta} - \boldsymbol{\theta}_a\|^2 - \frac{\vartheta(\boldsymbol{\Theta}, \boldsymbol{\Theta}_a)}{2\eta} \right)$$

for all $x \in \mathbb{R}^d$ and $M(x), \rho$ is as defined in Theorem E.1 itself. Hence we are done.

### E.4  Proofs for EDMALA

**Proposition E.7.** *For EDMALA( EDLP with MH step, refer Algorithm 1) the drift condition is satisfied with drift function $V(x_1, x_2) = \|x_1\|^2$.*

*Proof.* The proof follows from Theorem E.1 by observing that

$$PV(\boldsymbol{\theta}_a, \boldsymbol{\theta}) \leq \int \|\boldsymbol{\theta}_{a_1}\|^2 q((\boldsymbol{\theta}_a, \boldsymbol{\theta}), (\boldsymbol{\theta}_{a_1}, \boldsymbol{\theta}_1)) d\boldsymbol{\theta}_{a_1} + 1$$

$$\leq \lambda V(\boldsymbol{\theta}_a, \boldsymbol{\theta}) + (b + 1).$$

**Lemma E.8.** *Under Assumptions 5.1, 5.2, 5.3, and $\alpha < \frac{2}{M}$, for Markov chain $P$ in Algorithm 1, we have for any $\widetilde{\boldsymbol{\theta}}, \widetilde{\boldsymbol{\theta}}' \in \widetilde{\boldsymbol{\Theta}}$,*

$$p(\widetilde{\boldsymbol{\theta}}|\widetilde{\boldsymbol{\theta}}') \geq \epsilon_\gamma \frac{\exp\left\{\frac{1}{2}U(\boldsymbol{\theta}')\right\}}{\sum_{x \in \boldsymbol{\Theta}} \exp\left(\frac{U(x)}{2}\right)} \cdot \frac{\exp\left\{-\frac{1}{2\alpha_a} diam(\boldsymbol{\Theta}_a)^2\right\}}{\Phi_{\alpha_a}(\boldsymbol{\Theta}_a)}$$

*, where*

$$\epsilon_\gamma = \exp\left\{ \begin{array}{l} -\left(\frac{M}{2} + \frac{1}{\alpha} - \frac{m}{4}\right) diam(\boldsymbol{\Theta})^2 - \frac{1}{2}\|\nabla U(a)\| \, diam(\boldsymbol{\Theta}) \\ -\left(\frac{3\alpha_a}{8\eta^2} + \frac{2}{\eta}\right) \Delta(\boldsymbol{\Theta}, \boldsymbol{\Theta}_a)^2 + \frac{\vartheta(\boldsymbol{\Theta}, \boldsymbol{\Theta}_a)}{\eta} \end{array} \right\},$$

*with $a \in \arg\min_{\boldsymbol{\theta} \in \boldsymbol{\Theta}} \|\nabla U(\boldsymbol{\theta})\|$*

*Proof.* We follow a similar minorization proof style as of Lemma 5.3 from Pynadath et al. (2024). We know from Equation 15,

$$Z_\gamma(\widetilde{\boldsymbol{\theta}}) = \frac{1}{\sqrt{(2\pi\alpha_a)^d}} \int_{\boldsymbol{\theta}'_a} \sum_{\boldsymbol{\theta}' \in \boldsymbol{\Theta}} \exp\left(\frac{1}{2}\left(-U(\boldsymbol{\theta}) + U(\boldsymbol{\theta}')\right) - (\boldsymbol{\theta} - \boldsymbol{\theta}')^\top \left(\frac{1}{2\alpha}I + \frac{1}{4}\int_0^1 \nabla^2 U((1-s)\boldsymbol{\theta} + s\boldsymbol{\theta}') \, ds\right)(\boldsymbol{\theta} - \boldsymbol{\theta}')\right.$$

$$\left. -\frac{1}{2\eta}(\boldsymbol{\theta} - \boldsymbol{\theta}_a)^\top(\boldsymbol{\theta}' - \boldsymbol{\theta}'_a) - \frac{1}{2\alpha_a}\|\boldsymbol{\theta}'_a - \boldsymbol{\theta}_a\|^2 + \frac{4\eta - \alpha_a}{8\eta^2}\|\boldsymbol{\theta} - \boldsymbol{\theta}_a\|^2\right).$$

Recall, from Assumption 5.1,U is $M$-gradient Lipschitz, we have

$$\frac{1}{\alpha}I + \frac{1}{2}\int_0^1 \nabla^2 U((1-s)\boldsymbol{\theta} + s\boldsymbol{\theta}') \, ds) \geq \left(\frac{1}{\alpha} - \frac{M}{2}\right)I$$

Since $\alpha < \frac{2}{M}$, the matrix $\left(\frac{1}{\alpha} - \frac{M}{2}\right)I$is positive definite.

This implies,

$$Z_\gamma(\widetilde{\boldsymbol{\theta}}) \leq \frac{1}{\sqrt{2\pi\alpha_a}^d} \exp\left(-\frac{U(\boldsymbol{\theta})}{2} - \frac{\alpha_a}{8\eta^2}\|\boldsymbol{\theta} - \boldsymbol{\theta}_a\|^2 + \frac{1}{2\eta}\|\boldsymbol{\theta} - \boldsymbol{\theta}_a\|^2\right) \sum_{x \in \boldsymbol{\Theta}} \exp\left(\frac{U(x)}{2}\right)$$

$$\int_y \sum_x \exp\left(-\frac{1}{2\alpha_a}\|y - \boldsymbol{\theta}_a\|^2 - \frac{1}{2\eta}(\boldsymbol{\theta} - \boldsymbol{\theta}_a)^\top(x - y)\right) dy$$

$$\leq \sum_{x \in \boldsymbol{\Theta}} \exp\left(\frac{U(x)}{2}\right) \exp\left(-\frac{U(\boldsymbol{\theta})}{2} + \frac{1}{2\eta}(\|\boldsymbol{\theta} - \boldsymbol{\theta}_a\|^2 - \vartheta(\boldsymbol{\Theta}, \boldsymbol{\Theta}_a))\right)$$

$$\leq \sum_{x \in \boldsymbol{\Theta}} \exp\left(\frac{U(x)}{2}\right) \exp\left(-\frac{U(\boldsymbol{\theta})}{2} + \frac{\Delta(\boldsymbol{\Theta}, \boldsymbol{\Theta}_a)^2 - \vartheta(\boldsymbol{\Theta}, \boldsymbol{\Theta}_a)}{2\eta}\right)$$

Since Assumption 5.2 holds true in this setting, we have an $m > 0$ such that for any $\boldsymbol{\theta} \in conv(\boldsymbol{\Theta})$

$$-\nabla^2 U(\boldsymbol{\theta}) \geq m\,I.$$

From this, one notes that

$$Z_\gamma(\widetilde{\boldsymbol{\theta}}) \geq \frac{1}{\sqrt{2\pi\alpha_a}^d} \exp\left\{-\frac{U(\boldsymbol{\theta})}{2} - \frac{\alpha_a}{8\eta^2}\|\boldsymbol{\theta} - \boldsymbol{\theta}_a\|^2 + \frac{1}{2\eta}\|\boldsymbol{\theta} - \boldsymbol{\theta}_a\|^2\right\} \exp\left\{-\frac{1}{2}\left(\frac{1}{\alpha} - \frac{m}{2}\right) \operatorname{diam}(\boldsymbol{\Theta})^2\right\}$$

$$\sum_{x\in\boldsymbol{\Theta}} \exp\left(\frac{U(x)}{2}\right) \int_y \sum_x \exp\left(-\frac{1}{2\alpha_a}\|y - \boldsymbol{\theta}_a\|^2 - \frac{1}{2\eta}(\boldsymbol{\theta} - \boldsymbol{\theta}_a)^\top(x - y)\right) dy$$

$$\geq \sum_{x\in\boldsymbol{\Theta}} \exp\left(\frac{U(x)}{2}\right) \exp\left\{-\frac{U(\boldsymbol{\theta})}{2} - \frac{\alpha_a}{8\eta^2}\|\boldsymbol{\theta} - \boldsymbol{\theta}_a\|^2 - \frac{1}{2}\left(\frac{1}{\alpha} - \frac{m}{2}\right) \operatorname{diam}(\boldsymbol{\Theta})^2 - \frac{1}{2\eta}\Delta(\boldsymbol{\Theta}, \boldsymbol{\Theta}_a)^2\right\}$$

$$\geq \sum_{x\in\boldsymbol{\Theta}} \exp\left(\frac{U(x)}{2}\right) \exp\left\{-\frac{U(\boldsymbol{\theta})}{2} - \frac{\alpha_a}{8\eta^2}\Delta(\boldsymbol{\Theta}, \boldsymbol{\Theta}_a)^2 - \frac{1}{2}\left(\frac{1}{\alpha} - \frac{m}{2}\right) \operatorname{diam}(\boldsymbol{\Theta})^2 - \frac{1}{2\eta}\Delta(\boldsymbol{\Theta}, \boldsymbol{\Theta}_a)^2\right\}$$

In other words,

$$\exp\left((-\frac{\alpha_a}{8\eta^2} - \frac{1}{2\eta})\Delta(\boldsymbol{\Theta}, \boldsymbol{\Theta}_a)^2 - \frac{1}{2}\left(\frac{1}{\alpha} - \frac{m}{2}\right) \operatorname{diam}(\boldsymbol{\Theta})^2\right) \leq \frac{Z_\gamma(\widetilde{\boldsymbol{\theta}})}{\sum_{x\in\boldsymbol{\Theta}} \exp\left(\frac{U(x)}{2}\right)\exp\left(-\frac{U(\boldsymbol{\theta})}{2}\right)} \leq \exp\left(\frac{\Delta(\boldsymbol{\Theta}, \boldsymbol{\Theta}_a)^2 - \vartheta(\boldsymbol{\Theta}, \boldsymbol{\Theta}_a)}{2\eta}\right)$$

Consequently,

$$\frac{\frac{Z_\gamma(\widetilde{\boldsymbol{\theta}})}{\sum_{x\in\boldsymbol{\Theta}} \exp\left(\frac{U(x)}{2}\right)\exp\left(-\frac{U(\boldsymbol{\theta})}{2}\right)}}{\frac{Z_\gamma(\widetilde{\boldsymbol{\theta}'})}{\sum_{x\in\boldsymbol{\Theta}} \exp\left(\frac{U(x)}{2}\right)\exp\left(-\frac{U(\boldsymbol{\theta}')}{2}\right)}} \geq \frac{\exp\left((-\frac{\alpha_a}{8\eta^2} - \frac{1}{2\eta})\Delta(\boldsymbol{\Theta}, \boldsymbol{\Theta}_a)^2 - \frac{(2-m\alpha)\operatorname{diam}(\boldsymbol{\Theta})^2}{4\alpha}\right)}{\exp\left(\frac{\Delta(\boldsymbol{\Theta}, \boldsymbol{\Theta}_a)^2 - \vartheta(\boldsymbol{\Theta}, \boldsymbol{\Theta}_a)}{2\eta}\right)}$$

This implies

$$\frac{Z_\gamma(\widetilde{\boldsymbol{\theta}})}{Z_\gamma(\widetilde{\boldsymbol{\theta}'})} \geq \exp\left(\frac{1}{2}(-U(\boldsymbol{\theta}) + U(\boldsymbol{\theta}'))\right) \frac{\exp\left((-\frac{\alpha_a}{8\eta^2} - \frac{1}{2\eta})\Delta(\boldsymbol{\Theta}, \boldsymbol{\Theta}_a)^2 - \frac{(2-m\alpha)\operatorname{diam}(\boldsymbol{\Theta})^2}{4\alpha}\right)}{\exp\left(\frac{\Delta(\boldsymbol{\Theta}, \boldsymbol{\Theta}_a)^2 - \vartheta(\boldsymbol{\Theta}, \boldsymbol{\Theta}_a)}{2\eta}\right)}$$

One notices from equation 9,

$$q_\gamma(\widetilde{\boldsymbol{\theta}'}|\widetilde{\boldsymbol{\theta}}) = \frac{Z_\gamma(\widetilde{\boldsymbol{\theta}})^{-1}}{\sqrt{(2\pi\alpha_a)^d}} \exp\left(\frac{1}{2}(-U(\boldsymbol{\theta}) + U(\boldsymbol{\theta}')) - (\boldsymbol{\theta} - \boldsymbol{\theta}')^\top\left(\frac{1}{2\alpha}I + \frac{1}{4}\int_0^1 \nabla^2 U((1-s)\boldsymbol{\theta} + s\boldsymbol{\theta}')ds\right)(\boldsymbol{\theta} - \boldsymbol{\theta}')\right.$$

$$\left. - \frac{1}{2\eta}(\boldsymbol{\theta} - \boldsymbol{\theta}_a)^\top(\boldsymbol{\theta}' - \boldsymbol{\theta}_a') - \frac{1}{2\alpha_a}\|\boldsymbol{\theta}_a' - \boldsymbol{\theta}_a\|^2 + \frac{4\eta - \alpha_a}{8\eta^2}\|\boldsymbol{\theta} - \boldsymbol{\theta}_a\|^2\right)$$

$$\geq \frac{Z_\gamma(\widetilde{\boldsymbol{\theta}})^{-1}}{\sqrt{(2\pi\alpha_a)^d}} \exp\left(\frac{1}{2}\left\langle\nabla U(\boldsymbol{\theta}), \boldsymbol{\theta}' - \boldsymbol{\theta}\right\rangle - \frac{1}{2\alpha}\|\boldsymbol{\theta} - \boldsymbol{\theta}'\|^2 - \frac{1}{2\eta}(\boldsymbol{\theta} - \boldsymbol{\theta}_a)^\top(\boldsymbol{\theta}' - \boldsymbol{\theta}_a')\right.$$

$$\left. - \frac{1}{2\alpha_a}\|\boldsymbol{\theta}_a' - \boldsymbol{\theta}_a\|^2 - \frac{\alpha_a}{8\eta^2}\|\boldsymbol{\theta} - \boldsymbol{\theta}_a\|^2\right)$$

We also note that

$$-\frac{1}{2}\left\langle\nabla U(\boldsymbol{\theta}), \boldsymbol{\theta}' - \boldsymbol{\theta}\right\rangle + \frac{1}{2\alpha}\|\boldsymbol{\theta} - \boldsymbol{\theta}'\|^2 = \frac{1}{2}\left\langle-\nabla U(\boldsymbol{\theta}) + \nabla U(a), \boldsymbol{\theta}' - \boldsymbol{\theta}\right\rangle + \frac{1}{2}\left\langle-\nabla U(a), \boldsymbol{\theta}' - \boldsymbol{\theta}\right\rangle + \frac{1}{2\alpha}\|\boldsymbol{\theta} - \boldsymbol{\theta}'\|^2$$

$$\leq \frac{1}{2}\left\langle-\nabla U(\boldsymbol{\theta}) + \nabla U(a), \boldsymbol{\theta}' - \boldsymbol{\theta}\right\rangle + \frac{1}{2}\left\langle-\nabla U(a), \boldsymbol{\theta}' - \boldsymbol{\theta}\right\rangle + \frac{1}{2\alpha}diam(\boldsymbol{\Theta})^2$$

$$\leq \frac{1}{2}\left\|-\nabla U(\boldsymbol{\theta}) + \nabla U(a)\right\|\|\boldsymbol{\theta}' - \boldsymbol{\theta}\| + \frac{1}{2}\left\|\nabla U(a)\right\|\|\boldsymbol{\theta}' - \boldsymbol{\theta}\| + \frac{1}{2\alpha}diam(\boldsymbol{\Theta})^2$$

$$\leq \frac{1}{2}\|-\nabla U(\boldsymbol{\theta}) + \nabla U(a)\|diam(\boldsymbol{\Theta}) + \frac{1}{2}\|\nabla U(a)\|diam(\boldsymbol{\Theta}) + \frac{1}{2\alpha}diam(\boldsymbol{\Theta})^2$$

$$\leq \frac{1}{2}(M\|a - \boldsymbol{\theta}\|)diam(\boldsymbol{\Theta}) + \frac{1}{2}\|\nabla U(a)\|diam(\boldsymbol{\Theta}) + \frac{1}{2\alpha}diam(\boldsymbol{\Theta})^2$$

$$\leq \left(\frac{1}{2}M + \frac{1}{2\alpha}\right) diam(\boldsymbol{\Theta})^2 + \frac{1}{2}\|\nabla U(a)\| diam(\boldsymbol{\Theta}).$$

Combining, we get

$$q_\gamma(\widetilde{\boldsymbol{\theta}'}|\widetilde{\boldsymbol{\theta}}) \geq \frac{Z_\gamma(\widetilde{\boldsymbol{\theta}})^{-1}}{\sqrt{(2\pi\alpha_a)^d}} \exp\left\{ (-\frac{M}{2} - \frac{1}{2\alpha})\mathrm{diam}(\boldsymbol{\Theta})^2 - \frac{1}{2}\|\nabla U(a)\|\mathrm{diam}(\boldsymbol{\Theta}) - \frac{1}{2\eta}(\boldsymbol{\theta} - \boldsymbol{\theta}_a)^\top(\boldsymbol{\theta}' - \boldsymbol{\theta}_a') - \frac{1}{2\alpha_a}\|\boldsymbol{\theta}_a' - \boldsymbol{\theta}_a\|^2 - \frac{\alpha_a}{8\eta^2}\|\boldsymbol{\theta} - \boldsymbol{\theta}_a\|^2 \right\}$$

$$\geq \frac{\frac{1}{\sqrt{(2\pi\alpha_a)^d}} \exp\left\{ (-\frac{M}{2} - \frac{1}{2\alpha})\mathrm{diam}(\boldsymbol{\Theta})^2 - \frac{1}{2}\|\nabla U(a)\|\mathrm{diam}(\boldsymbol{\Theta}) - \frac{1}{2\eta}(\boldsymbol{\theta} - \boldsymbol{\theta}_a)^\top(\boldsymbol{\theta}' - \boldsymbol{\theta}_a') - \frac{1}{2\alpha_a}\|\boldsymbol{\theta}_a' - \boldsymbol{\theta}_a\|^2 - \frac{\alpha_a}{8\eta^2}\|\boldsymbol{\theta} - \boldsymbol{\theta}_a\|^2 \right\}}{\sum_{x\in\boldsymbol{\Theta}} \exp\left(\frac{U(x)}{2}\right) \exp\left(-\frac{U(\boldsymbol{\theta})}{2} + \frac{\Delta(\boldsymbol{\Theta},\boldsymbol{\Theta}_a)^2 - \vartheta(\boldsymbol{\Theta},\boldsymbol{\Theta}_a)}{2\eta}\right)}$$

$$\geq \frac{\exp\left\{ -\frac{1}{2\alpha_a}\mathrm{diam}(\boldsymbol{\Theta}_a)^2 \right\}}{\Phi_{\alpha_a}(\boldsymbol{\Theta}_a)} \frac{\exp\left\{ (-\frac{M}{2} - \frac{1}{2\alpha})\mathrm{diam}(\boldsymbol{\Theta})^2 - \frac{1}{2}\|\nabla U(a)\|\mathrm{diam}(\boldsymbol{\Theta}) + (-\frac{1}{2\eta} - \frac{\alpha_a}{8\eta^2})\Delta(\boldsymbol{\Theta},\boldsymbol{\Theta}_a)^2 \right\}}{\sum_{x\in\boldsymbol{\Theta}} \exp\left(\frac{U(x)}{2}\right) \exp\left(-\frac{U(\boldsymbol{\theta})}{2} + \frac{\Delta(\boldsymbol{\Theta},\boldsymbol{\Theta}_a)^2 - \vartheta(\boldsymbol{\Theta},\boldsymbol{\Theta}_a)}{2\eta}\right)}$$

Acceptance Ratio,

$$\rho(\widetilde{\boldsymbol{\theta}'} \mid \widetilde{\boldsymbol{\theta}}) = \left( \frac{\pi(\widetilde{\boldsymbol{\theta}'})q_\gamma(\widetilde{\boldsymbol{\theta}} \mid \widetilde{\boldsymbol{\theta}'})}{\pi(\widetilde{\boldsymbol{\theta}})q_\gamma(\widetilde{\boldsymbol{\theta}'} \mid \widetilde{\boldsymbol{\theta}})} \right)$$

$$= \exp\left\{ U(\boldsymbol{\theta}') - U(\boldsymbol{\theta}) + \frac{1}{2\eta}(\|\boldsymbol{\theta} - \boldsymbol{\theta}_a\|^2 - \|\boldsymbol{\theta}' - \boldsymbol{\theta}_a'\|^2) \right\} \frac{\tilde{Z}}{\tilde{Z}}.$$

$$\exp\left\{ U(\boldsymbol{\theta}) - U(\boldsymbol{\theta}') - \frac{1}{2\eta}(\|\boldsymbol{\theta} - \boldsymbol{\theta}_a\|^2 - \|\boldsymbol{\theta}' - \boldsymbol{\theta}_a'\|^2) - \frac{\alpha_a}{8\eta^2}(\|\boldsymbol{\theta}' - \boldsymbol{\theta}_a'\|^2 - \|\boldsymbol{\theta} - \boldsymbol{\theta}_a\|^2) \right\} \frac{Z_\gamma(\widetilde{\boldsymbol{\theta}})}{Z_\gamma(\widetilde{\boldsymbol{\theta}'})}$$

$$= \exp\left\{ -\frac{\alpha_a}{8\eta^2}(\|\boldsymbol{\theta}' - \boldsymbol{\theta}_a'\|^2 - \|\boldsymbol{\theta} - \boldsymbol{\theta}_a\|^2) \right\} \frac{Z_\gamma(\widetilde{\boldsymbol{\theta}})}{Z_\gamma(\widetilde{\boldsymbol{\theta}'})}$$

where $\tilde{Z}$ is the normalizing constant for $\pi(\widetilde{\boldsymbol{\theta}})$.

with Acceptance Probability

$$\mathcal{A}(\widetilde{\boldsymbol{\theta}'} \mid \widetilde{\boldsymbol{\theta}}) = \left( \rho(\widetilde{\boldsymbol{\theta}'} \mid \widetilde{\boldsymbol{\theta}}) \wedge 1 \right)$$

and consider the transition kernel as

$$p(\widetilde{\boldsymbol{\theta}'} \mid \widetilde{\boldsymbol{\theta}}) = \left( \mathcal{A}(\widetilde{\boldsymbol{\theta}'} \mid \widetilde{\boldsymbol{\theta}}) \right) q_\gamma(\widetilde{\boldsymbol{\theta}'} \mid \widetilde{\boldsymbol{\theta}}) + \left( 1 - L(\widetilde{\boldsymbol{\theta}}) \right) \delta_{\widetilde{\boldsymbol{\theta}}}(\widetilde{\boldsymbol{\theta}'})$$

where $\delta_{\widetilde{\boldsymbol{\theta}}}(\widetilde{\boldsymbol{\theta}'})$ is the Kronecker delta function and $L(\widetilde{\boldsymbol{\theta}})$ is the total acceptance probability from the point $\widetilde{\boldsymbol{\theta}}$
with

$$L(\widetilde{\boldsymbol{\theta}}) = \int_{\boldsymbol{\theta}_a' \in \boldsymbol{\Theta}_a} \sum_{\boldsymbol{\theta}' \in \boldsymbol{\Theta}} \left( \rho([\boldsymbol{\theta}'^T, \boldsymbol{\theta}_a'^T]^T \mid \widetilde{\boldsymbol{\theta}}) \wedge 1 \right) q_\gamma([\boldsymbol{\theta}'^T, \boldsymbol{\theta}_a'^T]^T | \widetilde{\boldsymbol{\theta}}) \quad d\boldsymbol{\theta}_a'$$

We note

$$p(\widetilde{\boldsymbol{\theta}'} \mid \widetilde{\boldsymbol{\theta}}) = \left( \mathcal{A}(\widetilde{\boldsymbol{\theta}'} \mid \widetilde{\boldsymbol{\theta}}) \right) q_\gamma(\widetilde{\boldsymbol{\theta}'} \mid \widetilde{\boldsymbol{\theta}}) + \left( 1 - L(\widetilde{\boldsymbol{\theta}}) \right) \delta_{\widetilde{\boldsymbol{\theta}}}(\widetilde{\boldsymbol{\theta}'})$$

$$\geq \left( \mathcal{A}(\widetilde{\boldsymbol{\theta}'} \mid \widetilde{\boldsymbol{\theta}}) \right) q_\gamma(\widetilde{\boldsymbol{\theta}'} \mid \widetilde{\boldsymbol{\theta}})$$

$$= \left( \rho(\widetilde{\boldsymbol{\theta}'} \mid \widetilde{\boldsymbol{\theta}}) \wedge 1 \right) q_\gamma(\widetilde{\boldsymbol{\theta}'} \mid \widetilde{\boldsymbol{\theta}})$$

$$= \exp\left\{ -\frac{\alpha_a}{8\eta^2} \left( \|\boldsymbol{\theta}' - \boldsymbol{\theta}'_a\|^2 - \|\boldsymbol{\theta} - \boldsymbol{\theta}_a\|^2 \right) \right\} \frac{Z_\gamma(\widetilde{\boldsymbol{\theta}})}{Z_\gamma(\widetilde{\boldsymbol{\theta}'})} q_\gamma(\widetilde{\boldsymbol{\theta}'} \mid \widetilde{\boldsymbol{\theta}})$$

$$\geq \exp\left\{ -\frac{\alpha_a}{8\eta^2} \|\boldsymbol{\theta}' - \boldsymbol{\theta}'_a\|^2 \right\} \frac{Z_\gamma(\widetilde{\boldsymbol{\theta}})}{Z_\gamma(\widetilde{\boldsymbol{\theta}'})} q_\gamma(\widetilde{\boldsymbol{\theta}'} \mid \widetilde{\boldsymbol{\theta}})$$

$$\geq \exp\left\{ -\frac{\alpha_a}{8\eta^2} \Delta(\boldsymbol{\Theta}, \boldsymbol{\Theta}_a)^2 + \frac{1}{2}(-U(\boldsymbol{\theta}) + U(\boldsymbol{\theta}')) \right\} \frac{\exp\left( -\frac{\alpha_a}{8\eta^2} - \frac{1}{2\eta} \right) \Delta(\boldsymbol{\Theta}, \boldsymbol{\Theta}_a)^2 - \frac{(2-m\alpha)\mathrm{diam}(\boldsymbol{\Theta})^2}{4\alpha}}{\exp\left( \frac{\Delta(\boldsymbol{\Theta}, \boldsymbol{\Theta}_a)^2 - \vartheta(\boldsymbol{\Theta}, \boldsymbol{\Theta}_a)}{2\eta} \right)} q_\gamma(\widetilde{\boldsymbol{\theta}'} \mid \widetilde{\boldsymbol{\theta}})$$

$$\geq \exp\left\{ -\frac{\alpha_a}{8\eta^2} \Delta(\boldsymbol{\Theta}, \boldsymbol{\Theta}_a)^2 + \frac{1}{2}(-U(\boldsymbol{\theta}) + U(\boldsymbol{\theta}')) \right\} \frac{\exp\left( -\frac{\alpha_a}{8\eta^2} - \frac{1}{2\eta} \right) \Delta(\boldsymbol{\Theta}, \boldsymbol{\Theta}_a)^2 - \frac{(2-m\alpha)\mathrm{diam}(\boldsymbol{\Theta})^2}{4\alpha}}{\exp\left( \frac{\Delta(\boldsymbol{\Theta}, \boldsymbol{\Theta}_a)^2 - \vartheta(\boldsymbol{\Theta}, \boldsymbol{\Theta}_a)}{2\eta} \right)}$$

$$\cdot \frac{\exp\left\{ -\frac{1}{2\alpha_a} \mathrm{diam}(\boldsymbol{\Theta}_a)^2 \right\}}{\Phi_{\alpha_a}(\boldsymbol{\Theta}_a)} \frac{\exp\left\{ (-\frac{M}{2} - \frac{1}{2\alpha})\mathrm{diam}(\boldsymbol{\Theta})^2 - \frac{1}{2}\|\nabla U(a)\|\mathrm{diam}(\boldsymbol{\Theta}) + \left( -\frac{1}{2\eta} - \frac{\alpha_a}{8\eta^2} \right) \Delta(\boldsymbol{\Theta}, \boldsymbol{\Theta}_a)^2 \right\}}{\sum_{x \in \boldsymbol{\Theta}} \exp\left( \frac{U(x)}{2} \right) \exp\left( -\frac{U(\boldsymbol{\theta})}{2} + \frac{\Delta(\boldsymbol{\Theta}, \boldsymbol{\Theta}_a)^2 - \vartheta(\boldsymbol{\Theta}, \boldsymbol{\Theta}_a)}{2\eta} \right)}$$

$$= \frac{\exp\left\{ -\frac{1}{2\alpha_a} \mathrm{diam}(\boldsymbol{\Theta}_a)^2 \right\}}{\Phi_{\alpha_a}(\boldsymbol{\Theta}_a)} \frac{\exp\left\{ \frac{1}{2} U(\boldsymbol{\theta}') \right\}}{\sum_{x \in \boldsymbol{\Theta}} \exp\left( \frac{U(x)}{2} \right)} \exp\left\{ \left( -\frac{3\alpha_a}{8\eta^2} - \frac{2}{\eta} \right) \Delta(\boldsymbol{\Theta}, \boldsymbol{\Theta}_a)^2 + \frac{\vartheta(\boldsymbol{\Theta}, \boldsymbol{\Theta}_a)}{\eta} \right\}$$

$$\cdot \exp\left\{ (-\frac{M}{2} - \frac{1}{\alpha} + \frac{m}{4})\mathrm{diam}(\boldsymbol{\Theta})^2 - \frac{1}{2}\|\nabla U(a)\|\mathrm{diam}(\boldsymbol{\Theta}) \right\}$$

$$= \epsilon_\gamma \frac{\exp\left\{ \frac{1}{2} U(\boldsymbol{\theta}') \right\}}{\sum_{x \in \boldsymbol{\Theta}} \exp\left( \frac{U(x)}{2} \right)} \frac{\exp\left\{ -\frac{1}{2\alpha_a} \mathrm{diam}(\boldsymbol{\Theta}_a)^2 \right\}}{\Phi_{\alpha_a}(\boldsymbol{\Theta}_a)}$$

Note: Proof of Theorem 5.6 follows from using Lemma $E.8$.

# F   Additional Experimental Results

## F.1   4D Joint Bernoulli

To provide additional insights into the functionality of EDLP samplers, we explore their behavior on the 4D Joint Bernoulli Distribution, which serves as the simplest low-dimensional case among our experiments. This aids in visualizing and understanding the sampling process.

### F.1.1   Target Distribution

The following represents the probability mass function (PMF) for the 4D Joint Bernoulli Distribution used in our test case. The distribution has 16 states with the corresponding probabilities:

### F.1.2   Flatness Diagnostics

Under the experimental setup outlined in Section 6, we present the true Eigenspectrum of the Hessian, derived from the discrete samples collected for EDULA, EDMALA, DULA, and DMALA (Figure 11). This visualization is inspired by Section 6.3 of (Li & Zhang, 2024), where diagonal Fisher information matrix approximation was used to plot the Eigenvalues. The alignment of the Eigenvalues closer to 0 indicates that the sampled data corresponds to a flatter curvature of the energy function.

EDMALA and EDULA, specifically designed with entropy-aware flatness optimization, exhibit eigenvalue distributions that are notably tighter and more concentrated around zero compared to their non-entropic counterparts, DMALA and DULA.

$$P_{\boldsymbol{\Theta}}(\boldsymbol{\theta}) = \begin{cases} 0.07688 & \text{if } \boldsymbol{\theta} = 0000, \\ 0.04725 & \text{if } \boldsymbol{\theta} = 0001, \\ 0.12500 & \text{if } \boldsymbol{\theta} = 0010, \\ 0.01667 & \text{if } \boldsymbol{\theta} = 0011, \\ 0.08688 & \text{if } \boldsymbol{\theta} = 0100, \\ 0.07688 & \text{if } \boldsymbol{\theta} = 0101, \\ 0.07688 & \text{if } \boldsymbol{\theta} = 0110, \\ 0.16756 & \text{if } \boldsymbol{\theta} = 0111, \\ 0.04725 & \text{if } \boldsymbol{\theta} = 1000, \\ 0.05825 & \text{if } \boldsymbol{\theta} = 1001, \\ 0.01667 & \text{if } \boldsymbol{\theta} = 1010, \\ 0.04725 & \text{if } \boldsymbol{\theta} = 1011, \\ 0.07688 & \text{if } \boldsymbol{\theta} = 1100, \\ 0.04725 & \text{if } \boldsymbol{\theta} = 1101, \\ 0.01900 & \text{if } \boldsymbol{\theta} = 1110, \\ 0.01335 & \text{if } \boldsymbol{\theta} = 1111. \end{cases}$$

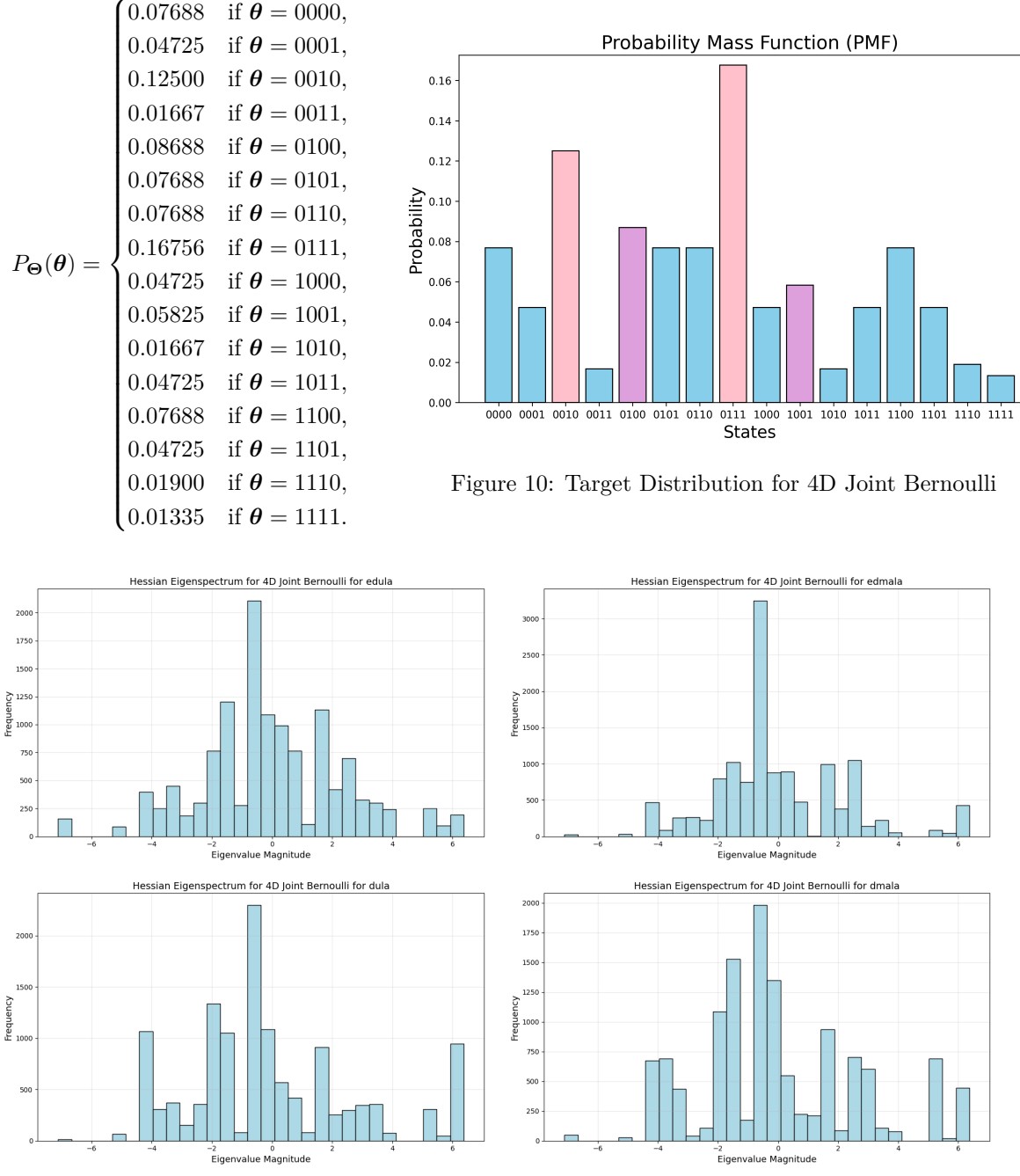

Figure 10: Target Distribution for 4D Joint Bernoulli

Figure 11: Eigenspectra of EDULA, EDMALA, DULA, and DMALA's performance on a Bernoulli distribution.

Quantitatively, EDULA demonstrates a lower spectral dispersion, evidenced by a lower standard deviation (std = 2.401) and narrower interquartile range (IQR = 3.031), relative to DULA (std = 2.832, IQR = 3.466). Similarly, EDMALA outperforms DMALA in terms of spectral concentration, achieving a standard deviation of 2.197 and IQR of 2.747, compared to DMALA's standard deviation of 2.700 and IQR of 3.224. Furthermore, visual inspection corroborates these quantitative findings; EDMALA and EDULA feature fewer extreme eigenvalues and outliers, reflecting biasing into sampling from flatter regions. Collectively, these results affirm that our entropy-guided methods (EDMALA, EDULA) effectively traverse flatter, aligning well with their intended design objectives.

### F.1.3 Frequency Counts

We report the actual frequency counts for each sampler in Section 6. DMALA and EDMALA exhibit the MH Acceptance Probabilities of $1.000 \pm 0.000$ and $0.176 \pm 0.190$ respectively. Table 5 the raw count data for $N = 1000$ (based on the 3200 total samples collected across 4 chains after 200 iterations of burn-in per chain). The raw probability mass in flat modes(F) and sharp modes(S) highlights EDLP's flat mode seeking behavior in finite time.

Table 5: Frequency counts for the 4D Joint Bernoulli distribution after $N = 1000$ iterations.

| State | EDULA | EDMALA | DULA | DMALA |
|---|---|---|---|---|
| 0000 | 266 | 276 | 208 | 479 |
| 0001 | 231 | 117 | 109 | 45 |
| **0010(S)** | 250 | 87 | 344 | 691 |
| 0011 | 157 | 24 | 11 | 49 |
| **0100(F)** | 353 | 520 | 214 | 165 |
| 0101 | 301 | 220 | 339 | 107 |
| 0110 | 219 | 137 | 260 | 391 |
| **0111(S)** | 192 | 426 | 873 | 444 |
| 1000 | 189 | 74 | 162 | 89 |
| **1001(F)** | 157 | 725 | 64 | 101 |
| 1010 | 109 | 3 | 63 | 213 |
| 1011 | 96 | 43 | 57 | 18 |
| 1100 | 287 | 233 | 234 | 264 |
| 1101 | 187 | 262 | 162 | 42 |
| 1110 | 120 | 21 | 39 | 73 |
| 1111 | 86 | 32 | 61 | 29 |

### F.1.4 Asymptotic Behavior

We retain the same setup as in Section 6. We now run our samplers for longer to gauge the asymptotic behavior. From Table 6, flatness bias vanishes in EDLP as more samples are collected. This shows that EDLP stays faithful to $\pi(\boldsymbol{\theta})$ in the asymptotic limit.

Table 6: Comparison of samplers using TVD and KL divergence at different iteration counts.

| Sampler | Iterations ($n$) | TVD | KL Divergence |
|---|---|---|---|
| DULA | 10k | 0.0961 | 0.0438 |
| | 100k | 0.0490 | 0.0061 |
| EDULA | 10k | 0.1442 | 0.0868 |
| | 100k | 0.1408 | 0.0763 |
| DMALA | 10k | 0.0440 | 0.0079 |
| | 100k | 0.0215 | 0.0012 |
| EDMALA | 10k | 0.0734 | 0.0236 |
| | 100k | 0.0227 | 0.0016 |

### F.1.5 Finite Time Behavior

To highlight EDLP's mode scouting behavior as a function of time, we adopt a similar sampling analysis as EMCMC(Li & Zhang, 2024), Appendix E.1. We aim to show that, EDLP shows benefit when DLP does not equilibriate. Specifically we extend our 4D Bernoulli Example(Section 6) to include a sharp mode(0111) and a flat mode(1001)( See Figure 12). Also, to ensure that *fight is fair*, we assign both our sharp mode and flat mode the same probability mass.

$$
P_{\Theta}(\boldsymbol{\theta}) = \begin{cases}
0.040 & \text{if } \boldsymbol{\theta} = 0000, \\
0.100 & \text{if } \boldsymbol{\theta} = 0001, \\
0.050 & \text{if } \boldsymbol{\theta} = 0010, \\
0.025 & \text{if } \boldsymbol{\theta} = 0011, \\
0.045 & \text{if } \boldsymbol{\theta} = 0100, \\
0.025 & \text{if } \boldsymbol{\theta} = 0101, \\
0.030 & \text{if } \boldsymbol{\theta} = 0110, \\
0.110 & \text{if } \boldsymbol{\theta} = 0111, \\
0.100 & \text{if } \boldsymbol{\theta} = 1000, \\
0.110 & \text{if } \boldsymbol{\theta} = 1001, \\
0.040 & \text{if } \boldsymbol{\theta} = 1010, \\
0.100 & \text{if } \boldsymbol{\theta} = 1011, \\
0.050 & \text{if } \boldsymbol{\theta} = 1100, \\
0.100 & \text{if } \boldsymbol{\theta} = 1101, \\
0.050 & \text{if } \boldsymbol{\theta} = 1110, \\
0.025 & \text{if } \boldsymbol{\theta} = 1111.
\end{cases}
$$

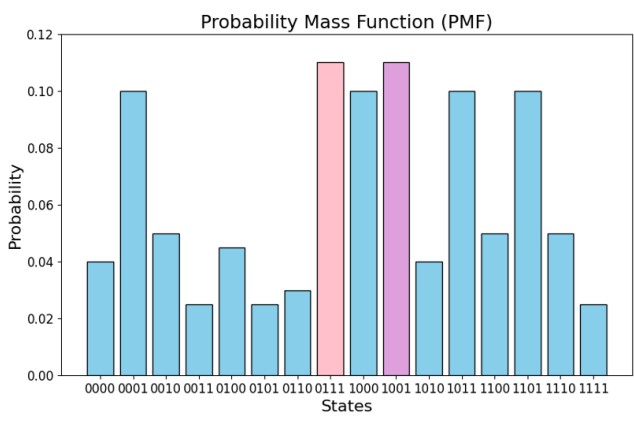

Figure 12: Target Distribution for 4D Joint Bernoulli

For each sampler, we track what state each chain visits using transition heatmaps, with no thinning, no burnin for a total of 100 iterations to model finite-time behavior. We run this simulation under sharp mode and flat mode initialization.

Figure 13 illustrates that DLP samplers, when initialized in the sharp mode, exhibit minimal to no movement. In contrast, EDLP samplers swiftly transition to the flat mode and achieve stabilization.

Similarly, from Figure 14 DLP samplers, quickly escape the flat mode and find sharp modes. EDLP samplers, on the other hand, choose to remain at the flat mode.

The results clearly demonstrates that EDLP is successfully able to find flat modes under a computing budget irrespective of initialization.

## F.2 TSP

### F.2.1 Gradient Computation for Binary Matrix Representation

To enable gradient-based exploration of the discrete TSP landscape, we represent the tour as a binary matrix $\boldsymbol{X} \in \{0,1\}^{n \times n}$. We define a continuous relaxation $U : \mathbb{R}^{n \times n} \to \mathbb{R}$ where the total tour cost is computed via the matrix product $\boldsymbol{P} = \boldsymbol{X} \cdot \boldsymbol{C}$, where $\boldsymbol{C} \in \mathbb{R}^{n \times 2}$ denotes city coordinates. The cost function, defined as the negative sum of Euclidean distances between consecutive cities $\sum_{i=1}^{n} \|\boldsymbol{p}_i - \boldsymbol{p}_{i+1}\|_2$, is differentiable with respect to the entries of $\boldsymbol{X}$. We compute $\nabla_{\boldsymbol{X}} U(\boldsymbol{X})$ using standard backpropagation, which provides a directional force toward configurations with shorter tour lengths.

### F.2.2 Why DLP is inadept for structural and cost similarity?

In principle, standard gradient-based samplers are topology-agnostic. One could theoretically impose structural and cost similarity directly by augmenting the energy function with an explicit regularizer, that penalizes local sensitivity, such as:

$$
U_{\text{new}}(\boldsymbol{\theta}) = U_{\text{original}}(\boldsymbol{\theta}) + \lambda \sum_{\boldsymbol{\theta}' \in \mathcal{N}(\boldsymbol{\theta})} \left( U(\boldsymbol{\theta}') - U(\boldsymbol{\theta}) \right)^2
$$

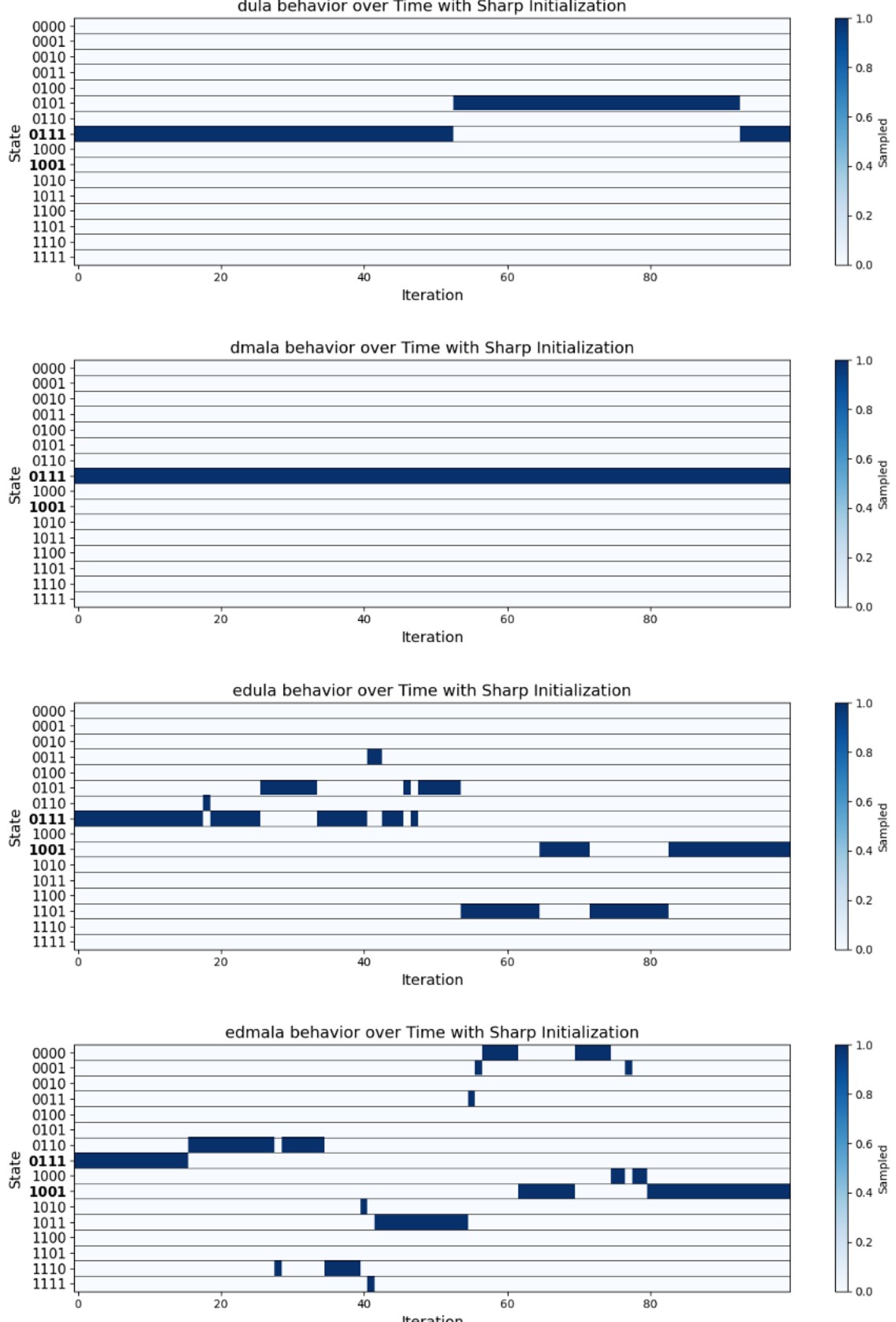

Figure 13: Finite-time behavior under sharp initialization.

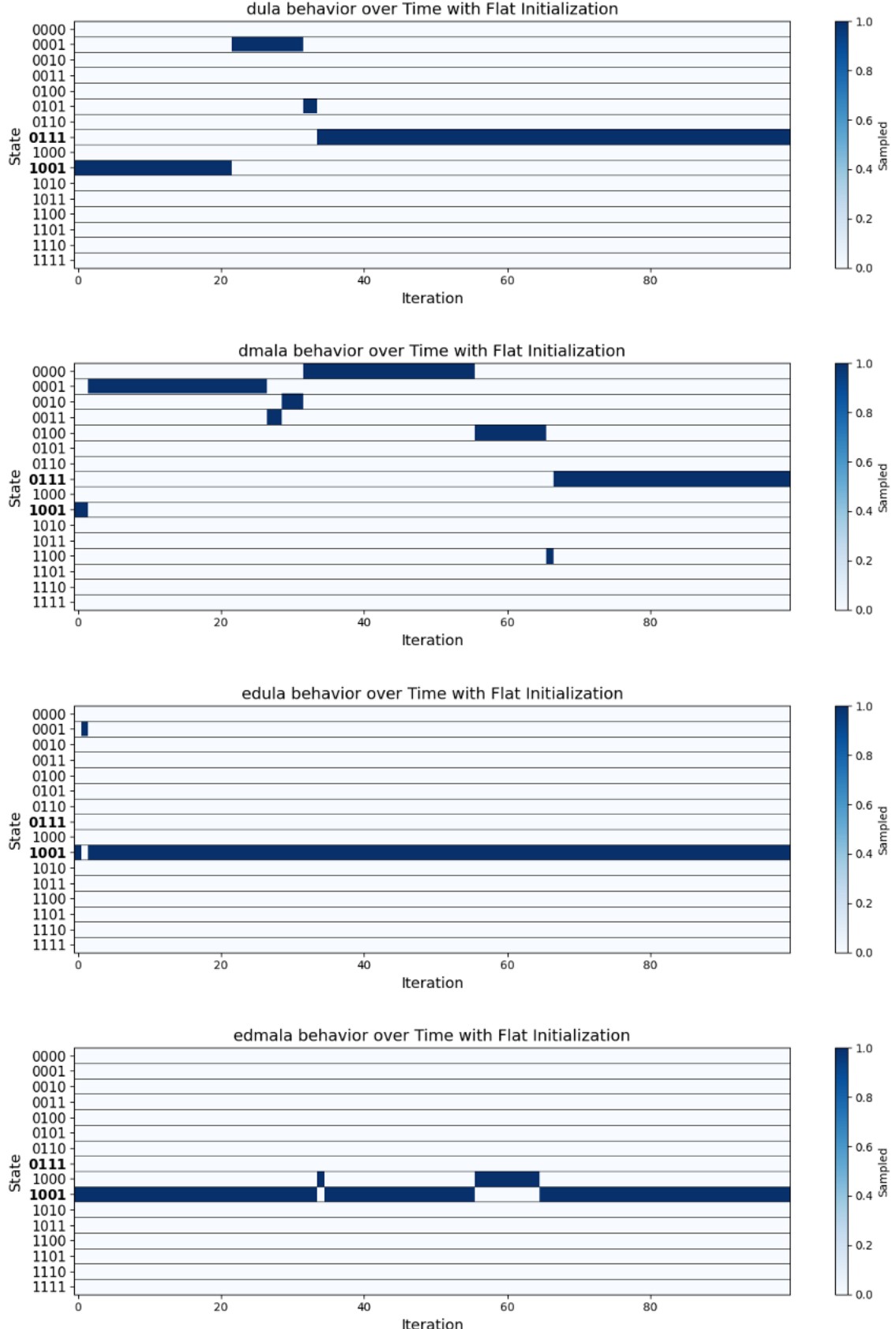

Figure 14: Finite-time behavior under flat initialization.

, where $\mathcal{N}(\boldsymbol{\theta})$ denotes the set of immediate neighbors (e.g., all states reachable by a single edge-swap). However, this approach is computationally intractable for large $n$ as calculating the sensitivity term requires evaluating the energy function for every possible neighbor for every sample.

EDLP introduces a topological bias via the local entropy coupling. The coupling effectively smooths the landscape, allowing the sampler to *see* the volume of the solution basin. Thus, we circumvent the need for expensive, explicit summation over the neighborhood, allowing us access to robust solutions.

### F.2.3   The Cost of Robustness.

Table 7: Solution quality characteristics across samplers.

| Sampler | Average Cost | Unique Solutions | Validity Rejection Rate |
|---------|--------------|------------------|-------------------------|
| DULA | $881.17 \pm 76.11$ | 108 | 0.9677 |
| DMALA | $727.13 \pm 67.81$ | 21 | 0.9677 |
| EDULA | $890.88 \pm 53.07$ | 44 | 0.9676 |
| EDMALA | $855.33 \pm 20.85$ | 13 | 0.9676 |

Recall, we enforce structural validity by rejecting any proposal that fails the unique-city-visit constraint (i.e., a configuration that does not correspond to a permutation matrix). The Validity Rejection Rate measures the frequency at which the sampler proposes structurally invalid tours. We present the average cost, number of unique solutions found by each sampler, and the Validity Rejection Rate under settings of Section 6 as illustrated in Table 7.

In the context of Constrained Combinatorial Optimization (CCO), our primary objective is not raw minimization of the energy function, but rather the identification of high-quality, structurally stable solution basins. Theoretically, the introduction of the auxiliary variable $\boldsymbol{\theta}_a$ fundamentally alters the effective target distribution. By coupling the discrete variable to a continuous proxy via Langevin dynamics, we are no longer optimizing the original sharp energy $U(\boldsymbol{\theta})$. Instead, we are sampling from a smoothed joint distribution that penalizes isolated, brittle minima(see Lemma 4.1 in Section 4.1). Consequently, one should not expect entropic samplers to strictly outperform pure gradient-based methods in finding the absolute lowest theoretical cost. If the global minimum lies in a region of high curvature (a sharp peak), EDLP is explicitly designed to ignore it in favor of flatter, slightly higher-energy regions that offer robustness.

Notably, EDLP produces significantly fewer unique solutions compared to DLP. This reduction is not a failure of exploration, but a success of focus. The entropic coupling offers a leashing effect, preventing the sampler from wandering freely into high-energy or structurally dissimilar regions. It confines the search to the effective volume of the flat mode, ensuring that every sampled solution is a viable, robust candidate for the practitioner. DMALA and EDMALA show an acceptance probability of 0.1571 and 0.1275 respectively.

Importantly, the structural validity filter (each row and column summing to 1) is applied to the candidate proposal *before* the Metropolis–Hastings step: invalid proposals are treated as self-transitions, which preserve detailed balance for $\pi$ restricted to the valid permutation set. The MH ratio is computed using the unrestricted proposal density $q(.)$, consistent with this self-transition interpretation, and is therefore correct for the constrained target.

The rejection rate reflects the difficulty of navigating the permutation manifold within the larger sequence space. With 30! valid tours embedded in $30^{30}$ possible sequences, a uniform random proposal would land in the valid set with probability $\approx 10^{-12}$. Our observed validity rejection rates of 96–97% therefore indicate that the gradient-based proposal is highly informative by concentrating mass on the valid manifold despite its extreme sparsity. Crucially, these rates are consistent across DLP and EDLP, amongst their MH and non-MH-corrected variants, confirming that the validity filter acts identically across samplers and that performance differences arise from the flatness-informed proposal distribution itself, not from the rejection mechanism.

### F.2.4   Robustness Stress Test: Structural Perturbations.

To validate that flat modes represent robust basins, we perform post-hoc structural stress tests on our generated solutions in Section 6 by performing Node-level Perturbation, by randomly swapping two cities in the tour. We then calculate the absolute difference in cost between the modified and original tour i.e. Average Absolute Cost Delta ($|\Delta \text{Cost}|$). A solution basin is robust if the absolute cost change $|\Delta \text{Cost}|$ under these perturbations is minimal. Our results from Figure 8 indicate that EDLP-sampled tours consistently

Table 8: Post-hoc Evaluation on Generated Solutions

| Sampler | $|\Delta \text{Cost}|(\downarrow)$ |
|---------|-----------------------|
| DULA    | $26.9438 \pm 22.7665$ |
| DMALA   | $44.4166 \pm 37.0412$ |
| EDULA   | $22.1218 \pm 18.5424$ |
| EDMALA  | $\mathbf{16.6346 \pm 13.9761}$ |

exhibit lower $|\Delta \text{Cost}|$ values than those from non-entropic baselines, demonstrating that they are embedded in broader, more stable regions of the cost landscape.

### F.3   RBM

The effective sample sizes (ESS) of BG-1, HB-10-1, DULA, DMALA, EDULA, and EDMALA are $10.0629 \pm 6.6461$, $9.4165 \pm 6.0812$, $10.0888 \pm 8.1389$, $10.1356 \pm 7.8493$, $10.1238 \pm 8.0024$, and $9.0384 \pm 7.1102$, respectively. The ESS values across all samplers are of the same order of magnitude indicating consistency in exploration. DMALA and EDMALA show an acceptance probability of $0.9980 \pm 0.0040$ and $0.9780 \pm 0.0015$ respectively.

### F.3.1   Perceptual Quality Analysis

Table 9: PSNR and SSIM of different samplers on MNIST under clean, noisy, and occluded conditions.

| Sampler | Setting | PSNR ($\uparrow$) | SSIM ($\uparrow$) |
|---------|---------|-------------------|-------------------|
| HB-10-1 | Clean | $16.3555 \pm 0.0858$ | $0.5303 \pm 0.0014$ |
|         | Noisy | $15.9763 \pm 0.0697$ | $\mathbf{0.3941} \pm 0.0035$ |
|         | Occluded | $16.2720 \pm 0.0749$ | $0.4963 \pm 0.0017$ |
| BG-1 | Clean | $16.2492 \pm 0.1125$ | $0.5294 \pm 0.0025$ |
|      | Noisy | $15.9086 \pm 0.0885$ | $0.3938 \pm 0.0038$ |
|      | Occluded | $16.1613 \pm 0.0992$ | $0.4947 \pm 0.0024$ |
| DULA | Clean | $16.1160 \pm 0.1022$ | $0.5114 \pm 0.0030$ |
|      | Noisy | $15.7851 \pm 0.0815$ | $0.3907 \pm 0.0041$ |
|      | Occluded | $16.0187 \pm 0.0922$ | $0.4766 \pm 0.0028$ |
| DMALA | Clean | $16.3305 \pm 0.0709$ | $0.5291 \pm 0.0035$ |
|       | Noisy | $15.9547 \pm 0.0623$ | $0.3939 \pm 0.0032$ |
|       | Occluded | $16.2372 \pm 0.0632$ | $0.4950 \pm 0.0035$ |
| EDULA | Clean | $16.2979 \pm 0.0891$ | $0.5296 \pm 0.0028$ |
|       | Noisy | $15.9145 \pm 0.0729$ | $0.3881 \pm 0.0038$ |
|       | Occluded | $16.2153 \pm 0.0796$ | $0.4949 \pm 0.0027$ |
| EDMALA | Clean | $\mathbf{16.4678} \pm 0.0920$ | $\mathbf{0.5388} \pm 0.0031$ |
|        | Noisy | $\mathbf{16.0490} \pm 0.0750$ | $0.3931 \pm 0.0037$ |
|        | Occluded | $\mathbf{16.3792} \pm 0.0819$ | $\mathbf{0.5043} \pm 0.0033$ |

To augment our analysis on perceptual quality, we report several widely accepted metrics: Peak Signal Noise Ratio (PSNR) to measure the fidelity of the reconstructed images, and the Structural Similarity Index (SSIM) to assess the structural integrity of the reconstructions (Wang et al., 2004). EDMALA attains highest PSNR and highest SSIM (except for Noisy) among the samplers tested.

### F.3.2   Mode Analysis

We performed mode analysis to validate the diversity and quality of MNIST digit samples generated by various samplers. Mode analysis assesses whether each sampler can capture the full range of MNIST digit

classes (0-9) without falling into *mode collapse*, a phenomenon where a generative model fails to represent certain data modes, thus limiting diversity. We leveraged a *LeNet-5 convolutional neural network* (LeCun et al., 1998) trained on MNIST to classify each generated sample and produce a class distribution for each sampler. The choice of LeNet-5, a reliable architecture for digit recognition, ensures accurate class predictions, thus providing a robust method to assess the representativeness of the samples. We train the model for 10 epochs, and achieve a 98.85% accuracy on test data.

The results (Figure 15) from our analysis indicated that all samplers produced samples across all digit classes, showing no evidence of mode collapse. Although certain samplers exhibited a preference for specific classes these biases did not reach the level of complete mode omission. Each class was represented in the generated samples, confirming that the samplers achieved an acceptable level of *mode diversity*. By confirming that all classes are covered, we demonstrate that each sampler can adequately approximate the diversity of the MNIST dataset, assuring the samples' representativeness (Salimans et al.; Goodfellow et al., 2014).

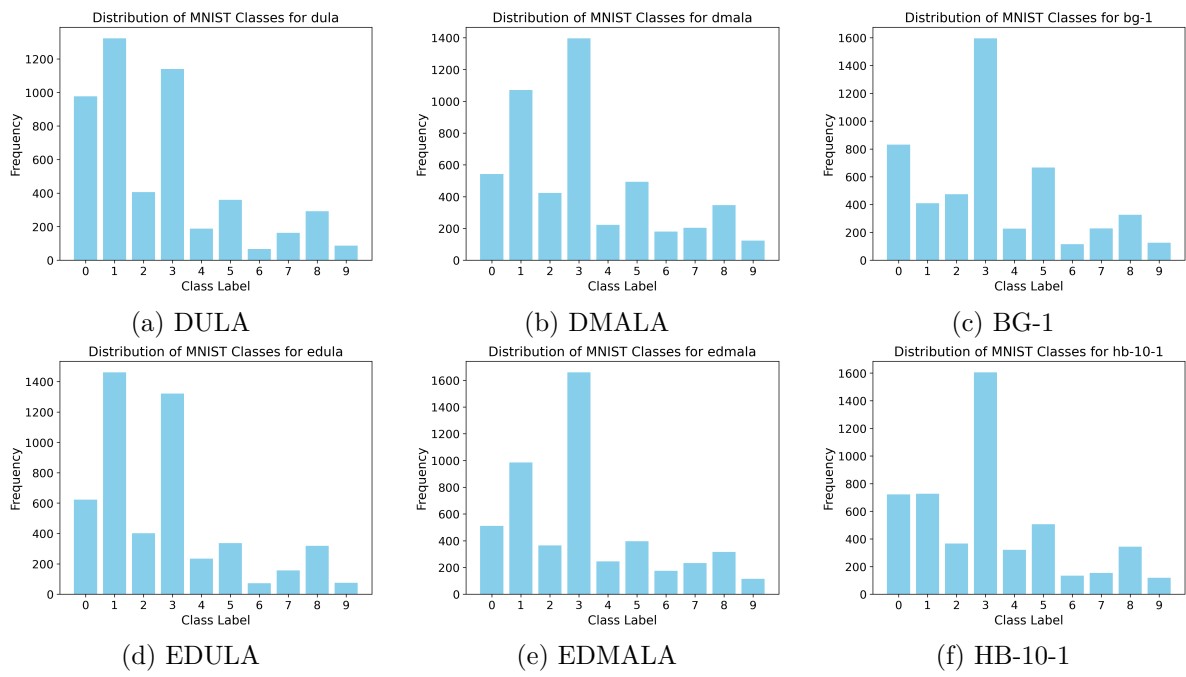

Figure 15: Mode Analysis

## F.4    BBNN

For the Adult Dataset, the ESS of Gibbs, GWG, DULA, DMALA, EDULA, and EDMALA are $1.1407 \pm 0.0201$, $1.2812 \pm 0.0110$, $11.6888 \pm 0.0613$, $11.1738 \pm 0.0712$, $12.6431 \pm 0.0571$, and $11.3632 \pm 0.0854$, respectively. Since the former two samplers only flip one coordinate at a time, their ESS remain poor compared to gradient based proposals like DLP and EDLP. DMALA and EDMALA show an acceptance probability of $0.860 \pm 0.050$ and $0.849 \pm 0.054$ respectively.

We report the Average Training Log-Likelihood for our experiments in Table 10. Importantly, when EDLP does not yield a substantial improvement, it still manages to avoid significantly impacting the training log-likelihood negatively. The sampling vector's dimensions are as follows: COMPAS(7,501), News(30,001), Adult(45,001), and Blog(27,801). For details of datasets used, refer to the Appendix of Zhang et al. (2022).

## F.5    Hyperparameters

All experiments were run on a single RTX A6000. The random seed used in all our experiments is `1234567`.

Table 10: Average Training Log-Likelihood

| Dataset | Gibbs | GWG | DULA | DMALA | EDULA | EDMALA |
|---|---|---|---|---|---|---|
| COMPAS | -0.4392 ± 0.0182 | -0.4091 ±0.0291 | -0.3611 ± 0.0134 | **-0.3443** ±0.0168 | -0.4147 ±0.0054 | -0.4203 ±0.0007 |
| News | -0.2372 ±0.0001 | -0.2375 ±0.0002 | -0.2348 ±0.0009 | -0.2345 ±0.0008 | **-0.2341** ±0.0007 | -0.2342 ±0.0008 |
| Adult | -0.4707 ±0.0043 | -0.4623 ±0.0070 | -0.3414 ± 0.0047 | -0.3228 ±0.0171 | -0.3246 ±0.0037 | **-0.3186** 0.0030 |
| Blog | -0.3920 ±0.0064 | -0.3682 ±0.0170 | -0.2708 ±0.011 | -0.2686 ±0.0114 | -0.2673 ±0.0032 | **-0.2654** ±0.0027 |

### F.5.1 Joint 4D Bernoulli

We use 4 chains. We run each sampler for 1000 iterations and burn the initial 200 samples. For more details on hyperparameters see Table 11.

Table 11: Hyperparameters for 4D Joint Bernoulli.

| Sampler | $\alpha$ | $\alpha_a$ | $\eta$ |
|---|---|---|---|
| DULA | 0.125 | - | - |
| EDULA | 0.200 | 0.100 | 0.100 |
| DMALA | 0.100 | - | - |
| EDMALA | 0.400 | 0.100 | 0.08 |

### F.5.2 TSP

We run our samplers for 100000 iterations with 50 refinements per sample, and burn 2000 initial iterations. For more details on hyperparameters see Table 12.

Table 12: Hyperparameters for TSP.

| Sampler | $\alpha$ | $\alpha_a$ | $\eta$ |
|---|---|---|---|
| DULA | 0.0001 | - | - |
| EDULA | 0.0001 | 0.001 | 0.1 |
| DMALA | 0.0001 | - | - |
| EDMALA | 0.0001 | 0.0001 | 0.1 |

### F.5.3 RBM

In our experiments, we generated 5000 images per sampler for the MNIST dataset, applying a thinning factor of 1000 to ensure diversity in the samples. We use a Gaussian noise with factor 0.1 to noise our samples. We train 5 CAEs in parallel. For more details on hyperparameters see Table 13.

### F.5.4 BBNN

We train 50 Binary Bayesian Neural Networks in parallel as in Section 6. We use $\alpha$ to 0.3 for the COMPAS dataset and $\alpha = 0.1$ for all other datasets. We run each sampler for 1200 iterations, with an initial burn-in of 200 samples. The tuning was done using validation-RMSE. For more details on hyperparameters see Table 14.

Table 13: Hyperparameters for RBM.

| Sampler | $\alpha$ | $\alpha_a$ | $\eta$ |
|---|---|---|---|
| DULA | 0.100 | - | - |
| EDULA | 0.200 | 0.0100 | 0.8 |
| DMALA | 0.1 | - | - |
| EDMALA | 0.08 | 0.01 | 0.25 |

Table 14: Hyper-parameter Settings for Binary Bayesian Neural Networks

| Dataset | EDULA | | EDMALA | |
|---|---|---|---|---|
| | $\alpha_a$ | $\eta$ | $\alpha_a$ | $\eta$ |
| COMPAS | 0.0100 | 4.00 | 0.0010 | 1.25 |
| News | 0.0040 | 2.00 | 0.0001 | 0.85 |
| Adult | 0.0001 | 2.00 | 0.0001 | 5.00 |
| Blog | 0.0100 | 1.00 | 0.0001 | 1.25 |

