# OpenReview forum: "Entropy-Guided Sampling of Flat Modes in Discrete Spaces"
_TMLR — Under review for TMLR_

### Review · Reviewer_q44D · 2026-04-21

**Summary Of Contributions:**

This paper proposes EDLP (Entropic Discrete Langevin Proposal), a modification of DLP for discrete sampling that introduces a continuous auxiliary variable and a quadratic coupling term. The idea is to use local entropy to steer the sampler toward flat basins rather than only sharp high probability modes. The paper gives two variants, EDULA and EDMALA, provides convergence results for both, and evaluates them on a synthetic Bernoulli example, TSP, RBMs on MNIST, and binary Bayesian neural networks.

**Strengths**

First of all, I think the joint construction $p(\theta,\theta_a)$ is neat. It gives a principled way to incorporate local entropy smoothing without relaxing the discrete variable itself. The sampler is not a vague flatness regularizer, rather it is a concrete MCMC proposal mechanism with a well-defined joint distribution.
Moreover, the theoretical results in Section 5 are core contribution.
Lastly, experiment settings are broad, not limited to one or a few toy examples.

**Weakneness**

The paper never really defines *flat mode* in a measurable way. The intuition is clear, but the evaluation relies on a loose notion of flatness. There are different ways of identifying flat mode depending on experiments, but nowhere do the authors give a single quantitative definition of flat mode occupancy that is measured across all experiments.
Furthermore, I feel the theory and experiments are weakly connected.
The paper’s theory assumes smoothness, bounded auxiliary space, and for EDMALA, local strong concavity. The introduction frames the theory as non-asymptotic convergence guarantees for locally log-concave discrete distributions. But the experiments are on: 1. TSP with validity constraints and rejection; 2. RBMs on MNIST; 3. binary BNN posteriors.
These are highly nonconvex and structurally very different from the analyzed regime.

**Audience:**

Yes

**Audience Explanation:**

The topic studied in the paper is relevant to ML researchers. Discrete sampling is widely used in various applications.

**Claims And Evidence:**

Yes

**Claims Explanation:**

While I found some weaknesses in the paper, I think the claim and results proposed in the paper are supported by concrete evidence and analysis. Also, the topics are relavent to ML researchers. Hence, the paper meets the TMLR's evaluation criteria.

**Requested Changes:**

Please find the weaknesses part above.

---

> ### Author Response · Authors · 2026-05-19
> **Rebuttal for Reviewer q44D**
>
> Dear Reviewer q44D,
>
> Thank you for providing valuable feedback on our work. We are delighted that you found our work **intuitive, principled, technically sound with concrete evidence and analysis, theoretically strong, and experimentally comprehensive**.
> ### 1. Measure of Flat Mode in Discrete Space
>
>
> We point out that local entropy (Equation 2, page 4) is our quantitative definition of flatness. It is a measurable scalar that captures the volume of energetically favorable neighbors around any state. It is a well-defined function used consistently throughout the paper as both the theoretical object and the algorithmic mechanism( fBP(Eqn 2. in [1]), EMCMC(Eqn 1. in [2])).
>
> Exact Local Entropy computation requires summation of the entire discrete state space i.e. $\mathcal{O}(|\Theta|)$ infeasible for large-scale applications across multiple chains. For instance, $|\Theta|=2^d$, where $d$=45k for BBNN. Hence we employ domain-appropriate, literature-wise established interpretable metrics validated to correlate with flatness benefits for downstream application: cost range and structural similarity in TSP (Table 1)— two tours in the same flat basin share many edges and have similar cost, surrogate NLL(negative log-likelihood) under perturbation in RBMs (Table 2), RMSE in BNNs (Table 3) as standard proxy for better generalization in flatness-awareness. This practice is aligned with [2], where Local Entropy is not computed and reported directly but Accuracy and Negative log-likelihood are used to demonstrate impact of flatness in Bayesian Deep Learning. Our Hessian eigenspectrum analysis (Figure 10), consistent with [2], provides additional geometric characterization.
>
>
>
> ### 2. Practicality of Assumptions and Theory-Experiment Connection
>
>
> **Boundedness of Auxiliary Space/Assumption 5.3**: Assumption 5.3 assumes boundedness on the auxiliary domain. This is supported empirically too. In practice, the auxiliary step-size $\alpha_a$ (typically tuned in the range $[0.001, 0.01]$) with practically useful value of η, ensures that $\theta_a$ remains tethered to the discrete target $\theta$, as evidenced by the stable $L_2$ norms reported in Figure 4.
>
> **Assumption 5.2/Local Strong Concavity**: We note that Assumption 5.2 requires *local* strong concavity in a neighborhood around each state, as opposed to EMCMC which requires overall strong convexity( Page 5, Assumption 1 in [2]).
>
> This makes our assumption explicitly compatible with the nonconvex landscapes in our experiments. Indeed, flat modes are characterized by regions where m_θ is small, which is permissible under this assumption
>
> **Standard Assumptions**: Assumptions 5.1 and 5.2 are standard in gradient-based discrete MCMC like ACS, HiSS[3,4], which have been evaluated on the same experimental regimes — RBMs, BBNNs, and TSP with rejection. Our theory-experiment relationship is therefore consistent with the broader discrete MCMC literature.
>
> We are grateful for your feedback. The clarifications requested have greatly helped strengthen the clarity of our work.
>
> [1] C. Baldassi,C. Borgs,J.T. Chayes,A. Ingrosso,C. Lucibello,L. Saglietti, & R. Zecchina,  Unreasonable effectiveness of learning neural networks: From accessible states and robust ensembles to basic algorithmic schemes, PNAS (2016).
>
> [2]  Bolian Li and Ruqi Zhang, Entropy-{MCMC}: Sampling from Flat Basins with Ease}, ICLR(2024)
>
> [3] Pynadath, Patrick, et al. "Gradient-based Discrete Sampling with Automatic Cyclical Scheduling." NeurIPS 2024
>
> [4] Pinaki Mohanty and Ruqi Zhang. Slithering Through Gaps: Capturing Discrete Isolated Modes via Logistic Bridging, AISTATS(2026)

---

### Review · Reviewer_pyY2 · 2026-04-22

**Summary Of Contributions:**

The authors introduce a new sampling approach, Entropic Discrete Langevin Proposal (EDLP), for exploring flat regions of discrete energy landscapes. Their proposed approach extends the Discrete Langevin Proposal (DLP) method by incorporating a local entropy regularizer (via a continuous auxiliary variable) to encourage sampling from a proposal distribution that is a `smoothed’ rendition of the target distribution. Based on EDLP, the authors introduce two sampling algorithms, EDULA (Entropic Discrete Unadjusted Langevin Algorithm) and EDMALA (Entropic Discrete Metropolis-Adjusted Langevin Algorithm) and provide theoretical support for their convergence. Lastly, through various toy and synthetic experiments (toy Bernoulli example, sampling for the travelling salesman problem, MNIST via Restricted Boltzmann Machines, and training binary Bayesian neural networks), the authors empirically demonstrate that EDLP enables improved sampling performance for problems requiring sampling from discrete energies with flat landscapes.

**Strengths**:

- The proposed method is straightforward to use and yields what appears to be definitive performance improvements across a variety of tasks.
- The authors provide detailed theoretical support for their proposed method.
- The authors evaluate their approach through four main (synthetic) empirical experiments.

**Weaknesses**:

- The experiments in this work all seem to be limited to synthetic/toy examples. I think it could be useful to apply this approach to some application settings in science; e.g. sampling molecules. For example, Tan et al. (2025) consider sampling molecules with Boltzmann generators, perhaps EDLP could be applied to this task. I don’t view lack of application experiments as a serious limitation, but believe it could help strengthen the paper.

References:

Tan, Charlie B., et al. "Scalable equilibrium sampling with sequential boltzmann generators." arXiv preprint arXiv:2502.18462 (2025).

**Audience:**

Yes

**Audience Explanation:**

Yes. I think this is a valuable direction of research aligned with recent efforts in improving sampling-based methods, particularly in the discrete setting.

**Broader Impact Concerns:**

This is primarily a theoretical work and does not carry any broader impact concerns.

**Claims And Evidence:**

Yes

**Claims Explanation:**

The central claim of this work is that the proposed method, EDLP, enables improved sampling from discrete energy functions with flat landscapes. In general, the experimental results seem to support this claim; however, it is not entirely obvious that some of the experimental settings, which are considered by the authors, require sampling from flat regions of the discrete space, specifically for the experiments in 6.2, 6.3, and 6.4. I encourage the authors to add additional discussion in these sections to elaborate on why these specific settings fit the requirement.

**Requested Changes:**

My requested changes are very minor:

- Provide further discussion in the experiments sections, elaborating why the settings considered require an algorithm for sampling in flat regions of discrete energy landscapes.
- If possible (and if applicable), provide an additional experiment in a scientific application domain (e.g. sampling molecules). This addition is more of a nice-to-have, and I do not view it as a deal breaker for this paper.

---

> ### Author Response · Authors · 2026-05-19
> **Rebuttal for Reviewer pyY2**
>
> Dear Reviewer pyY2,
>
> Thank you for providing valuable feedback on our work. In particular we are glad you found our work **straightforward with solid empirical benefits, theoretically strong, and experimentally comprehensive**.
>
> ### 1. Experimental Depth
> We would like to emphasize that in our current setup, only the 4D Joint Bernoulli experiment is a synthetic one (Section 6.1). All other experiments, conducted in Sections 6.2, 6.3, and 6.4, utilize real-world datasets. By demonstrating EDLP's application across six real-world datasets ( TSPLIB, MNIST, and 4 UCI datasets), our experimental setup is robust and closer to practical use. We have now added a paragraph before Section 6.1, better motivating the need for flatness based sampling for all our real-world settings.
>
>
> ###  2. Application for Scientific Modelling
> We thank the reviewer for the interesting suggestion. We note that Tan et al. (2025) operates in continuous Cartesian coordinate space, making direct application of EDLP, a discrete sampler, incompatible with that specific setting. However, we believe flatness-aware sampling has significant potential in discrete scientific domains, including protein sequence design, molecule generation, and allele configurations. Carefully identifying the need for flat mode sampling and validating performance using domain-appropriate measures is beyond our current scope, as we are not domain experts in aforementioned areas. We leave this as an exciting direction for future work and have added some discussion on this in Section 7.
>
> We are very grateful for your comments. Your assessment has helped position our work better and expand its scope.

---

### Review · Reviewer_3MHN · 2026-05-07

**Summary Of Contributions:**

The paper studies sampling from flat modes in discrete spaces. It proposes EDLP, a modification of the Discrete Langevin Proposal in which the discrete state is coupled to a continuous auxiliary variable through a quadratic term. This coupling induces a local-entropy or Moreau-envelope-like smoothing effect, and the resulting proposal is intended to guide the chain toward broad basins rather than isolated sharp modes. The authors instantiate the method in both unadjusted and Metropolis-adjusted forms, called EDULA and EDMALA, and provide non-asymptotic convergence statements under regularity and compactness assumptions. The empirical evaluation covers a small synthetic Bernoulli distribution, a TSP setting, RBM sampling on MNIST, and binary Bayesian neural networks on several tabular datasets.

The main strength of the paper is that it addresses a meaningful and somewhat underexplored question: how to exploit the geometry of discrete energy landscapes when flatness or robustness matters. The proposed algorithm is also simple: it modifies an existing gradient-informed discrete sampler by adding one continuous auxiliary variable and a coupling term. The paper makes a serious attempt to combine algorithm design, theory, and experiments rather than presenting only a heuristic.

The main weaknesses are that the central target-distribution story is not yet fully consistent, the theoretical assumptions and proofs do not convincingly support the broad claims made in the introduction, and several experiments measure downstream behavior rather than direct sampling correctness or flat-mode sampling fidelity. In particular, the paper alternates between saying that EDLP samples from the original target marginal and saying that it biases the stationary distribution toward flat modes. These are not the same claim. The experiments are suggestive, but in the current form they do not fully resolve this ambiguity.

**Additional Comments:**

The core idea is worth pursuing, but the current version may overstate what has been established. I would be much more positive if the authors clarified whether EDLP is a correct sampler for the original target, a sampler for a modified flatness-aware target, or a finite-time heuristic for finding broad basins. I am willing to update my assesment positively should the reviewers address my concerns.

**Audience:**

Yes

**Audience Explanation:**

The topic is relevant for a part of the TMLR audience. Discrete sampling remains important for energy-based models, combinatorial optimization, probabilistic inference, etc. over structured spaces. The idea of importing flatness-aware or local-entropy ideas into discrete MCMC is potentially useful, especially if it can be made precise as either a better proposal mechanism for the original target or as a controlled way to sample from a deliberately flatness-biased distribution.

The paper also tries to connect several communities: gradient-informed discrete MCMC, flat-minima methods, proximal or entropy-smoothed sampling, and robust combinatorial solution generation. I would expect some readers interested in discrete Langevin methods, BNNs, and robust discrete optimization to find the direction useful.

The issue is not lack of interest; it is that the current version needs a more careful statement of what distribution is being sampled, what guarantees are actually proved, and what the experiments demonstrate.

**Broader Impact Concerns:**

I do not see a severe broader-impact concern specific to the sampling algorithm itself.

**Claims And Evidence:**

No

**Claims Explanation:**

The paper contains an interesting idea and several suggestive experiments, but I do not think the current version supports its strongest claims with sufficiently accurate and clear evidence.

My main concern is the target-distribution interpretation. The paper introduces the joint distribution

$$
p(\theta,\theta_a) \propto \exp\left(U(\theta)-\frac{1}{2\eta}\|\theta-\theta_a\|^2\right),
$$

and states that its marginal over $\theta$ is the original target distribution. This is correct when $\theta_a$ is integrated over $\mathbb{R}^d$. However, the paper also repeatedly describes EDLP as biasing the sampler or the stationary distribution toward flat modes. These two claims need to be disentangled. If EDMALA is a valid MH sampler for the stated joint target, then its long-run marginal over $\theta$ should be the original target, not a flatness-reweighted target. Flatness preference may still appear as a finite-time proposal effect, or if the intended target is actually a modified distribution, but the current text does not make this distinction cleanly. This ambiguity affects the interpretation of the algorithm, the theory, and the experiments.

There is also a technical inconsistency between the marginal argument and the assumptions used for the theory. Lemma 4.1 relies on integrating the auxiliary variable over $\mathbb{R}^d$. Later, the convergence analysis assumes that $\theta_a$ lies in a compact set, and the appendix discusses truncated-normal updates. Under such truncation, the marginal over $\theta$ is generally no longer the original $\pi(\theta)$, unless additional conditions are imposed. The algorithm in the main text, however, appears to use an ordinary Gaussian Langevin update. The paper should therefore clarify whether the implemented sampler is truncated or untruncated, and what distribution it targets in each case.

I am also not fully convinced by the current convergence analysis. The theory is based on smoothness, compactness, and strong-concavity-type assumptions that are far from the multimodal discrete landscapes used to motivate the paper. This is not necessarily fatal, but the paper should be more explicit that the theory applies to a restricted setting and does not directly justify the empirical behavior on TSP, RBMs, or BNNs. Some of the bounds also appear to scale poorly with dimension and geometry, so the practical meaning of the non-asymptotic rates is unclear.

For EDULA, the claimed reversibility with respect to a “ghost” stationary distribution needs a more careful justification. Unadjusted Langevin-type chains do not generally inherit reversibility automatically, so this part of the proof should be made more explicit. I was also concerned by the presence of the partition function $Z$ in the bias bound of Proposition 5.4, since the distribution is invariant to additive shifts of $U$, whereas such a bound may not be. This suggests that either the bound needs to be reformulated or the dependence on normalization needs to be explained.

The empirical evidence is broad but not yet decisive. The 4D Bernoulli example is useful as an intuition check, but because the state space is tiny, the paper should provide exact diagnostics such as total variation distance, KL divergence, exact state frequencies, flat-vs-sharp basin mass, and acceptance rates/autocorrelation. The TSP experiment shows that EDLP samples routes that are more structurally similar and have a narrower cost range, but this is not the same as demonstrating robustness unless an explicit perturbation or downstream constraint test is provided. Moreover, the appendix shows that EDMALA can have worse average cost than DMALA, so the tradeoff between robustness and raw objective quality needs to be stated more carefully.

The RBM/MNIST evaluation is also indirect. Training a convolutional autoencoder on sampler-generated images and reporting reconstruction metrics does not directly establish that the sampler approximates the RBM distribution better or samples flat modes more faithfully. The reported “log-likelihood” should be precisely defined, since a standard autoencoder does not by itself provide a likelihood unless a probabilistic decoder or explicit density model is specified. The BBNN results are more aligned with the flat-minima motivation, but several improvements are small relative to the reported variability, and the paper should include clearer information about seeds, validation, hyperparameter tuning, and statistical significance.

Overall, I think the paper studies a worthwhile problem and proposes a potentially useful modification of DLP. However, the current evidence does not yet support the full set of claims about principled flat-mode sampling, target fidelity, non-asymptotic convergence, and consistent empirical superiority.

**Requested Changes:**

- **Critical: Clarify the target distribution and the meaning of flatness bias.**
   The paper could explicitly distinguish between three possible interpretations:
   (i) EDLP is a proposal mechanism whose MH-corrected version samples the original target marginal $\pi(\theta)$;
   (ii) EDLP samples a deliberately modified flatness-aware distribution over $\theta$;
   (iii) EDLP is mainly a finite-time mode-seeking heuristic that favors broad basins before full equilibration.
   These are different claims. The current text moves between them, and the paper should commit to a precise interpretation.

- **Critical: Resolve the compactness/truncation issue for $\theta_a$.**
   Lemma 4.1 uses integration over $\mathbb{R}^d$, while the convergence analysis assumes a compact auxiliary space. If the actual algorithm uses truncated-normal updates, this should be stated in Algorithm 1 and in the experimental setup. If the actual algorithm uses untruncated Gaussian updates, then the compactness assumption is not implemented. If compact truncation is used, the paper should explain whether the marginal over $\theta$ remains the original target or becomes a modified target.

- **Critical: Make the EDMALA target precise.**
   The MH acceptance ratio should be tied explicitly to the invariant distribution of the chain. If EDMALA targets the joint distribution in Lemma 4.1, then the paper should not claim that the stationary marginal over $\theta$ is biased toward flat modes. If it targets a different flatness-aware distribution, that distribution should be written down and the MH ratio should be derived for it.

- **Critical: Revisit the proof of Proposition 5.4.**
   The argument that EDULA is reversible with respect to a ghost distribution needs to be made fully explicit. For an unadjusted Langevin-type chain, reversibility is not automatic. The proof should either provide a clean detailed-balance verification or use a different invariant-measure perturbation argument. The dependence of the bound on the partition function $Z$ should also be checked, since the bound should not change under additive shifts of the energy $U$.

- **Critical: State the real scope of the theory.**
   The assumptions used in the convergence analysis are strong and do not appear to cover the main empirical settings. The paper should state clearly that the theory applies to a restricted locally log-concave or compactified setting, and should avoid implying that the same guarantees explain the behavior on TSP, RBM/MNIST, or BBNNs.

- **Add direct sampling diagnostics on the small Bernoulli example.**
   Since the state space has only 16 states, the paper should report exact total variation distance, KL divergence, exact state frequencies, flat-mode mass, sharp-mode mass, and acceptance rates/autocorrelation. This example is the cleanest place to verify whether the sampler is preserving the target distribution, biasing toward flat modes, or showing only finite-time preference.

- **Compare methods under equal computational budget, or state the difference.**
   EDLP introduces an auxiliary variable and, for EDMALA, a more complex acceptance ratio. The paper should include equal wall-clock or equal gradient-evaluation comparisons in the main experimental section. Equal-iteration comparisons are not sufficient to establish practical improvement.

- ** Strengthen the TSP robustness evaluation.**
   The TSP experiment currently reports pairwise mismatch count, Jaccard similarity, and range of cost. These are useful, but they do not directly test robustness. The paper should add an explicit perturbation-based or constraint-based evaluation, for example edge deletions, random cost perturbations, budget violations after local changes, or post-hoc feasibility under additional constraints. This would better support the claim that flat modes correspond to robust solution basins.

- **Clarify the TSP state representation and validity mechanism.**
   The paper represents tours as binary square matrices and rejects invalid proposals. It should give more detail on how gradients are computed for this representation, how often invalid proposals occur, whether rejection rates differ across samplers, and whether this validity filter changes the comparison.

- ** Clarify the RBM/MNIST evaluation.**
   The paper should define exactly what is meant by “log-likelihood” in Table 2. If the quantity comes from a convolutional autoencoder, then it is not a likelihood unless the autoencoder has an explicit probabilistic decoder or density model. The paper should either justify this metric formally or replace the wording with a more accurate reconstruction or surrogate score.

- **Important: Add stronger statistical reporting for the BBNN results.**
   The BNN experiments should report the number of independent seeds/splits, whether comparisons are paired, how hyperparameters are selected, and whether improvements are statistically meaningful. Some improvements are small relative to the reported standard deviations, so claims of consistent superiority should be moderated unless supported by significance or confidence intervals.

- ** Include a table of distributions and notation.**
   The paper uses several related distributions: the original target $\pi(\theta)$, the smoothed auxiliary marginal over $\theta_a$, the coupled joint distribution, the EDULA stationary distribution, and the EDMALA target. A short table would substantially improve clarity.

- **Reposition claims about novelty and scope.**
   Claims such as “first such bound” and broad statements about principled flatness-aware discrete sampling should be made more carefully. The paper should compare more precisely to locally informed proposals, auxiliary-variable MCMC for discrete variables, proximal sampling, entropy-MCMC, and focused belief propagation.

- **Important: Discuss the role of $\eta$ more concretely.**
   The sensitivity analysis is useful, but the main text should explain how $\eta$ affects finite-time flatness preference, asymptotic target fidelity, acceptance probability, and computational cost. The paper should also provide practical tuning guidance.

- **Important: Report acceptance rates and mixing diagnostics.**
   For the MH-adjusted samplers, acceptance rates should be reported across experiments. For all samplers, autocorrelation, effective sample size, or related mixing diagnostics would help determine whether improvements come from better exploration or from a changed sampling target.

- **Important: Make the relation to local entropy more explicit.**
   The paper should clarify whether the algorithm is maximizing or sampling according to local entropy, using local entropy only as an auxiliary smoothing device, or inducing local-entropy behavior through finite-time dynamics. These are related but distinct statements.

- **Move some technical overview material to the main method/theory sections.**
   The introduction contains a long technical overview that partly repeats later material and includes argumentative language. The paper would be clearer if the introduction stated the problem, method, and evidence more compactly, leaving technical details to Sections 4 and 5.


- **Add reproducibility details.**
   The paper should include the number of chains, burn-in, thinning, number of seeds, hardware, wall-clock budgets, and hyperparameter grids in one consolidated location.

---

> ### Author Response · Authors · 2026-05-19
> **Rebuttal for Reviewer 3MHN #1**
>
> Dear Reviewer 3MHN,
>
> Thank you for providing us with detailed feedback on our work. We are delighted to know that you found our work **impactful, straightforward, and tackling a problem that is underexplored**. We also value your feedback on how our work effectively **bridges the gap between algorithm design, theory, and practicality**.
>
>
> ### 1. What are we really sampling from?
>  We clarify EDLP is an MCMC algorithm whose main utility lies in seeking flat discrete basins in finite time, before full equilibration. The motivation is identical to EMCMC[1] i.e. targetting flat modes under a computational budget.
>
> From Bottom of Page 18 in [1],
> > "we are confident to claim that Entropy-MCMC prioritizes flatmodes under limited computational budget and will eventually converge to the fullposterior with adequate iterations".
>
> The flatness-seeking behavior is a non-asymptotic (finite-time) property of the chain's trajectory. In the limit of infinite iterations, the chain converges to the original marginal $\pi(\theta)$. In finite time, however, the coupling effectively 'steers' the proposal toward flat basins, providing a computational advantage in finding robust solutions before full equilibrium is reached.
>
> We have now better signposted this throughout our manuscript.
>
>
> ### 2. Compactness/Truncation of Auxiliary Variable  and  Theory-Experiment Scope
> We confirm for our experiments we use Gaussian Langevine Dynamics without truncation. We have now added a footnote stating that our assumptions(5.1-5.3) are purely for the ease of theoretical analysis and our experimental setup extends beyond this, similar to the one on Page 5 in [1]. We have now clarified the Remark in the Appendix. Also, please see #2 with Reviewer q44D.
>
> ### 3. EDMALA's target
> This is tied to #1. EDMALA is indeed the formal sampler for the Joint Distribution( Lemma 4.1), but the Joint Distribution itself is what creates the "flatness bias" in the first place. By construction, this joint target is explicitly 'flatness-aware' because the auxiliary variable $\theta_a$ smooths the energy landscape. Therefore, the steering toward flat modes comes from sampling from this smoothed joint distribution, in finite time. This is why our convergence guarantees are non-asymptotic in nature.
>
> We follow the established practice in prior flatness-aware MCMC methods like EMCMC[1] to derive non-asymptotic bounds for the joint distribution. Specifically we state in our manuscript:
>
> > Note that the convergence of the chains for both EDULA and EDMALA imply convergence of the marginals as the projection maps are continuous. In fact, deriving a rate of convergence for them is also possible, but we omit it here as that is not the goal of this paper. We focus our analysis on the joint distribution to better capture the flatness-seeking behavior induced by the auxiliary variable( as in Theorem 1 in~\citet{li2024entropymcmc}) in finite time.
>
>
> ### 4. Revisit the proof of Proposition 5.4.
> The reversibility of the EDULA chain with respect to the ghost distribution $\pi_\gamma$ is formally derived in our proof. As shown in the detailed balance verification, the transition kernel $q_\gamma$ and the ghost distribution $\pi_\gamma$ satisfy the symmetry condition: $\pi_\gamma(\tilde{\theta})q_\gamma(\tilde{\theta}'|\tilde{\theta}) = \pi_\gamma(\tilde{\theta}')q_\gamma(\tilde{\theta}|\tilde{\theta}')$.
>
> We confirm that the dependence on Z for EDULA is mathematically correct and is a consequence of building on DULA's backbone( Theorem 5.1 in [2]) as Z shows up as multiplicative factor in its asymptotic bias too.
>
> However, we do see the Reviewer's point on additional shifts in U. For EDULA's analysis, we replace Z with $|{\Theta}| \cdot \exp\left( \frac{M}{2} diam( {\Theta} ) \right)$. This naturally handles the shift arguement. We kept the Z factor to directly compare to DULA's asymptotic convergence.
>
> Irrespective of how we handle it, the idea is to show tha bias vanishes as $\alpha \to 0$.
>
> ### 5. Bernoulli 4D example Diagnostics
>
> We now provide KL, TVD for all 4 samplers in the main-text. We also provide MH-Acceptance Probabilities for DMALA and EDMALA. As requested we report frequency counts for all 4 samplers to highlight flat-mode preference in finite time. To support our claim that EDLP is a valid MCMC algorithm, we ran our samplers for longer duration to gauge Asymptotic behavior and report decreasing TVD and KL in the Appendix.
>
> ### 6. Timings for TSP and additional Results
> We have now added average sample time to highlight the computational tradeoff in Main Text. We have added discussion on gradient computation for our TSP problem(Appendix). We have also reported the Validity Rejection Rates and found them to be almost identical across DLP and EDLP( Appendix). Based on the Reviewer's recommendation, we have added an addition post-hoc verification for our generated samples, to strengthen our claim on flat basins housing robust solutions(Appendix).

---

> ### Author Response · Authors · 2026-05-19
> **Rebuttal for Reviewer 3MHN #2**
>
> ### 7. RBM/MNIST
> Thank You for pointing this out. We now clarify in the main-text that we use negative reconstruction error as a surrogate for negative log-likelihood(NLL). We also report the average time-taken-per-sample in the main-text. We report MH-acceptance probabilities and ESS in Appendix.
>
> ### 8. BBNN
> Our experimental setup—including dataset choices, the number of independent chains (50) is identical to the protocol established in Section 7.4 in [2]. Our sampling dimensions range from 7k to 45k. We confirm all samplers are evaluated on the same train-test split. The hyperparameter tuning was performed using validation-RMSE. We have softened the language: "consistently outperforms" to "demonstrates improved performance" in our main-text. For the sake of brevity and clarity, we present the ESS and MH acceptance on our most computationally challenging dataset, the Adult dataset(highest dimensionality, $d=45k$, $|\Theta|\approx 3.7011 \times 10^{13546}$) in the Appendix. We also direct our readers to Runtime Analysis done on Adult Dataset in Figure 5.
>
>
> ### 9. Novelty Claims, Scope, and Technical Overview
> We have now softened the novely claim in our introduction. Based on the Reviewer's recommendation, we have decided to dissolve the technical overview section throughout the manuscript to improve readibility and focus on EDLP's comparison to related works. We have added more discussion around locally informed proposals, entropy-MCMC, and focused belief propagation.
>
> ###  10. Discuss the role of $\eta$ more concretely.
> We now have a dedicated section on this in the main-text under Section 5.2, where we discuss the  impact of flatness parameter $\eta$ on finite-time flatness preference and asymptotic target fidelity.
>
> ### 11. Role of Local Entropy
> We clarify that EDLP induces local-entropy behavior through finite-time dynamics. The role of Local Entropy hence has been clarified in the 'Conclusion' and 'Why does EDLP work?'.
>
> ### 12. Table on Distributions and Hyperparameters
> We have now added a table on distributions for our readers. We have now included Hyperparameters and reproducibility details like chains, burin, thinning factor, at one consolidated location.
>
> ----
>
> In the meanwhile, we have also added working urls/links to all our citations for completeness. We have also front-loaded Theorem 5.7 (CLT emergence) from Appendix to Main-Text.
>
> Once again, we deeply appreciate your feedback on our work. Your comments have been extremely helpful in improving the quality and overall presentation of the manuscript.
>
>
> [1]  Bolian Li and Ruqi Zhang, Entropy-{MCMC}: Sampling from Flat Basins with Ease}, ICLR(2024)
>
> [2] Zhang, Ruqi, et al. "A Langevin-like Sampler for Discrete Distributions." ICML 2022

---

### Author Response · Authors · 2026-05-26
**Summary for AE**

Dear Action Editor,

Thank you for overseeing the review of our paper and for assembling a thorough and constructive set of reviews. We are grateful for the reviewers' time and detailed engagement with our work. All changes in the revised manuscript are highlighted in blue for ease of reference.

All three reviewers agree that the problem of flat-mode sampling in discrete spaces is a meaningful, interesting, and underexplored direction, and that EDLP is a principled and potentially impactful contribution to this area. Reviewers pyY2 and q44D were broadly positive, and their concerns centered primarily on better motivating why the experimental settings — TSP, RBMs, and binary BNNs — specifically require or benefit from flat-mode sampling. We have addressed this by adding a dedicated short paragraph before Section 6.1 that motivates the need for flatness-aware sampling for real world settings. Regarding the definition of flatness in discrete spaces, we clarify that local entropy (Equation 2) is our precise, quantitative definition with origins in seminal flatness-aware works. Direct computation of local entropy is infeasible at scale, so we employ downstream metrics to capture its impact, consistent with the practice in Entropy-MCMC. Regarding theoretical assumptions, Assumption 5.2 requires only local strong concavity in a neighborhood around each state — a strictly weaker and more practical condition than the global strong convexity, compatible with non-convex regimes. To remove any residual concern, we have also added a footnote explicitly stating that our assumptions are for the ease of theoretical analysis, and that our experimental setups extend beyond this regime, again consistent with established practice in the literature.

Reviewer 3MHN provided an extensive and detailed feedback. They liked the paper's core idea and its attempt to bridge algorithm design, theory, and experiments, but requested additional evidence and significant narrative clarification — particularly around what distribution EDLP targets, the compactness assumption for the auxiliary variable, and the scope of the theoretical guarantees. We have addressed each of these critical points in full: we now clearly position EDLP as a finite-time flat-mode-seeking algorithm whose asymptotic marginal recovers the original target (consistent with the framing in the Entropy-MCMC paper), we have resolved the truncation ambiguity, and we have scoped the theory more carefully as requested. On the experimental side, we have added exact diagnostics (KL, TVD, state frequencies) to the 4D Bernoulli example, wall-clock timing comparisons, TSP post-hoc robustness evaluation, clarified the RBM metric as a reconstruction surrogate, and added ESS and acceptance rate reporting across experiments. We believe every critical and important point raised by Reviewer 3MHN has been addressed in the revised manuscript.

The modifications made to the manuscript do not alter the core contributions of the paper. The changes are clarifications on existing claims, improvements to presentation, and additional experiments requested by the reviewers. The algorithm, theory, and key empirical findings remain intact. We believe the revisions have substantially strengthened the paper and are happy to provide any further clarification.

Best,

The Authors